# CS4ML: A general framework for active learning with arbitrary data based on Christoffel functions

**Ben Adcock**
Department of Mathematics
Simon Fraser University
ben_adcock@sfu.ca

**Juan M. Cardenas**
Department of Mathematics
Simon Fraser University
juan_manuel_cardenas@sfu.ca

**Nick Dexter**
Department of Scientific Computing
Florida State University
nick.dexter@fsu.edu

## Abstract

We introduce a general framework for active learning in regression problems. Our framework extends the standard setup by allowing for general types of data, rather than merely pointwise samples of the target function. This generalization covers many cases of practical interest, such as data acquired in transform domains (e.g., Fourier data), vector-valued data (e.g., gradient-augmented data), data acquired along continuous curves, and, multimodal data (i.e., combinations of different types of measurements). Our framework considers random sampling according to a finite number of sampling measures and arbitrary nonlinear approximation spaces (model classes). We introduce the concept of *generalized Christoffel functions* and show how these can be used to optimize the sampling measures. We prove that this leads to near-optimal sample complexity in various important cases. This paper focuses on applications in scientific computing, where active learning is often desirable, since it is usually expensive to generate data. We demonstrate the efficacy of our framework for gradient-augmented learning with polynomials, Magnetic Resonance Imaging (MRI) using generative models and adaptive sampling for solving PDEs using Physics-Informed Neural Networks (PINNs).

## 1   Introduction

The standard regression problem in machine learning involves learning an approximation to a function $f^* : D \subseteq \mathbb{R}^d \to \mathbb{R}$ from training data $\{(\theta_i, f^*(\theta_i))\}_{i=1}^m \subset D \times \mathbb{R}$. The approximation is sought in a set of functions $\mathbb{F}$, typically termed a *model class*, *hypothesis set* or *approximation space*, and is often computed by minimizing the empirical error (or risk) over the training set, i.e.,

$$\hat{f} \in \underset{f \in \mathbb{F}}{\operatorname{argmin}} \frac{1}{m} \sum_{i=1}^m |f^*(\theta_i) - f(\theta_i)|^2.$$

In this paper, we develop a generalization of this problem. This allows for general types of data (i.e., not just discrete function samples), including multimodal data, and random sampling from arbitrary distributions. In particular, this framework facilitates *active learning* by allowing one to optimize the sampling distributions to obtain near-best generalization from as few samples as possible.

37th Conference on Neural Information Processing Systems (NeurIPS 2023).

## 1.1 Motivations

Typically, the sample points $\theta_i$ in the above problem are drawn i.i.d. from some fixed distribution. Yet, in many applications of machine learning there is substantial freedom to choose the sample points in a more judicious manner to increase the generalization performance of the learned approximation. These applications are often highly *data starved*, thus making a good choice of sample points extremely valuable. Notably, many recent applications of machine learning in scientific computing often have these characteristics. An incomplete list of such applications include: Deep Learning (DL) for Partial Differential Equations (PDEs) via so-called Physics-Informed Neural Networks (PINNs) [39, 61, 83, 105, 112]; deep learning for parametric PDEs and operator learning [17, 49, 53, 66, 73, 75, 118]; machine learning for computational imaging [5, 79, 84, 88, 100, 107, 110]; and machine learning for discovering dynamics of physical systems [24, 25, 109].

In many such applications, the training data does not consist of pointwise samples of a target function. For example, in computational imaging, the samples are obtained through an integral transform such as the Fourier transform in Magnetic Resonance Imaging (MRI) or the Radon transform in X-ray Computed Tomography (CT) [5, 13, 21, 45]. In other applications, each sample may correspond to the value of $f^*$ (or some integral transform thereof) along a continuous curve. Applications where this occurs include: seismic imaging, where physical sensors record continuously in time but at discrete spatial locations; discovering governing equations of physical systems, where each sample may be a full solution trajectory [109]; MRI, since the MR scanner samples along a sequence of continuous curves in $k$-space [80, 89]; and many more. In other applications, each sample may be a vector. For instance, in *gradient-augmented* learning problems – with applications to PINNs for PDEs [54, 120] and learning parametric PDEs in Uncertainty Quantification (UQ) [9, 56, 78, 82, 103, 98, 99], as well as various other DL settings [36] – each sample is a vector in $\mathbb{R}^{d+1}$ containing the values of $f$ and its gradient at $\theta_i$. In other applications, each sample may be an element of a function space. This occurs in parametric PDEs and operator learning, where each sample is the solution of a PDE (i.e., an element of a Hilbert space) corresponding to the given parameter value. Finally, many problems involve *multimodal data*, where one has $C > 1$ different types of training data. This situation occurs in various applications, including: multi-sensor imaging systems [33] such as parallel MRI (which is ubiquitous in modern clinical practice) [5, 89]; PINNs for PDEs, where the data arises from the domain, the initial conditions and the boundary conditions; and multifidelity modelling [44, 102].

## 1.2 Contributions

In this paper, we introduce a general framework for active learning in which

1. $f^*$ need not be a scalar-valued function, but simply an element of a Hilbert space $\mathbb{X}$;
2. the data arises from arbitrary linear operators, which may be scalar- or vector-valued;
3. the data may be multimodal, arising from $C \geq 1$ of different types of linear operators;
4. the approximation space $\mathbb{F}$ may be an arbitrary linear or nonlinear space;
5. sampling is random according to arbitrary probability measures;
6. the *sample complexity*, i.e., the relation between $\mathbb{F}$ and the number of samples required to attain a near-best generalization error, is given explicitly in terms of the so-called *generalized Christoffel functions* of $\mathbb{F}$;
7. using this explicit expression, the sampling measures can be optimized, leading to essentially optimal sample complexity in various cases.

This framework, *Christoffel Sampling for Machine Learning (CS4ML)*, is very general. Indeed, we are unaware of any other active learning strategy that can simultaneously address such a broad class of problems. It is inspired by previous work on function regression in linear spaces [2, 34, 57, 60] and is also a generalization of *leverage score sampling* [11, 30, 32, 41, 46, 50, 85] for active learning. See Appendix A for further discussion. It generalizes these approaches by (a) considering abstract objects in Hilbert spaces, (b) allowing for arbitrary linear sampling operators, as opposed to just function samples, (c) allowing for multimodal data, and (d) considering general linear or nonlinear approximation spaces. We demonstrate its practical efficacy on a diverse set of test problems. These are: **(i) Polynomial regression with gradient-augmented data**, **(ii) MRI reconstruction using generative models** and **(iii) Adaptive sampling for solving PDEs with PINNs**. Our solution to (iii) introduces a general adaptive sampling strategy, termed *Christoffel Adaptive Sampling*, that can be applied to any DL-based regression problem (i.e., not simply PINNs for PDEs).

## 2 The CS4ML framework and main results

### 2.1 Main definitions

Let $(\Omega, \mathcal{F}, \mathbb{P})$ be a probability space, which is the underlying probability space of the problem. In particular, our various results hold in probability with respect to the probability measure $\mathbb{P}$. Let $\mathbb{X}$ be a separable Hilbert space and $\mathbb{X}_0 \subseteq \mathbb{X}$ be a normed vector subspace of $\mathbb{X}$, termed the *object space*. Our goal is to learn an element (the object) $f^* \in \mathbb{X}_0$ from training data. We consider a multi-modal measurement model in which $C \geq 1$ different processes generate this data. For each $c = 1, \ldots, C$, we assume there is a measure space $(D_c, \mathcal{A}_c, \rho_c)$, termed the *measurement domain*, which parametrizes the possible measurements. We will often consider the case where this is a probability space, but this is not strictly necessary for the moment. Next, we assume there are semi-inner product spaces $\mathbb{Y}_c$, $c = 1, \ldots, C$, termed *measurement spaces*, to which the measurements belong, and mappings

$$L_c : D_c \to \mathcal{B}(\mathbb{X}_0, \mathbb{Y}_c), \quad c = 1, \ldots, C,$$

where $\mathcal{B}(\mathbb{X}_0, \mathbb{Y}_c)$ denotes the space of bounded linear operators $\mathbb{X}_0 \to \mathbb{Y}_c$. We refer to $L_c$ as *sampling operator* for the $c$th measurement process. Note that we consider the sampling operators $L_c$ as fixed and specified by the problem under consideration. However, we make the following assumption, which states that we have the flexibility to query each sampling operator at an arbitrary point in the measurement domain.

**Assumption 2.1 (Active learning)** For each $c$, we may query $L_c(\theta)(f^*)$ at any $\theta \in D_c$.

This assumption is a direct extension of that made in standard active learning (see Example 2.3 below). It holds in all our main examples. Given this assumption, our approach is to selecting sample points is to draw them randomly according to certain *sampling measures* $\mu_1, \ldots, \mu_C$ (defined on $D_1, \ldots, D_C$, respectively), which can then be optimized to ensure as good generalization as possible. To this end, we also make the following assumption.

**Assumption 2.2 (Sampling measures)** Each sampling measure $\mu_c$ is a probability measure on $(D_c, \mathcal{A}_c)$ that is absolutely continuous with respect to $\rho_c$ and its Radon–Nikodym derivative is positive almost everywhere. In other words, $\mathrm{d}\mu_c(\theta) = \nu_c(\theta)\,\mathrm{d}\rho_c(\theta)$ for some measurable function $\nu_c : D_c \to \mathbb{R}$ that positive almost everywhere and satisfies $\int_{D_c} \nu_c(\theta)\,\mathrm{d}\rho_c(\theta) = 1$.

We now define the training data. Let $m_1, \ldots, m_C$ be the (not necessarily equal) measurement numbers, i.e., $m_c$ is the number of measurements from the $c$th measurement process. Let $\Theta_c : \Omega \to D_c$ be independent $D_c$-valued random variables, where $\Theta_c \sim \mu_c$ for each $c$. For each $c = 1, \ldots, C$, let $\Theta_{ic}$, $i = 1, \ldots, m_c$, be independent realizations of $\Theta_c$. Then the training data is

$$\{(\Theta_{ic}, y_{ic})\}_{i,c=1}^{m_c, C}, \quad \text{where } y_{ic} = L_c(\Theta_{ic})(f^*) \in \mathbb{Y}_c, \ i = 1, \ldots, m_c \ c = 1, \ldots, C. \qquad (2.1)$$

Later, in Section 4.3 we also allow for noisy measurements. Note that we consider the *agnostic learning* setting, where $f^* \notin \mathbb{F}$ and the noise can be adversarial, as this setting is most appropriate to the various applications considered (see also [50]).

**Example 2.3 (Active learning in standard regression)** The above framework extends the standard active learning problem in regression. In the classic regression problem, $D \subseteq \mathbb{R}^d$ is a domain and $\mathbb{X} = L^2_\rho(D)$ is the space of square-integrable functions $f^* : D \to \mathbb{R}$ with respect to a measure $\rho$. Note that $\rho$ is considered fixed – it is the measure with respect to which we measure the error. To embed this problem into the above framework, we let $\mathbb{X}_0 = C(\overline{D})$ be the space of continuous functions on $D$, $C = 1$, the measurement domain $D_c = D$ be equal to the domain of the function, $\rho_1 = \rho$ and the measurement space $\mathbb{Y}_1 = \mathbb{R}$ (with the Euclidean inner product). We then define the sampling operator $L_1(\theta)(f^*) = f^*(\theta)$ as the pointwise evaluation operator. In particular, for a measure $\mu = \mu_1$ satisfying Assumption 2.2, the training data (2.1) is $(\Theta_i, f^*(\Theta_i))$ with $\Theta_i \sim_{\text{i.i.d.}} \mu$. Hence, the aim is to choose the measure $\mu$ (or equivalently, its Radon-Nikodym derivative $\nu$) to ensure as good generalization as possible.

Next, we let $\mathbb{F} \subseteq \mathbb{X}_0$ be a subset within which we seek to learn $f^*$. We term $\mathbb{F}$ the *approximation space*. Note this could be a linear space such as a space of algebraic or trigonometric polynomials, or a nonlinear space such as the space of sparse Fourier functions, a space of functions with sparse

representations in a multiscale system such as wavelets, the space corresponding to a single neuron model, or a space of deep neural networks. We discuss a number of examples later. Given $\mathbb{F}$ and $f^* \in \mathbb{X}_0$, in this work we consider an approximation $\hat{f} \in \mathbb{F}$ defined via the empirical least-squares fit

$$\hat{f} \in \underset{f \in \mathbb{F}}{\operatorname{argmin}} \sum_{c=1}^{C} \frac{1}{m_c} \sum_{i=1}^{m_c} \frac{1}{\nu_c(\Theta_{ic})} \|y_{ic} - L_c(\Theta_{ic})(f)\|_{\mathbb{Y}_c}^2. \tag{2.2}$$

However, regardless of the method employed, one cannot expect to learn $f^*$ from (2.1) without an assumption on the sampling operators. Indeed, consider the case $L_c \equiv 0, \forall c$. The following assumption states that the sampling operators should be sufficiently 'rich' to approximately preserve the $\mathbb{X}$-norm. As we see later, this is a natural assumption, which holds in many cases of interest.

**Assumption 2.4 (Nondegeneracy of the sampling operators)** The mappings $L_c$ are such that the maps $D_c \to \mathbb{Y}_c, \theta \mapsto L_c(\theta)(f)$ are measurable for every $f \in \mathbb{X}_0$. Moreover, the functions $D_c \to \mathbb{C}, \theta \mapsto \|L_c(\theta)(f)\|_{\mathbb{Y}_c}^2$ are integrable and satisfy, for constants $0 < \alpha \le \beta < \infty$,

$$\alpha \|f\|_{\mathbb{X}}^2 \le \sum_{c=1}^{C} \int_{D_c} \|L_c(\theta)(f)\|_{\mathbb{Y}_c}^2 \, d\rho_c(\theta) \le \beta \|f\|_{\mathbb{X}}^2, \quad \forall f \in \mathbb{X}_0. \tag{2.3}$$

In order to successfully learn $f^*$ from the samples (2.1), we need to (approximately) preserve nondegeneracy when the integrals in (2.3) are replaced by discrete sums. Notice that

$$\alpha \|f\|_{\mathbb{X}}^2 \le \mathbb{E} \left( \sum_{c=1}^{C} \frac{1}{m_c} \sum_{i=1}^{m_c} \frac{1}{\nu(\Theta_{ic})} \|L_c(\Theta_{ic})(f)\|_{\mathbb{Y}_c}^2 \right) \le \beta \|f\|_{\mathbb{X}}^2, \quad \forall f \in \mathbb{X}_0, \tag{2.4}$$

where $\mathbb{E}$ denotes the expectation with respect to all the random variables $(\Theta_{ic})_{i,c}$. Our subsequent analysis involves deriving conditions under which this holds with high probability. We shall say that *empirical nondegeneracy* holds with constant $0 < \epsilon < 1$ for a draw $(\Theta_{ic})_{i,c}$ if

$$(1-\epsilon)\alpha \|f\|_{\mathbb{X}}^2 \le \sum_{c=1}^{C} \frac{1}{m_c} \sum_{i=1}^{m_c} \frac{1}{\nu(\Theta_{ic})} \|L_c(\Theta_{ic})(f)\|_{\mathbb{Y}_c}^2 \le (1+\epsilon)\beta \|f\|_{\mathbb{X}}^2, \quad \forall f \in \mathbb{F} - \mathbb{F}, \tag{2.5}$$

where $\mathbb{F} - \mathbb{F} = \{f_1 - f_2 : f_1, f_2 \in \mathbb{F}\}$. See Appendix A for some background on this definition.

Finally, we need one further assumption. This assumption essentially says that the action of the sampling operator $L_c$ on $\mathbb{F}$ is nontrivial, in the sense that, for almost all $\theta \in D_c$, there exists a $f \in \mathbb{F}$ that has a nonzero measurement $L_c(\theta)(f)$. This assumption is reasonable, since without it, there is little hope to fit the measurements with an approximation from the space $\mathbb{F}$.

**Assumption 2.5 (Nondegeneracy of $\mathbb{F}$ with respect to $L_c$)** For each $c = 1, \ldots, C$ and almost every $\theta_c \in D_c$, there exists an element $f_c \in \mathbb{F}$ such that $L_c(\theta_c)(f_c) \ne 0$.

## 2.2 Summary of main results

The goal of this work is to derive sampling measures that ensure a quasi-optimal generalization bound from as little training data as possible. These measures are given in terms of the following function.

**Definition 2.6** (Generalized Christoffel function). Let $\mathbb{X}_0$ be normed vector space, $\mathbb{Y}$ be a semi-inner product space, $(D, \mathcal{A}, \rho)$ be a measure space, $L : D \to \mathcal{B}(\mathbb{X}_0, \mathbb{Y})$ be such that the function $D \to \mathbb{Y}, \theta \mapsto L(\theta)(f)$ is measurable for every $f \in \mathbb{X}_0$ and $\mathbb{F} \subseteq \mathbb{X}_0$, $\mathbb{F} \ne \{0\}$. The *Generalized Christoffel function of $\mathbb{F}$ with respect to $L$* is the function

$$K(\mathbb{F})(\theta) = \sup\{\|L(\theta)(f)\|_{\mathbb{Y}}^2 / \|f\|_{\mathbb{X}}^2 : f \in \mathbb{F}, \ f \ne 0\}, \quad \theta \in D.$$

If $\mathbb{F} = \{0\}$, then we set $K(\mathbb{F})(\theta) = 0$. Further, we also define $\kappa(\mathbb{F}) = \int_D K(\mathbb{F})(\theta) \, d\rho(\theta)$.

In the standard regression problem (see Example 2.3), Definition 2.6 reduces to

$$K(\mathbb{F})(\theta) = \sup\{|f(\theta)|^2 / \|f\|_{L_\rho^2(D)}^2 : f \in \mathbb{F}, \ f \ne 0\}, \quad \theta \in D. \tag{2.6}$$

This is a well-known object, which is often referred to as the *Christoffel function* when $\mathbb{F}$ is a linear subspace. It is also equivalent (up to a scaling) to the *leverage score function*. See Appendix A.2 for further discussion. Definition 2.6 extends these notions from linear subspaces to nonlinear spaces and from pointwise samples to arbitrary sampling operators $L$.

To state our main result, we also need the following. Given $\sigma > 0$ we define the *shrinkage* operator $\mathcal{T}_\sigma : \mathbb{X} \to \mathbb{X}$ by $\mathcal{T}_\sigma(f) = \min\{1, \sigma/\|f\|_{\mathbb{X}}\} f, \forall f \in \mathbb{X}$.

**Theorem 2.7.** *Consider the setup of Section 2.1, where $\mathbb{F} - \mathbb{F} = \cup_{j=1}^d \mathbb{F}_j$ is a union of $d \in \mathbb{N}$ subspaces of dimension at most $n \in \mathbb{N}$. Let $K_c$, $c = 1, \ldots, C$, be as in Definition 2.6 for $(D, \mathcal{A}, \rho) = (D_c, \mathcal{A}_c, \rho_c)$ and $L = L_c$. Suppose that the $K_c(\mathbb{F} - \mathbb{F})$ are integrable, define*

$$\mathrm{d}\mu_c^\star(\theta) = K_c(\mathbb{F} - \mathbb{F})(\theta)\,\mathrm{d}\rho_c(\theta)/\kappa_c(\mathbb{F} - \mathbb{F}), \quad c = 1, \ldots, C, \tag{2.7}$$

*and suppose that*

$$m_c \asymp \alpha^{-1} \cdot \kappa_c(\mathbb{F} - \mathbb{F}) \cdot \log(2nd/\delta), \quad c = 1, \ldots, C \tag{2.8}$$

*for some $0 < \delta < 1$. Then the following hold.*

   (i) *Empirical nondegeneracy (2.5) holds with $\epsilon = 1/2$ with probability at least $1 - \delta$.*
   (ii) *For any $f^* \in \mathbb{X}_0$, $\|f^*\|_{\mathbb{X}} \leq 1$, the estimator $\check{f} = \mathcal{T}_1(\hat{f})$, where $\hat{f}$ is as in (2.2), satisfies*

$$\mathbb{E}\|f^* - \check{f}\|_{\mathbb{X}}^2 \lesssim (\beta/\alpha) \cdot \inf_{f \in \mathbb{F}} \|f^* - f\|_{\mathbb{X}}^2 + \delta. \tag{2.9}$$

   (iii) *The total number of samples $m = m_1 + \cdots + m_C$ is at most log-linear in $nd$, namely,*

$$m \lesssim (\beta/\alpha) \cdot nd \cdot \log(2nd/\delta). \tag{2.10}$$

The choice (2.7) is a particular type of importance sampling, which we term *Christoffel Sampling (CS)*. In particular, for the standard regression problem (see Example 2.3) the resulting measure is

$$\mathrm{d}\mu^\star(y) \propto K(\mathbb{F} - \mathbb{F})(y)\,\mathrm{d}\rho(y), \tag{2.11}$$

where $K(\mathbb{F} - \mathbb{F})(y)$ is as in (2.6) with $\mathbb{F}$ replaced by $\mathbb{F} - \mathbb{F}$. As we discuss in Appendix A.2, this is equivalent to *leverage score sampling* for the standard regression problem (see, e.g., [2, 12, 34, 32, 46, 60]). Therefore, Theorem 2.7 can be considered a generalization of leverage score sampling from standard regression to significantly more general types of linear, multimodal measurements. Specifically, it considers the general setting of arbitrary Hilbert spaces $\mathbb{X}$, nonlinear approximation spaces $\mathbb{F}$, and $C > 1$ arbitrary, linear sampling operators $L_c$ taking values in arbitrary semi-inner product spaces $\mathbb{Y}_c$. To the best of our knowledge this generalization is new. We remark, however, that Theorem 2.7 is also related to the recent work [43]. See Appendix A.3 for further discussion.

The sampling measure (2.7) is 'optimal' in the sense that it minimizes a sufficient condition over all possible sampling measures $\mu_c$ (see Lemma 4.6). Doing so leads to CS and Theorem 2.7. The measurement condition (2.10) states that CS leads to a desirable sample complexity bound (2.10) that is at worse log-linear in $nd$. In particular, when $d = 1$ (i.e., $\mathbb{F}$ is a linear space) the sample complexity is near-optimal (and in this case, one also has $\mathbb{F} - \mathbb{F} = \mathbb{F}$). In general, the bound (2.10) is near-optimal in terms of $n$ when $d$ is small (and independent of $n$). Unfortunately, $d$ can be large in important cases such as sparse regression. In this case, however, the linear dependence on $d$ in (2.10) – which follows from (2.8) via the crude estimate $\sum_{c=1}^C \kappa_c(\mathbb{F} - \mathbb{F}) \leq \beta nd$ – can be lessened by using the specific structure of the approximation space. See Appendix A.4.

Theorem 2.7 is a simplification and combination of our several main results. In Theorem 4.2 we consider more general approximation spaces, which need not be unions of finite-dimensional subspaces. See also Remark 4.5. Finally, we note that $\check{f}$ only differs from $\hat{f}$ by a shrinkage operator, which is a technical step needed to bound the error in expectation. See Section 4.3. An 'in probability' bound can be obtained without this additional complication, albeit with less succinct right-hand side.

## 3   Examples and numerical experiments

Theorem 2.7 shows that CS can be a near-optimal active learning strategy. We now show its efficacy via a series of examples. These examples also highlight how the generality of the framework allows it to tackle a wide range of problems. We consider cases where the theory applies directly and others

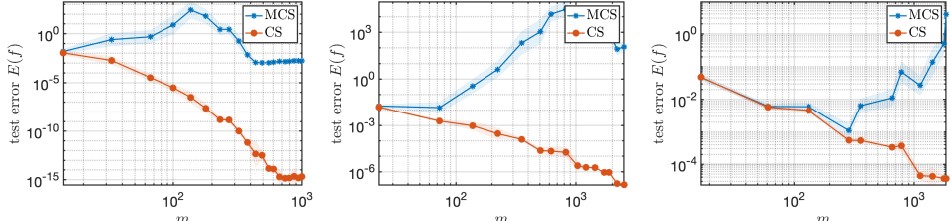

Figure 1: **CS for gradient-augmented polynomial regression.** Plots of the average regression error versus $m$ for MCS and CS for gradient-augmented polynomial regression of the function $f^*(\theta) = \exp(-(\theta_1 + \ldots + \theta_d)/(2d))$ for $d = 2, 4, 8$ (left to right). In both cases, we use the scaling $m = \lceil \max\{n, n\log(n)\}/(d+1) \rceil$. In this and other figures, the solid lines show the mean value and the shaded region indicates one standard deviation. See Section B.5 for further details.

where it does not. In all cases, we present performance gains over inactive learning, i.e., *Monte Carlo Sampling (MCS)* from the underlying probability measure. Another matter we address is how to sample from the measures (2.7) in practice, this being very much dependent on the problem considered. Full experimental details for each application can be found in Appendices B–D.

### 3.1 Polynomial regression with gradient-augmented data

Many machine learning applications involve regressing a smooth, multivariate function using a model class consisting of algebraic polynomials. Often, the training data consists of function values. Yet, as noted in Section 1.1, there are many settings where one can acquire both function values and values of its gradient. In this case, the training data takes the form $\{(\theta_i, f(\theta_i), \nabla f(\theta_i))\}_{i=1}^m \subset \mathbb{R}^d \times \mathbb{R}^{d+1}$. As we explain in Appendix B.1, this problem fits into our framework with $C = 1$, where $\rho$ is the standard Gaussian measure on $\mathbb{R}^d$, $\mathbb{X} = H_\rho^1(\mathbb{R}^d)$ is the Sobolev space of weighted square-integrable functions with weighted square-integrable first-order derivatives and $\mathbb{Y} = \mathbb{R}^{d+1}$ with the Euclidean inner product. It therefore provides a first justification for allowing non-scalar valued sampling operators. Note that several important variations on this problem also fit into our general framework with $C > 1$. See Appendix B.7.

We consider learning a function from such training data in a sequence of nested polynomial spaces $\mathbb{F}_1 \subset \mathbb{F}_2 \subset \cdots$. These spaces are based on *hyperbolic cross* index sets, which are particularly useful in multivariate polynomial regression tasks [1, 37]. See Appendix B.2 for the formal definition.

In Fig. 1 we compare gradient-augmented polynomial regression with CS versus MCS from the Gaussian measure $\rho$. See Appendices B.3–B.5 for details on how we implement CS in this case. CS gives a dramatic improvement over MCS. CS is theoretically near-optimal in the sense of Theorem 2.7 – i.e., it provably yields a log-linear sample complexity – since the approximation spaces are linear subspaces in this example. On the other hand, MCS fails, with the error either not decreasing or diverging. The reason is the log-linear scaling, which is not sufficient to ensure a generalization error bound of the form (2.9) for MCS. As we demonstrate in Appendix B.6, MCS requires a much more severe scaling of $m$ with $n$ to ensure a generalization bound of this type.

### 3.2 MRI reconstruction using generative models

Reconstructing an image from measurements is a fundamental task in science, engineering and industry. In many applications – in particular, medical imaging modalities such as MRI – one wishes to reduce the number of measurements while still retaining image quality. As noted, techniques based on DL have recently led to significant breakthroughs in image recovery tasks. One promising approach involves using generative models [16, 20, 67]. First, a generative model is trained on a database of relevant images, e.g., brain images in the case of MRI. Then the image recovery problem is formulated as a regression problem, such as (2.2), where $\mathbb{F}$ is the range of the generative model. Note that in this problem, the training data consists of a finite set of frequencies and the corresponding values of the Fourier transform of the unknown image.

As we explain in Appendices C.1–C.3, this problem fits into our general framework. In Fig. 2 we demonstrate the efficacy of CS for Fourier imaging with generative models. In this example, the

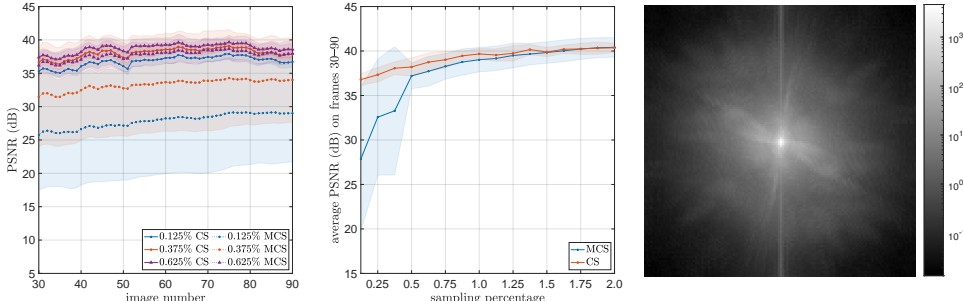

Figure 2: **CS for MRI reconstruction using generative models.** Plots of (left) the average PSNR vs. image number of the 3-dimensional brain MR image for both CS and MCS methods at 0.125%, 0.375%, and 0.625% sampling percentages, (middle) the average PSNR computed over the frames 30 to 90 of the image vs. sampling percentage for both CS and MCS methods, and (right) the empirical function $\widetilde{K}$ used for the CS procedure.

generative model was trained on a database of 3D MRI brain images (see Appendix C.5). This experiment simulates a 3D image reconstruction problem in MRI, where the measurements are samples of the Fourier transform of the unknown image taken along horizontal lines in $k$-space (a sampling strategy commonly known as *phase encoding*). The active learning problem involves judiciously choosing $m$ horizontal lines in $k$-space to enhance the generalization performance of the learning procedure. In Fig. 2, we compare the average Peak Signal-to-Noise Ratio (PSNR) versus frame (i.e., 2D image slice) number for CS versus MCS (i.e., uniform random sampling) for reconstructing an unknown image. We observe a significant improvement, especially in the challenging regime where the sampling percentage (the ratio of the number of measurements to the image size) is low.

This example lies close to our main theorem, but is not fully covered by it. See Appendix C.3. The space $\mathbb{F}$ (the range of a generative model) is not a finite union of subspaces. However, it is known that certain generative models (namely, those based on ReLU activation functions) are subsets of unions of subspaces of controllable size [16]. Further, we do not sample exactly from (2.7) in this case, but rather an empirical approximation to it (see Appendix C.4). Nevertheless, our experiments show a significant performance gain from CS in this case, despite it lying strictly outside of our theory.

This application justifies the presence of arbitrary (i.e., non-pointwise) sampling operators in our framework. It is another instance of non-scalar valued sampling operators, since each measurement is a vector of frequencies values along a horizontal line in $k$-space. Fig. 2 considers $C = 1$ sampling operators. However, as we explain Appendix C.7, the important extension of this setup to *parallel* MRI (which is standard in clinical practice) can be formulated in our framework with $C > 1$.

### 3.3  Adaptive sampling for solving PDEs with PINNs

In our final example, we apply this framework to solving PDEs via Physics-Informed Neural Networks (PINNs). PINNs have recently shown great potential and garnered great interest for approximating solutions of PDEs [42, 105, 112]. It is typical in PINNs to generate samples via Monte Carlo Sampling (MCS). Yet, this suffers from a number of limitations, including low accuracy.

We use this general framework combined with the *adaptive basis viewpoint* [35] to devise a new adaptive sampling procedure for PINNs. Here, a Deep Neural Network (DNN) with $n$ nodes in its penultimate layer is viewed as an element of the linear subspace spanned the $n$ functions defined by this layer's nodes. Our method then proceeds as follows. First, we use an initial set of $m = m_1$ samples and train the corresponding PINN $\Psi = \Psi_1$. Then, we use the adaptive basis viewpoint to construct a subspace $\mathbb{F}_1$ with $\Psi_1 \in \mathbb{F}_1$. Next, we draw $m_2 - m_1$ samples using CS for the subspace $\mathbb{F}_1$, and use the set of $m = m_2$ samples to train a new PINN $\Psi = \Psi_2$, using the weights and biases of $\Psi_1$ as the initialization. We then repeat this process, alternating between generating new samples via CS and retraining the network, to obtain a sequence of PINNs $\Psi_1, \Psi_2, \ldots$ approximating the solution of the PDE. We term this procedure *Christoffel Adaptive Sampling (CAS)*. See Appendices D.1–D.4 for further information.

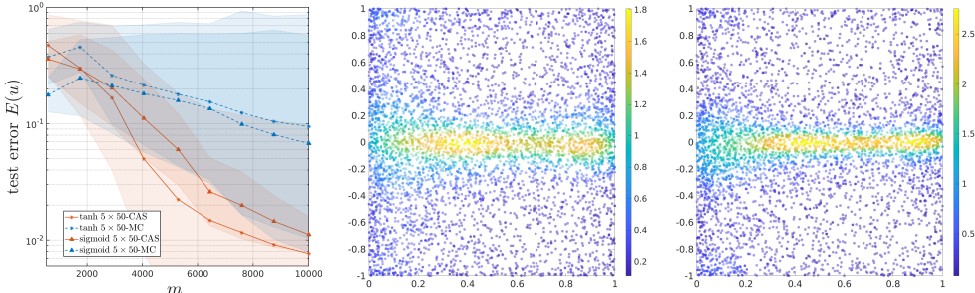

Figure 3: **CAS for solving PDEs with PINNs.** Plots of (left) the average test error $E(u^*)$ versus the number of samples used for training $\tanh$ and sigmoid $5 \times 50$ DNNs using CAS (solid line) and MCS (dashed line), (middle) the samples generated by the CAS for the $\tanh$ $5 \times 50$ DNN and (right) the samples generated by CAS for the sigmoid $5 \times 50$ DNN. The colour indicates the density of the points.

In Fig. 3 we show that this procedure gives a significant benefit over MCS in terms of the number of samples needed to reach a given accuracy. See Appendix D.5 for further details on the experimental setup. Note that both approaches use the same DNN architecture and are trained in exactly the same way (optimizer, learning rate schedule, number of epochs). Thus, the benefit is fully derived from the sampling strategy. The PDE considered (Burger's equations) exhibits shock formation as time increases. As can be seen in Fig. 3, CAS adapts to the unknown PDE solution by clustering samples near this shock, to better recover the solution than MCS.

As we explain in Appendix D.2, this example also justifies $C > 1$ in our general framework, as we have three sampling operators related to the PDE and its initial and boundary conditions. We note, however, that this example falls outside our theoretical analysis, since the sampling operator stemming from the PDE is nonlinear and the method is implemented in an adaptive fashion. In spite of this, however, we still see a nontrivial performance boost from the CS-based scheme.

## 4 Theoretical analysis

In this section, we present our theoretical analysis. Proofs can be found in Appendix E.

### 4.1 Sample complexity

We first establish a sufficient condition for empirical nondegeneracy (2.5) to hold in terms of the numbers of samples $m_c$ and the generalized Christoffel functions of suitable spaces.

**Definition 4.1** (Subspace covering number). Let $(\mathbb{X}, \|\cdot\|_{\mathbb{X}})$ be a quasisemi-normed vector space, $\mathbb{F}$ be a subset of $\mathbb{X}$, $n \in \mathbb{N}_0$ and $t \geq 0$. A collection of subspaces $\mathbb{F}_1, \ldots, \mathbb{F}_d$ of $\mathbb{X}$ of dimension at most $n$ is a $(\|\cdot\|_{\mathbb{X}}, n, t)$-*subspace covering* of $\mathbb{F}$ if for every $f \in \mathbb{F}$ there exists a $j \in \{1, \ldots, d\}$ and a $f_j \in \mathbb{F}_j \cap \mathbb{F}$ such that $\|f - f_j\|_{\mathbb{X}} \leq t$. The *subspace covering number* $\mathcal{N}(\mathbb{F}, \|\cdot\|_{\mathbb{X}}, n, t)$ of $\mathbb{F}$ is the smallest $d$ such that there exists a $(\|\cdot\|_{\mathbb{X}}, n, t)$-subspace covering of $\mathbb{F}$ consisting of $d$ subspaces.

In this definition, we consider a zero-dimensional subspace as a singleton $\{x\} \subseteq \mathbb{X}$. Therefore, the subspace covering number $\mathcal{N}(\mathbb{F}, \|\cdot\|_{\mathbb{X}}, 0, t)$ is precisely the classical covering number $\mathcal{N}(\mathbb{F}, \|\cdot\|_{\mathbb{X}}, t)$. We also remark that if $\mathbb{F} = \cup_{j=1}^{d} \mathbb{F}_j$ is itself a union of $d$ subspaces of dimension at most $n$, then $\mathcal{N}(\mathbb{F}, \|\cdot\|_{\mathbb{X}}, n', t) = d$ for any $t \geq 0$ and $n' \geq n$.

We now also require the following notation. If $\mathbb{F} \subseteq \mathbb{X}$, we set $\mathcal{B}(\mathbb{F}) = \{f / \|f\|_{\mathbb{X}} : f \in \mathbb{F} \backslash \{0\}\}$.

**Theorem 4.2** (Sample complexity for empirical nondegeneracy). *Consider the setup of Section 2.1. Let $\mathbb{F} \subseteq \mathbb{X}_0$, $n \in \mathbb{N}_0$, $0 < \epsilon, \delta < 1$ and $\mathbb{F}_1, \ldots, \mathbb{F}_d$ be a $(\|\cdot\|, n, t)$-subspace covering of $\mathcal{B}(\mathbb{F} - \mathbb{F})$, where $t = \sqrt{\alpha\epsilon/2}$ and $\|\cdot\|^2 = \sum_{c=1}^{C} \operatorname{ess\,sup}_{\theta_c \sim \rho_c} \{\|L_c(\theta_c)(\cdot)\|_{\mathbb{Y}_c}^2 / \nu(\theta_c)\}$. Suppose that*

$$m_c \geq c_{\epsilon/2} \cdot \alpha^{-1} \cdot \operatorname*{ess\,sup}_{\theta \sim \rho_c} \left\{ K_c(\widetilde{\mathbb{F}})(\theta) / \nu_c(\theta) \right\} \cdot \log(2d \max\{n, 1\}/\delta), \quad \forall c = 1, \ldots, C, \quad (4.1)$$

*where $c_{\epsilon/2}$ is as in (E.6) and $\widetilde{\mathbb{F}} = \cup_{i=1}^{d} \mathbb{F}_i$. Then (2.5) holds with probability at least $1 - \delta$.*

This theorem gives the desired condition (4.1) for empirical nondegeneracy (2.5). It is interesting to understand when this condition can be replaced by one involving the function $K_c$ evaluated over $\mathbb{F} - \mathbb{F}$ rather than its cover $\widetilde{\mathbb{F}}$. We now examine two important cases where this is possible.

**Corollary 4.3** (Sample complexity for unions of subspaces). *Suppose that $\mathbb{F} - \mathbb{F} = \cup_{j=1}^{d} \mathbb{F}_j$ is a union of $d$ subspaces of dimension at most $n \in \mathbb{N}$. Then* (4.1) *is equivalent to*

$$m_c \geq c_{\epsilon/2} \cdot \alpha^{-1} \cdot \operatorname*{ess\,sup}_{\theta \sim \rho_c} \{K_c(\mathbb{F} - \mathbb{F})(\theta)/\nu_c(\theta)\} \cdot \log(2nd/\delta), \quad c = 1, \ldots, C.$$

**Corollary 4.4** (Sample complexity for classical coverings). *Consider Theorem 4.2 with $n = 0$. Then* (4.1) *is implied by the condition*

$$m_c \geq c_{\epsilon/2} \cdot \alpha^{-1} \cdot \operatorname*{ess\,sup}_{\theta \sim \rho_c} \{K_c(\mathbb{F} - \mathbb{F})(\theta)/\nu_c(\theta)\} \cdot \log(2d/\delta), \quad c = 1, \ldots, C.$$

**Remark 4.5** Theorem 4.2 is formulated generally in terms of subspace coverings. This is done so that the scenarios covered in Corollaries 4.3 and 4.4 are both straightforward consequences. Our main examples in Section 3 are based on the (unions of) subspaces case. While this assumption is relevant for these and many other examples, there are some key problems that do not satisfy it. For example, in low-rank matrix (or tensor) recovery [27, 38, 108, 117], the space $\mathbb{F}$ of rank-$r$ matrices (or tensors) is not a finite union of finite-dimensional subspaces. But tight bounds for its covering number are known [26, 106], meaning it fits within the setup of Corollary 4.4.

## 4.2 Christoffel sampling

Theorem 4.2 reduces the question of identifying optimal sampling measures to that of finding weight functions $\nu_c$ that minimize the corresponding essential supremum in (4.1). The following lemma shows that this is minimized by choosing $\nu_c$ proportional to the generalized Christoffel function.

**Lemma 4.6** (Optimal sampling measure). *Let $L$, $\mathbb{F}$, $K(\mathbb{F})$ and $\kappa(\mathbb{F}) < \infty$ be as in Definition 2.6. Suppose that for almost every $\theta \in D$ there exists a $f \in \mathbb{F}$ such that $L(\theta)(f) \neq 0$. Then*

$$\operatorname*{ess\,sup}_{\theta \sim \rho}\{K(\mathbb{F})(\theta)/\nu(\theta)\} \geq \kappa(\mathbb{F})$$

*for any measurable $\nu : D \to \mathbb{R}$ that is positive almost anywhere and satisfies $\int_D \nu(\theta)\,d\rho(\theta) = 1$. Moreover, this optimal value is attained by the function $\nu^\star(\theta) = K(\mathbb{F})(\theta)/\kappa(\mathbb{F})$, $\theta \in D$.*

In view of this lemma, to optimize the sample complexity bound (4.1) we choose sampling measures

$$d\mu_c^\star(\theta) = K_c(\widetilde{\mathbb{F}})(\theta)\,d\rho_c(\theta)/\kappa_c(\mathbb{F}), \quad c = 1, \ldots, C. \tag{4.2}$$

As noted, we term this *Christoffel Sampling (CS)*. This leads to a sample complexity bound

$$m_c \geq c_{\epsilon/2} \cdot \alpha^{-1} \cdot \kappa_c(\widetilde{\mathbb{F}}) \cdot \log(2d\max\{n,1\}/\delta), \quad c = 1, \ldots, C. \tag{4.3}$$

In the case of subspaces (or union thereof with small $d$), this approach leads to a near-optimal sample complexity bound for the total number of measurements $m := m_1 + \cdots + m_C$.

**Corollary 4.7** (Near-optimal sampling for unions of subspaces). *Suppose that $\mathbb{F} - \mathbb{F} = \cup_{j=1}^{d} \mathbb{F}_j$ is a union of $d$ subspaces of dimension at most $n \in \mathbb{N}$. Then $\sum_{c=1}^{C} \kappa_c(\mathbb{F} - \mathbb{F}) \leq \beta dn$. Therefore, choosing the sampling measures as in* (4.2) *with $\widetilde{\mathbb{F}} = \mathbb{F} - \mathbb{F}$ and the number of samples $m_c \leq c_{\epsilon/2} \cdot \alpha^{-1} \cdot \kappa_c(\mathbb{F} - \mathbb{F}) \cdot \log(2nd/\delta)$ leads to the overall sample complexity bound*

$$m \leq c_{\epsilon/2} \cdot (\beta/\alpha) \cdot n \cdot d \cdot \log(2nd/\delta).$$

We note in passing that $\sum_{c=1}^{C} \kappa_c(\mathbb{F} - \mathbb{F}) \geq \alpha n/p$ whenever $\dim(\mathbb{Y}_c) \leq p \in \mathbb{N}$, $\forall c$. See Theorem E.4. Therefore, the sample complexity bound is at least $\mathcal{O}_{\epsilon,\alpha,p}(n\log(nd/\delta))$.

It is worth comparing these bounds with MCS. Suppose that the $\rho_c$'s are probability measures, in which case MCS is well defined. For MCS, we have $\nu_c \equiv 1$, $\forall c$, and therefore the corresponding measurement condition is

$$m_c \geq c_{\epsilon/2} \cdot \alpha^{-1} \cdot \lambda_c(\widetilde{\mathbb{F}}) \cdot \log(2d\max\{n,1\}/\delta), \quad c = 1, \ldots, C,$$

where $\lambda_c(\widetilde{\mathbb{F}}) = \operatorname*{ess\,sup}_{\theta \sim \rho_c} K_c(\widetilde{\mathbb{F}})(\theta) \geq \kappa_c(\widetilde{\mathbb{F}})$. Comparing with (4.3), we conclude that the benefit of CS over MCS in terms of sample complexity corresponds to the difference between the supremum of the Christoffel function (i.e., $\lambda_c(\widetilde{\mathbb{F}})$) and its average (i.e., $\kappa_c(\widetilde{\mathbb{F}})$). Thus, if $K_c$ has sharp peaks – as it does, for instance, in Fig. 2 – one expects a significant improvement from CS over MCS.

### 4.3 Generalization error bound and noisy data

Thus far, we have derived CS and shown parts (i) and (iii) of Theorem 2.7. In this section we establish the generalization bound in part (ii). For additional generality, we now consider noisy samples

$$y_{ic} = L_c(\Theta_{ic})(f^*) + e_{ic} \in \mathbb{Y}_c, \quad i = 1, \ldots, m_c \, c = 1, \ldots, C, \tag{4.4}$$

where the $e_{ic}$ represent measurement noise (see Remark E.6 for some further discussion on the noise term). We also consider inexact minimizers. Specifically, we say that $\hat{f} \in \mathbb{F}$ is a $\gamma$-*minimizer* of (2.2) for some $\gamma > 0$ if it yields a value of the objective function that is within $\gamma$ of the minimum value. For example, $\hat{f}$ may be the output of some training algorithm used to solve (2.2).

**Theorem 4.8.** *Let* $0 < \epsilon, \delta, \gamma < 1$ *and consider the setup of Section 2.1, except with noisy data* (4.4). *Suppose that* (4.1) *holds and also that* $\rho_c(D_c) < \infty$, $\forall c$. *Then, for any* $f^* \in \mathbb{X}_0$ *and* $\sigma \geq \|f^*\|_{\mathbb{X}}$, *the estimator* $\check{f} = \mathcal{T}_\sigma(\hat{f})$, *where* $\hat{f}$ *is a* $\gamma$-*minimizer of* (2.2), *satisfies*

$$\mathbb{E}\|f^* - \check{f}\|_{\mathbb{X}}^2 \leq 3 \left(1 + \frac{4\beta}{(1 - \epsilon)\alpha}\right) \inf_{f \in \mathbb{F}} \|f^* - f\|_{\mathbb{X}}^2 + 4\sigma^2 \delta + \frac{12}{(1 - \epsilon)\alpha} N + \frac{4}{(1 - \epsilon)\alpha} \gamma^2,$$

*where* $N = \sum_{c=1}^{C} \frac{\rho_c(D_c)}{m_c} \sum_{i=1}^{m_c} \|e_{ic}\|_{\mathbb{Y}_c}^2$.

## 5 Conclusions, limitations and future work

We conclude by noting several limitations and areas for future work. First, in this work, we have striven for breadth – i.e., highlighting the efficacy of CS4ML across a diverse range of examples – rather than depth. Our aim is to show that the well-known ideas for active learning in standard regression (e.g., leverage score sampling) can be extended to a very general setting. We do not claim that CS is the best possible method for each example, and as such, our experimental results only compare against (inactive) MCS. In each application considered, there are other domain-specific strategies that are known to outperform MCS. See [5, 6, 22, 55, 74, 104, 111] and references therein for Fourier imaging, and [10, 31, 52, 51, 86, 113] in the case of PINNs. CS is a general framework, and is arguably more theoretically grounded and less heuristic than some such methods. Nonetheless, further investigation is needed to ascertain which method is best in each setting. In a similar vein, while Figs. 1–3 show significant performance gains from our method, the extent of such gains depends heavily on the problem. In Appendices B.6 and D.6 we discuss cases where the gains are far more marginal. In the PINNs example in particular, additional studies are needed to see if CAS leads to benefits across a wider spectrum of PDEs. Moreover, there is the intriguing possibility of using CS as the starting point for more advanced active learning schemes – for example, by generalizing the linear-sample sparsification techniques of [32] or the 'boosting' techniques of [58].

Second, as noted previously, a limitation of our theoretical analysis is the log-linear scaling with respect to the number of subspaces $d$ (see Theorem 2.7, part (iii)). While this can be overcome in cases such as sparse regression (see Appendix A.4), we expect a more refined theoretical analysis may be able to tackle this problem in the general setting. Another limitation of our analysis is that the sample complexity bound in Theorem 4.2 involves $K_c$ evaluated over the subspace cover $\widetilde{\mathbb{F}}$, rather than simply $\mathbb{F} - \mathbb{F}$ itself. We anticipate this can also be improved through a more sophisticated argument. This would help close the theoretical gap in the generative models example considered in this paper. Another interesting theoretical direction involves reducing the sample complexity from log-linear to linear (when $\mathbb{F}$ is a linear subspace), by extending, for example, [32].

Finally, we reiterate that our framework and theoretical analysis are both very general. Consequently, there are many other potential problems to which we can apply this work. As noted, the main practical hurdle in applying this framework to other problems is to determine how to sample from the optimal measure (2.7), or some computationally tractable surrogate. Some other problems of interest include low-rank matrix or tensor recovery, as mentioned briefly in Remark 4.5, sparse regression using random feature models, as was recently developed in [64], active learning for single neuron models, as developed in [50], and operator learning [17, 73]. These are interesting avenues for future work.

## Acknowledgments and Disclosure of Funding

BA acknowledges the support of the Natural Sciences and Engineering Research Council of Canada of Canada (NSERC) through grant RGPIN-2021-611675. ND acknowledges the support of Florida State University through the CRC 2022-2023 FYAP grant program.

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

# A  Further literature and discussion

In this appendix, we provide some additional discussion that relates our work to existing literature.

## A.1  Classical Christoffel functions

Let $\mathbb{F}$ be a linear subspace of the Hilbert space $\mathbb{X} = L^2_\rho(D)$. The *Christoffel function* of $\mathbb{F}$ is the function $\theta \mapsto 1/K(\mathbb{F})(\theta)$, where

$$K(\mathbb{F})(\theta) = \sup\left\{ \frac{|f(\theta)|^2}{\|f\|^2_{\mathbb{X}}} : f \in \mathbb{F},\ v \neq 0 \right\}, \quad \theta \in D.$$

This is clearly a special case of Definition 2.6 with $\mathbb{X}_0 = C(\overline{D})$ and $L : D \to \mathcal{B}(\mathbb{X}_0, \mathbb{R})$ given by $L(\theta)(f) = f(\theta)$. The Christoffel function is a classical object in approximation theory, most typically associated with the case where $\mathbb{F}$ is a space of algebraic polynomials [97]. Note that $K(\mathbb{F})$ is precisely the diagonal of the reproducing kernel of $\mathbb{F}$ in $\mathbb{X}$. It also follows immediately from Lemma E.1 that

$$K(\mathbb{F})(\theta) = \sum_{i=1}^{n} |f_i(\theta)|^2,$$

for any orthonormal basis $\{f_i\}_{i=1}^n$ of $\mathbb{F}$.

## A.2  Relation to leverage scores and leverage score sampling

Let $D$ be a domain with a measure $\lambda$ (typically the Lebesgue measure), $\mathbb{F}$ be a class of functions $f : D \to \mathbb{C}$ and $\rho$ be a probability measure over $D$ with Radon-Nikodym derivative $p = \mathrm{d}\rho/\mathrm{d}\lambda$. Then the *leverage score* is defined as

$$\tau(\theta) = \tau(\mathbb{F}, p)(\theta) = \sup\left\{ \frac{|f(\theta)|^2 p(\theta)}{\|f\|^2_p} : f \in \mathbb{F}\backslash\{0\} \right\}, \tag{A.1}$$

where $\|f\|^2_p = \int_D |f(\theta)|^2 p(\theta)\,\mathrm{d}\lambda(\theta) = \int_D |f(\theta)|^2\,\mathrm{d}\rho(\theta)$. See, e.g., [46] and references therein. Like with CS, *leverage score sampling* is a type of importance sampling where one draws samples randomly according to the probability density proportional to $\tau$ (or, often in practice, some easy-to-compute upper bound $\widetilde{\tau}$). We remark in passing that leverage score sampling is sometimes known as *effective resistance* or *coherence motivated* sampling in the literature. Note that in the literature on leverage scores, the empirical nondegeneracy condition (2.5) is sometimes referred to as the *subspace embedding* condition (see, e.g., [119]).

It is readily seen that (A.1) a special case of Definition 2.6. Let $\mathbb{X} = L^2_\rho(D)$. Then there are several different ways to formulate this.

   (i) Consider $D$ equipped with the measure $\lambda$ and set $L(\theta)(f) = \sqrt{p(\theta)}f(\theta)$. Then nondegeneracy (2.3) holds with $\alpha = \beta = 1$ and we have

$$K(\mathbb{F})(\theta) = \sup\left\{ \frac{|f(\theta)|^2 p(\theta)}{\|f\|^2_{\mathbb{X}}} : f \in \mathbb{F},\ f \neq 0 \right\} =: K_a(\theta).$$

   (ii) Alternatively, consider $D$ equipped with the measure $\rho$ and set $L(\theta)(f) = f(\theta)$. Then nondegeneracy (2.3) holds with $\alpha = \beta = 1$ and we have

$$K(\mathbb{F})(\theta) = \sup\left\{ \frac{|f(\theta)|^2}{\|f\|^2_{\mathbb{X}}} : f \in \mathbb{F},\ f \neq 0 \right\} =: K_b(\theta).$$

Notice that $K_a(\theta) = \tau(\mathbb{F}, p)(\theta)$ is precisely the leverage score (A.1). Conversely, $K_b(\theta) = \tau(\mathbb{F}, p)(\theta)/p(\theta)$ differs from the leverage score (A.1) by the factor $p(\theta)$. Whether to include $p(\theta)$ in the definition is one of convention. What is important is that both (i) and (ii) lead to exactly the same Christoffel sampling sampling measure $\mu^\star$ in (2.11), which is equivalent to leverage score sampling for the standard regression problem. Indeed, this measure is

$$\mathrm{d}\mu^\star(\theta) \propto K_a(\theta)\,\mathrm{d}\lambda(\theta) = \tau(\mathbb{F}, p)(\theta)\,\mathrm{d}\lambda(\theta)$$

for case (i) or

$$\mathrm{d}\mu^{\star}(\theta) \propto K_b(\theta)\,\mathrm{d}\rho(\theta) = \frac{\tau(\mathbb{F}, p)(\theta)}{p(\theta)} p(\theta)\,\mathrm{d}\lambda(\theta) = \tau(\mathbb{F}, p)(\theta)\,\mathrm{d}\lambda(\theta)$$

for case (ii).

(Statistical) leverage scores are an old concept [29] that have recently found many applications in machine learning, including randomized numerical linear algebra [119], kernel methods [8, 12, 46, 47, 94], active learning [12, 30, 41, 46, 85], data analysis [77] and beyond. Leverage scores are perhaps most commonly encountered in the discrete setting. Here the *matrix leverage score* of a matrix a matrix $A \in \mathbb{R}^{N \times n}$ is defined as

$$\tau(i) = \tau(A)(i) = \max_{\substack{x \in \mathbb{R}^d \\ Ax \neq 0}} \frac{(Ax)_i^2}{\|Ax\|_2^2} = a_i^\top (A^\top A)^{-1} a_i, \quad i = 1, \dots, N, \tag{A.2}$$

where $a_i$ is the $i$th row of $A$. This is also a special case of Definition 2.6. Indeed, let $\mathbb{X} = \mathbb{R}^N$ and $\mathbb{Y} = \mathbb{R}$, both equipped with the Euclidean inner product, $D = \{1, \dots, N\}$ be equipped with the uniform measure $\rho$, $L(i)(x) = x_i$ for a vector $x = (x_i)_{i=1}^N \in \mathbb{R}^N$ and

$$\mathbb{F} = \{Ax : x \in \mathbb{R}^n\} \subseteq \mathbb{R}^N.$$

Then (2.3) holds with $\alpha = \beta = 1$ (notice that $\rho$ is the uniform measure, not the uniform probability measure) and the generalized Christoffel function satisfies

$$K(\mathbb{F})(i) = \sup\left\{ \frac{\|L(f)(i)\|_{\mathbb{Y}}^2}{\|f\|_{\mathbb{X}}^2} : f \in \mathbb{F},\ f \neq 0 \right\} = \sup\left\{ \frac{(Ax)_i^2}{\|Ax\|_2^2} : x \in \mathbb{R}^n,\ Ax \neq 0 \right\} = \tau(A)(i).$$

There are a number of generalizations of leverage scores in the literature. See, e.g., [101]. However, to the best of our knowledge, none of these are similar to the setting considered in this paper (in particular, Definition 2.6).

## A.3   Other related work

Our general framework and results are related to recent work [43]. Here the authors consider approximating functions in arbitrary (linear or nonlinear) approximation spaces (model classes) via an empirical least-squares regression based on a semi-norm that depends on a (random) variable $y \in Y \subseteq \mathbb{R}^d$. In a similar spirit to Theorem 4.2 and Theorem 4.8, [43] establishes sample complexity and generalization error bounds, and then use the former to optimize the sampling measure. Specifically, [43, Thm. 2.8 and Cor. 2.9] is similar to Theorem 4.2 with $d = 0$ (i.e., the case of classical coverings as opposed to subspace coverings) and [43, Thm. 2.12] is similar to Theorem 4.8, albeit without the additional technical steps used in the proof of Theorem 4.8 to obtain an expected error bound involving only the $\mathbb{X}$-norm on the left- and right-hand sides.

This framework is related to our framework with $C = 1$ sampling operators and with $\mathbb{X}$ being a Banach space of functions. In fact, it is somewhat more general in this case: (i) no inner product structure is assumed and, (ii) each measurement (which are implicit in the semi-norm in [43]) could belong to a different space. Note that one could readily incorporate (ii) into our framework, by making the target space $\mathbb{Y}_c$ depend on $\theta_c$. However, (i) leads to fundamentally weaker sample complexity guarantees. For example, in the case of a linear subspace $\mathbb{F}$, the bounds in [43] scale log-cubically in $n = \dim(\mathbb{F})$, i.e., like $c \cdot n^3 \cdot \log(n)$ (see [43, Sec. 3.1]), whereas the results in this work are log-linear, i.e., $c \cdot n \cdot \log(n)$ (see Corollary 4.7), and therefore near-optimal. A similar suboptimal log-cubic scaling also arises in the case where $\mathbb{F}$ is a nonlinear space of $s$-sparse expansions in an orthonormal basis (see [43, Sec. 3.2]). See Appendix A.4 for more on this example.

As noted, the root cause of this suboptimal scaling is (i). In our framework, we impose that $\mathbb{X}$ is a Hilbert space and each $\mathbb{Y}_c$ is a semi-inner product space. This allows us to use matrix concentration techniques (specifically, the matrix Chernoff bound [115]) in combination with the technique of subspace coverings (Definition 4.1) to obtain better sample complexity bounds. We remark in passing that the special case of $d = 0$ in our main sample complexity bound Theorem 4.2 (which, as noted above, is similar to [43]) does not require an inner product structure, i.e., it holds if $\mathbb{X}$ is a Banach space and each $\mathbb{Y}_c$ is a semi-normed vector space. However, if applied to the case of linear subspaces, it would also lead to suboptimal sample complexity estimates much as in [43, Sec. 3.1].

## A.4 The case of sparse regression

In this section, we briefly discuss the application of our general framework to sparse regression. For convenience, we shall assume that $\mathbb{X} = L_\rho^2(D)$ is the space of weighted square-integrable functions $f : D \to \mathbb{C}$, $\mathbb{X} = C(\overline{D})$, $C = 1$, $\mathbb{Y} = \mathbb{Y}_1 = \mathbb{C}$ with the Euclidean inner product and $L = L_1$ is the pointwise evaluation operator $L(\theta)(f) = f(\theta)$. However, the following discussion can be readily generalized to general setting considered in Section 2.1.

Let $\Phi = \{\phi_i\}_{i=1}^N \subset \mathbb{X}$ be a collection of elements in $\mathbb{X}$, where $N \in \mathbb{N}$. We assume that this is a *Riesz system*, i.e., there exist constants $0 < a \le b < \infty$ such that

$$a \sum_{i=1}^N |c_i|^2 \le \left\| \sum_{i=1}^N c_i \phi_i \right\|_{L_\rho^2(D)}^2 \le b \sum_{i=1}^N |c_i|^2, \quad \forall c_i \in \mathbb{C}.$$

Note that $a = b = 1$ if and only if $\Phi$ is an orthonormal system. Now let $1 \le s \le N$ and define

$$\mathbb{F} = \mathbb{F}_s = \left\{ \sum_{i \in S} c_i \phi_i : c_i \in \mathbb{C}, \forall i, \ S \subseteq \{1, \ldots, N\}, \ |S| = s \right\}.$$

Then $\mathbb{F}$ is the set of all $s$-sparse expansions in the system $\Phi$. In this case, we refer to (2.2) as a *sparse regression* problem. Notice that empirical nondegeneracy Eq. (2.5) is in this case equivalent to the *Restricted Isometry Property (RIP)* [48] for the matrix $A = \frac{1}{\sqrt{m}} (\phi_j(\Theta_i))_{i,j=1}^{m,N} \in \mathbb{C}^{m \times N}$.

Several remarks are in order. First, notice that $\mathbb{F}_s - \mathbb{F}_s = \mathbb{F}_{2s}$. Second, $\mathbb{F}_s$ is union-of-subspaces model. Indeed, $\mathbb{F}_s$ is precisely a union of $d = \binom{N}{s}$ subspaces, each corresponding to a particular choice of support set $S \subseteq \{1, \ldots, N\}$, $|S| = s$. When $N$ and $s$ are large, this is a large number, which makes the bound in Corollary 4.7 meaningless.

However, in this case, we can obtain a more appealing bound. Notice that the generalized Christoffel function

$$K(\mathbb{F}_s)(\theta) = \max_{\substack{S \subseteq \{1, \ldots, N\} \\ |S| \le s}} \left\{ \frac{|f(\theta)|^2}{\|f\|_{L_\rho^2(D)}^2} : f = \sum_{i \in S} c_i \phi_i, \ c_i \in \mathbb{C}, \forall i \right\}.$$

It is a simple argument to see that

$$|f(\theta)|^2 \le s \cdot a^{-1} \cdot \max_{i=1,\ldots,N} |\phi_i(\theta)|^2 \cdot \|f\|_{L_\rho^2(D)}^2.$$

Therefore,

$$K(\mathbb{F}_s)(\theta) \le s \cdot a^{-1} \cdot \widetilde{K}(\theta), \qquad \widetilde{K}(\theta) := \max_{i=1,\ldots,N} |\phi_i(\theta)|^2.$$

The function $\widetilde{K}_c$ serves two purposes. First, it can be used to upper bound $\kappa(\mathbb{F}_s) = \int_D K(\mathbb{F}_s)(\theta) \, d\rho(\theta)$, i.e., the term appearing in the estimate in Corollary 4.7. Second, it can also be used as a convenient surrogate from which to sample, as computing $\widetilde{K}(\theta)$ is typically easier than computing the true Christoffel function $K(\mathbb{F}_s)(\theta)$.

Suppose now that $\|\phi_i\|_{L_\rho^\infty(D)} = \mathrm{ess\,sup}_{\theta \sim D} |\phi_i(\theta)|^2 \le K$ for all $i = 1, \ldots, N$. In this case, $\Phi$ is known as a *bounded* Riesz system [23]. Then

$$K(\mathbb{F}_s)(\theta) \le s \cdot a^{-1} \cdot \widetilde{K}(\theta) \le s \cdot K \cdot a^{-1}.$$

Using this and the well-known estimate $d = \binom{N}{s} \le (eN/s)^s$, we deduce from Corollary 4.3 the sample complexity bound

$$m \gtrsim c_{\epsilon/2} \cdot a^{-1} \cdot s \cdot (s \cdot \log(eN/s) + \log(s/\delta)).$$

This is a substantial improvement on the bound implied directly by Corollary 4.7, which depends linearly on $d = \binom{N}{s}$. It also improves the bound of [43, Sec. 3.2] from log-cubic to log-quadratic (recall the discussion in Appendix A.3). However, the reader will note that it is not optimal, since $m$ scales log-quadratically in $s$. One can improve this bound to log-linear by using a more sophisticated argument specific to the sparse regression case. See, e.g., [4, 23, 48] and references therein. The

extent to which this argument can be extended to the most general setting of our framework is a problem for future work.

Unfortunately, many Riesz systems of practical interest are not bounded. In this case, one has the sample complexity bound

$$m \gtrsim c_{\epsilon/2} \cdot \widetilde{\kappa} \cdot a^{-1} \cdot s \cdot (s \cdot \log(eN/s) + \log(s/\delta)), \tag{A.3}$$

where

$$\widetilde{\kappa} = \int_D \max_{i=1,\dots,N} |\phi_i(\theta)|^2 \, \mathrm{d}\rho(\theta).$$

The bound (A.3) grows log-quadratically in $s$, as desired (it can also be improved to log-linear as mentioned above). However, the constant $\widetilde{\kappa}$ may depend on $N$. We refer to [4] for an extensive discussion on this issue. It is notable that $\widetilde{\kappa}$ can be estimated numerically. In cases of interest such as sparse multivariate polynomial regression, it is either bounded or mildly growing in $N$ [4].

## B  Polynomial regression with gradient-augmented data

In this appendix, we describe the example considered in Section 3.1. Let $f^* : \mathbb{R}^d \to \mathbb{R}$. In the gradient-augmented learning problem, we consider training data of the form

$$\left\{ \left( \theta_i, (\partial_k f^*(\theta_i))_{k=0}^d \right) \right\}_{i=1}^m,$$

where (for convenience) $\partial_0 f^* = f^*$ and $\partial_k f^* = \partial f^* / \partial \theta_k$ otherwise.

### B.1  Formulation in terms of the general sampling framework

We now describe how to formulate the gradient-augmented learning problem as a special case of the general framework. Let $C = 1$, $D = D_1 = (\mathbb{R}^d, \rho)$ where $d \geq 1$ and $\rho$ be the tensor-product Gaussian measure on $\mathbb{R}^d$, i.e. $\mathrm{d}\rho = (2\pi)^{-d/2} \mathrm{e}^{-\frac{1}{2}\|\theta\|_2^2} \mathrm{d}\theta$. Then we let

$$\mathbb{X} = H_\rho^1(\mathbb{R}^d) = \left\{ \partial_k f \in L_\rho^2(\mathbb{R}^d) \right\}$$

be the Sobolev space of weighted square-integrable functions with respect to $\rho$ with weak first-order derivatives that are also square-integrable with respect to $\rho$. This is a Hilbert space with inner product

$$\langle f, g \rangle_{\mathbb{X}} = \sum_{k=0}^d \int_{\mathbb{R}^d} \partial_k f(\theta) \partial_k g(\theta) \, \mathrm{d}\rho(\theta).$$

Next, we let $\mathbb{X}_0 = C^1(\overline{D})$ and $\mathbb{Y}$ be the Hilbert space

$$\mathbb{Y} = (\mathbb{C}^{d+1}, \langle \cdot, \cdot \rangle_2),$$

where $\langle \cdot, \cdot \rangle_2$ is the Euclidean inner product. Then, we define the sampling operator $L : D \to \mathcal{B}(\mathbb{X}_0, \mathbb{Y})$ as

$$L(\theta)(f) = (\partial_k f(\theta))_{k=0}^d, \tag{B.1}$$

Notice that

$$\int_D \|L(\theta)(f)\|_{\mathbb{Y}}^2 \, \mathrm{d}\rho(\theta) = \sum_{k=0}^d \int_{\mathbb{R}^d} |\partial_k f(\theta)|^2 \, \mathrm{d}\rho(\theta) = \|f\|_{L_\rho^2(D)}^2 + \sum_{k=1}^d \|\partial_k f\|_{L_\rho^2(D)}^2 = \|f\|_{H_\rho^1(\mathbb{R}^d)}^2.$$

Thus, Assumption 2.2 holds for this example with $\alpha = \beta = 1$.

### B.2  The approximation space: multivariate Hermite polynomials

The standard Hermite polynomials $H_k(x)$ are eigenfunctions of the Sturm–Liouville problem

$$(wu')' + 2kwu = 0,$$

where $w(x) = \mathrm{e}^{-x^2}$. They are orthogonal on $\mathbb{R}$ with respect to this function, specifically,

$$\int_{-\infty}^{\infty} H_n(x) H_m(x) w(x) \, \mathrm{d}x = \sqrt{\pi} 2^n n! \epsilon_{n,m}.$$

They satisfy the three term recurrence with $H_0(x) = 1$, $H_1(x) = 2x$ and

$$H_{n+1}(x) = 2xH_n(x) - 2nH_{n-1}(x), \quad n = 1, 2, \ldots, \tag{B.2}$$

and their derivatives satisfy the relation $H_0' = 0$ and

$$H_n' = 2nH_{n-1}, \quad n = 1, 2, \ldots \tag{B.3}$$

See, e.g., [28, Sec. 2.6.2] for standard properties of Hermite polynomials.

It is more convenient for our purposes to consider the standard Gaussian (probability) measure $\mathbb{R}$, i.e.,

$$d\rho(\theta) = \frac{1}{\sqrt{2\pi}} e^{-\frac{1}{2}\theta^2} d\theta.$$

Then the orthonormal Hermite polynomials (sometimes known as the "probabilists' version") are given by

$$\psi_n(\theta) = \frac{1}{\sqrt{2^n n!}} H_n(\theta/\sqrt{2}).$$

Manipulating (B.2), we see that the orthonormal Hermite polynomials satisfy $\psi_0(\theta) = 1$, $\psi_1(\theta) = y$ and

$$\psi_{n+1}(\theta) = \frac{\theta}{\sqrt{n+1}} \psi_n(\theta) - \sqrt{\frac{n}{n+1}} \psi_{n-1}(\theta), \quad n = 1, 2, \ldots \tag{B.4}$$

Also, we have $\psi_n' = \frac{1}{\sqrt{2^n n!}} \frac{1}{\sqrt{2}} H_n'(\theta/\sqrt{2}) = \frac{1}{\sqrt{2^n n!}\sqrt{2}} 2n H_{n-1}(\theta/\sqrt{2})$, and therefore $\psi_0' = 0$ and

$$\psi_n'(\theta) = \sqrt{n}\psi_{n-1}(\theta), \quad n = 1, 2, \ldots. \tag{B.5}$$

Using this, we readily deduce that the functions

$$\phi_n(\theta) = \frac{\psi_n(\theta)}{\sqrt{1+n}}, \quad n = 0, 1, \ldots,$$

form an orthonormal basis of the space

$$H_\rho^1(\mathbb{R}) = \{f \in L_\rho^2(\mathbb{R}) : f' \in L_\rho^2(\mathbb{R})\},$$

i.e., the Sobolev space of square-integrable functions $f : \mathbb{R} \to \mathbb{R}$ (with respect to $\rho$) with square-integrable (weak) first derivatives.

We extend to $d$ dimensions via tensorization. We let

$$d\rho(\theta) = \frac{1}{\sqrt{2\pi}} e^{-\frac{1}{2}\|\theta\|_2^2} d\theta$$

and consider the tensor Hermite polynomials

$$\Psi_\nu(\theta) = \psi_{\nu_1}(\theta_1) \cdots \psi_{\nu_d}(\theta_d), \qquad \nu = (\nu_i)_{i=1}^d \in \mathbb{N}_0^d, \ \theta = (\theta_i)_{i=1}^d \in \mathbb{R}^d.$$

Leveraging what we did in the one-dimensional case, we can then construct an orthonormal basis of the Sobolev space

$$H_\rho^1(\mathbb{R}^d) = \{f \in L_\rho^2(\mathbb{R}^d) : \partial_k f \in L_\rho^2(\mathbb{R}^d), k = 1, \ldots, d\}$$

as $\{\Phi_\nu\}_{\nu \in \mathbb{N}_0^d}$, where

$$\Phi_\nu = \frac{\Psi_\nu}{\sqrt{1 + \nu_1 + \ldots + \nu_d}}, \quad \nu = (\nu_i)_{i=1}^d \in \mathbb{N}_0^d.$$

We conclude by defining the approximation space $\mathbb{F}$. Fix an index $p \in \mathbb{N}_0$ and let

$$S = S_p = \left\{ \nu = (\nu_i)_{i=1}^d \in \mathbb{N}_0^d : \prod_{i=1}^d (\nu_i + 1) \leq p + 1 \right\} \subset \mathbb{N}_0^d. \tag{B.6}$$

This is the so-called *hyperbolic cross* index set, which is particularly well suited for multivariate polynomial approximation [1, 37]. Using this, we define the linear approximation space

$$\mathbb{F} = \text{span}\{\Phi_\nu : \nu \in S\} \subset H_\rho^1(\mathbb{R}^d).$$

Notice that Assumption 2.5 holds for this choice of $\mathbb{F}$. Indeed, $\mathbb{F}$ contains the constant function $f(\theta) = 1$, $\forall \theta \in \mathbb{R}^d$, which satisfies $L(\theta)(f) = (1, 0, \ldots, 0)^\top \neq 0$.

## B.3 Numerical solution of the regression problem

We now describe the numerical solution of the regression problem (2.2) in this case. Let $\theta_1, \ldots, \theta_m \in \mathbb{R}^d$ be sample points and $\nu : D \to \mathbb{R}$ be measurable and finite and positive almost everywhere. Then this problem takes the form

$$\hat{f} \in \underset{f \in \mathbb{F}}{\operatorname{argmin}} \frac{1}{m} \sum_{i=1}^{m} \frac{1}{\nu(\theta_i)} \sum_{k=0}^{d} |y_{ik} - \partial_k f(\theta_i)|^2, \tag{B.7}$$

where $y_{ik} = \partial_k f^*(\theta_i) + e_{ik}$ is the $k$th component of the $i$th measurement of $f^*$. For convenience, we assume the ordering $S = \{\nu^1, \ldots, \nu^n\}$. Let $f \in \mathbb{F}$ be arbitrary and write $f = \sum_{i=1}^n c_i \Phi_{\nu^i}$ and $c = (c_i)_{i=1}^n$. Then

$$\frac{1}{\sqrt{\nu(\theta_i)m}} \partial_k f(\theta_i) = (A_k c)_i\,,$$

where

$$A_k = \left( \frac{1}{\sqrt{\nu(\theta_i)m}} \partial_k \Phi_{\nu^j}(\theta_i) \right)_{i,j=1}^{m,n}.$$

Also, let

$$b_k = \left( \frac{1}{\sqrt{\nu(\theta_i)m}} y_{ik} \right)_{i=1}^{m}$$

and set

$$A = \begin{bmatrix} A_0 \\ \vdots \\ A_d \end{bmatrix}, \quad b = \begin{bmatrix} b_0 \\ \vdots \\ b_d \end{bmatrix}.$$

Then

$$\frac{1}{m} \sum_{i=1}^{m} \frac{1}{\nu(\theta_i)} \sum_{k=0}^{d} |y_{ik} - \partial_k f(\theta_i)|^2 = \|Ac - b\|_2^2.$$

Thus, the algebraic least-squares problem

$$\hat{c} \in \underset{c \in \mathbb{C}^n}{\operatorname{argmin}} \|Ac - b\|_2^2, \tag{B.8}$$

is equivalent to (B.7), in the sense that every solution of this problem yields a solution $\hat{f} = \sum_{i=1}^n \hat{c}_i \Phi_{\nu^i}$ of (B.7) and vice versa.

Notice (B.8) is an $m(d+1) \times n$ algebraic least-squares problem, which can be easily solved using standard linear algebra software once the matrix $A$ has been computed. This is done using the relations (B.4) and (B.5).

## B.4 Christoffel sampling

We now explain how to perform CS in this example. Directly sampling from the measure (2.7) is potentially difficult for this problem, since a simple characterization of the generalized Christoffel function $K(\mathbb{F})(\theta)$ is lacking. However, we observe that

$$\frac{1}{d+1} \widetilde{K}(\mathbb{F})(\theta) \leq K(\mathbb{F})(\theta) \leq \widetilde{K}(\mathbb{F})(\theta), \tag{B.9}$$

where

$$\widetilde{K}(\mathbb{F})(\theta) = \sum_{\nu \in S} \|L(\theta)(\Phi_\nu)\|_{\mathbb{Y}}^2 = \sum_{\nu \in S} \sum_{k=0}^{d} |\partial_k \Phi_{\nu^i}(\theta)|^2 = \sum_{\nu \in S} \sum_{k=0}^{d} \frac{|\psi'_{\nu_k}(\theta_k)|^2 \prod_{\substack{l=1 \\ l \neq k}}^{d} |\psi_{\nu_l}(\theta_l)|^2}{1 + \|\nu\|_1}. \tag{B.10}$$

See Lemma E.1. Notice that

$$\int_D \widetilde{K}(\mathbb{F})(\theta) \, d\rho(\theta) = \sum_{\nu \in S} \int_{\mathbb{R}^d} \sum_{k=0}^{d} |\partial_k \Phi_\nu(\theta)|^2 \, d\rho(\theta) = \sum_{\nu \in S} \|\Phi_\nu\|_{H^1_\rho(\mathbb{R}^d)}^2 = n,$$

since the functions $\Phi_\nu$ are orthonormal. Therefore, we can use $\widetilde{K}(\mathbb{F})(\theta)$ to construct a sampling measure as

$$\mathrm{d}\tilde{\mu}(\theta) = \tilde{\nu}(\theta)\,\mathrm{d}\rho(\theta) = \frac{\widetilde{K}(\mathbb{F})(\theta)}{n}\,\mathrm{d}\rho(\theta).$$

Since $\mathrm{ess\,sup}_{\theta\sim\rho}\{K(\mathbb{F})(\theta)/\tilde{\nu}(\theta)\} \leq n$, Corollary 4.3 implies that sampling from $\tilde{\mu}$ yields optimal, log-linear sample complexity. As it is so closely related to the generalized Christoffel function (see (B.9)), we continue to refer to this as Christoffel sampling.

One can sample from $\tilde{\mu}$ by observing that it is an additive mixture [34] of tensor-product probability measures, each of which can be sampled from efficiently [96]. However, we opt for a simpler approach based on [2, 92] involving finite grids and discrete measures. Let $Z = \{z_i\}_{i=1}^N \subset D$ grid of $N \in \mathbb{N}$ points, with each point $z_i$ drawn i.i.d. from the measure $\rho$. Then, we now replace the measure $\rho$ by the finite measure

$$\bar{\rho} = \frac{1}{N}\sum_{i=1}^N \delta_{z_i}. \tag{B.11}$$

After doing this, the Christoffel function becomes a discrete function supported on the grid $Z$, which can be expressed as

$$\widetilde{K}(\mathbb{F})(z) = \sum_{j=1}^n \sum_{k=0}^d |\partial_k\chi_j(z)|^2, \quad z \in Z,$$

where $\{\chi_j\}_{j=1}^n$ is an orthonormal basis of $\mathbb{F}$ with respect to the discrete $H_{\bar{\rho}}^1(\mathbb{R})$-inner product. Now define the matrix

$$B = \begin{bmatrix} B_0 \\ \vdots \\ B_d \end{bmatrix}, \qquad B_k = \left(\frac{1}{\sqrt{N}}\partial_k\Phi_{\nu^j}(z_i)\right)_{i,j=1}^{N,n}.$$

Consider the QR factorization $B = QR$, where $Q = (q_{ij}) \in \mathbb{C}^{(d+1)N\times n}$ has orthonormal columns and $R \in \mathbb{C}^{n\times n}$ is upper triangular. Then the $j$th column of $Q$ contains precisely the values $\partial_k\chi_j(z_i)$, $i = 1,\ldots,N$, $k = 0,\ldots,d$. Using this, we get

$$\widetilde{K}(\mathbb{F})(z_i) = N\sum_{j=1}^n \sum_{k=0}^d |q_{i+kN,j}|^2, \quad i = 1,\ldots,N.$$

Notice that

$$\int \widetilde{K}(\mathbb{F})(z)\,\mathrm{d}\bar{\rho}(z) = \sum_{i=1}^N \sum_{j=1}^n \sum_{k=0}^d |q_{i+kN,j}|^2 = n,$$

and therefore

$$\nu(z) = \frac{\widetilde{K}(\mathbb{F})(z_i)}{n}.$$

Hence, we can compute the function $\widetilde{K}(\mathbb{F})$ over the grid $Z$ and then use this to sample from the optimal measure $\mu$, which in this case is the discrete measure with $\theta \sim \mu$ if

$$\mathbb{P}(\theta = z_i) = \frac{\widetilde{K}(\mathbb{F})(z_i)}{nN}, \quad i = 1,\ldots,N.$$

## B.5   Additional experimental details

As described in Appendix B.4 we first draw a random grid from the measure $\rho$. In our experiments, we choose the number of grid points as $N = 50,000$. Note that this grid is drawn once, prior to any subsequent computations. In Fig. 1 we compute a test error as the relative Sobolev norm error over this grid. This is done using the following expression:

$$E(f^*) = \frac{\|f^* - \hat{f}\|_{H_{\bar{\rho}}^1(\mathbb{R}^d)}}{\|f^*\|_{H_{\bar{\rho}}^1(\mathbb{R}^d)}} = \sqrt{\frac{\sum_{k=0}^d \frac{1}{N}\sum_{i=1}^N |\partial_k(f^* - \hat{f})(z_i)|^2}{\sum_{k=0}^d \frac{1}{N}\sum_{i=1}^N |\partial_k f^*(z_i)|^2}}. \tag{B.12}$$

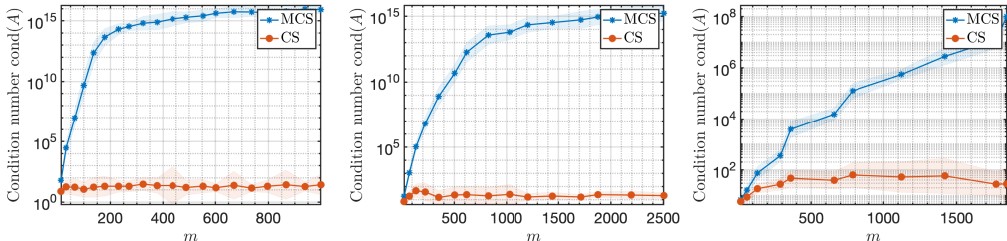

Figure 4: **CS for gradient-augmented polynomial regression.** Plots of the average condition number $\mathrm{cond}(A)$ versus $m$ for MCS and CS for gradient-augmented polynomial regression for $d = 2, 4, 8$ (left to right). In both cases, we use the scaling $m = \lceil \max\{n, n\log(n)\}/(d+1) \rceil$.

The experiments shown in Fig. 1 proceed as follows. First, we generate a sequence of hyperbolic cross index sets (B.6) of orders $1 \leq p_1 < p_2 < \cdots < p_L$. Second, for each $l = 1, \ldots, L$, we choose $m = m_l$ according to the scaling

$$m = \lceil \max\{n, n\log(n)\}/(d+1) \rceil, \tag{B.13}$$

where $n = n_l = |S_{p_l}|$. Then, we draw $m$ samples randomly using either MCS or CS (as described above), compute the approximation $\hat{f}$ using (B.8) and then compute the error using (B.12). We then repeat this process for a total of $T = 25$ trials.

The graphs in Fig. 1 and this appendix show the geometric mean (the main curve) and plus/minus one (geometric) standard deviation (the shaded region). The reason for using the geometric mean is because the errors are plotted in log-scale on the $y$-axis. See [1, Sec. A.1] for further rationale behind this choice of visualization.

These experiments were conducted on a 2012 MacBook Pro with a 2.7 GHz Intel Core i7 CPU and 16 GB of DDR3-1600 RAM running MATLAB R2020b. Total runtime for these experiments was roughly three days.

### B.6 Further experiments and discussion

Let $0 \leq \alpha' \leq \beta' < \infty$ be the optimal constants in the empirical nondegeneracy condition (2.5). In this example, for a collection of sample points $\theta_1, \ldots, \theta_m$, these are given by

$$\alpha' = \inf\left\{ \frac{1}{m}\sum_{k=0}^{d}\sum_{i=1}^{m} \frac{1}{\nu(\theta_i)}|\partial_k f(\theta_i)|^2 : f \in \mathbb{F}, \ \|f\|_{\mathbb{X}} \leq 1 \right\}$$

$$\beta' = \sup\left\{ \frac{1}{m}\sum_{k=0}^{d}\sum_{i=1}^{m} \frac{1}{\nu(\theta_i)}|\partial_k f(\theta_i)|^2 : f \in \mathbb{F}, \ \|f\|_{\mathbb{X}} \leq 1 \right\}$$

(here we recall that $\mathbb{F} - \mathbb{F} = \mathbb{F}$, since $\mathbb{F}$ is a subspace). By expanding $f \in \mathbb{F}$ in the orthonormal basis $\{\Phi_\nu : \nu \in S\}$ we readily deduce that

$$\alpha' = \sigma_{\min}(A), \qquad \beta' = \sigma_{\max}(A).$$

Thus, these constants can be computed numerically. Moreover, their ratio $\beta'/\alpha' = \mathrm{cond}(A)$ is precisely the 2-norm condition number of $A$. In Fig. 4, we plot $\mathrm{cond}(A)$ for both schemes using the same experimental setup as in Fig. 1. As predicted by our theory, CS leads to a bounded condition number for all $m$, that is at most roughly $10^2$ in magnitude. However, MCS leads to an exploding condition number. This indicates that the scaling used Eq. (B.13) is not sufficient to ensure a good generalization bound, which is the reason for the poor performance of MCS in Fig. 1.

To examine this further, in Fig. 5 we perform the following experiment. For each value of $n = n_l$, $l = 1, \ldots, L$, we compute the minimum value of $m = m_l$ such that $\mathrm{cond}(A) \leq \mathrm{tol}$, where $\mathrm{tol} = 10$. This is done as follows. We first set $m = n/(d+1)$ and compute $\mathrm{cond}(A)$. If $\mathrm{cond}(A) > \mathrm{tol}$, we increment $m$ by one, draw new samples, re-compute $\mathrm{cond}(A)$ and check whether $\mathrm{cond}(A) \leq \mathrm{tol}$. If not, we repeat this step. Finally, we repeat the whole process for a total of $T = 25$ trials.

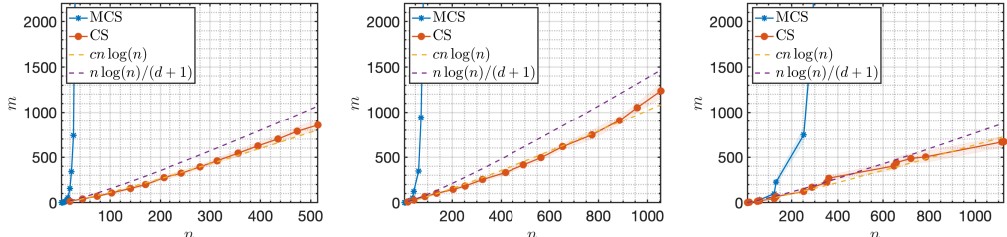

Figure 5: **CS for gradient-augmented polynomial regression.** Plots of minimum value of $m$ that which yields a condition number $\mathrm{cond}(A) \leq \mathsf{tol}$ for both MCS and CS, where $\mathsf{tol} = 10$ and $d = 2, 4, 8$ (left to right).

In Fig. 5, we show the results of this computation for MCS and CS. We plot both the arithmetic mean (the main curve) and plus/minus one standard deviation (the shaded region). As is evident, the CS scheme leads to a much less severe scaling. Further, as predicted by our main result, this scaling behaves as $m \propto n \log(n)$. On the other hand, MCS exhibits a severe scaling. When $d = 2$, for examples, one needs over 2000 sample points whenever $n \geq 30$. Notice that this exceedingly poor scaling for polynomial regression with MCS has been widely documented in the non-gradient augmented setting [57, 90, 114]. These experiments show gradient-augmented sampling also suffers from a similar effect.

As noted in Section 5, a limitation of this approach is that the benefit it delivers depends very much on the problem setting. For example, suppose we change the domain $D$ to the bounded hypercube $D = [-1,1]^d$ and consider polynomial regression using orthonormal polynomials with respect to the uniform probability measure on $D$. These polynomials can be constructed using tensor-products of the one-dimensional Legendre polynomials, much as was done in the previous case with the Hermite polynomials. In particular, analogous recurrence relations to (B.2) and (B.3) hold [28, Sec. 2.3], which allows for efficient construction of the least-squares matrix $A$ once more. In Fig. 6 we display the approximation errors and condition numbers for MCS and CS for this problem. The comparison with Fig. 1 and Fig. 4 is striking. Especially in higher ($d \geq 4$) dimensions MCS performs nearly as well as CS, both in terms of the approximation error and condition number. This behaviour is also reflected in scaling computations, which are shown in the third row of Fig. 6. In low dimensions, MCS suffers from a worse scaling that CS, but it is nowhere near as severe as in the case of unbounded domains. Moreover, as the dimension increases the scalings become much closer. Note that these observations are well known in the non-gradient augmented setting [57, 90].

### B.7 Variation involving $C > 1$

We conclude this appendix by briefly describing several variations on the gradient-augmented regression problem that take advantage of the multimodal ($C > 1$) formulation of the general sampling framework.

**Partial gradient samples**

In some of the aforementioned applications, one may only be able to afford fewer gradient samples than function evaluations [56, 103]. This can be handled as followed. Let $C = 2$, $m_1$ be the number of function only samples and $m_2$ be the number of gradient-augmented samples. Then we define $\mathbb{X} = H^1_\rho(\mathbb{R}^d)$, $D_1 = D_2 = \mathbb{R}^d$, $\rho_1 = \rho_2 = \rho$, $\mathbb{Y}_1 = (\mathbb{C}, \langle \cdot, \cdot \rangle_2)$, $\mathbb{Y}_2 = (\mathbb{C}^{d+1}, \langle \cdot, \cdot \rangle_2)$ and

$$L_1(\theta)(f^*) = c_0 f^*(\theta), \qquad L_2(\theta)(f^*) = (c_k \partial_k f^*(\theta))_{k=0}^d,$$

where $c_0 = 1/\sqrt{2}$ and $c_k = 1$ otherwise. Then we have

$$\sum_{c=1}^C \int_{D_c} \|L(\theta)(f^*)\|^2_{\mathbb{Y}_c} \, \mathrm{d}\rho_c(\theta) = \int_D |f^*(\theta)|^2 \, \mathrm{d}\rho(\theta) + \sum_{k=0}^d \int_{\mathbb{R}^d} |\partial_k f^*(\theta)|^2 \, \mathrm{d}\rho(\theta) = \|f^*\|^2_{H^1_\rho(\mathbb{R}^d)}.$$

Thus, empirical nondegeneracy holds (2.5) as before.

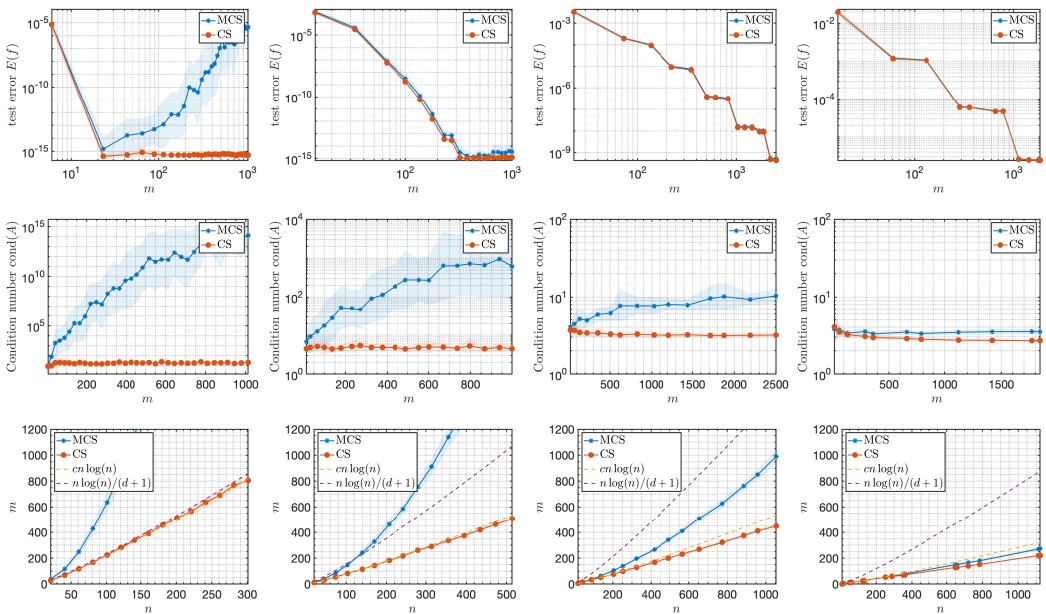

Figure 6: **CS for gradient-augmented polynomial regression.** Plots of the polynomial regression error (top row), condition number (middle row) and scalings (bottom row) for $d = 1, 2, 4, 8$ (left to right) when using the Legendre polynomials on $D = [-1, 1]^d$ instead of the Hermite polynomials on $D = \mathbb{R}^d$. These rows recreate Figs. 1, 4 and 5, respectively, for this setting.

## Hierarchical sampling

A limitation of the standard CS is that it is not well suited to *hierarchical* approximation. Often in practice, rather than a single approximation, one may wish to compute a sequence of approximations $\hat{f}_1, \hat{f}_2, \ldots$ it a nested sequence of subspaces $\mathbb{F}_1 \subset \mathbb{F}_2 \subset \cdots$ using a nested sequence of samples $\{\theta_i\}_{i=1}^{m_1} \subset \{\theta_i\}_{i=1}^{m_2} \subset \cdots$. Standard CS is not well suited to this problem, since when the subspace changes from $\mathbb{F}_k$ to $\mathbb{F}_{k+1}$, so does the generalized Christoffel function. Therefore the existing points are drawn from the wrong distribution for optimal sampling in the new subspace $\mathbb{F}_{k+1}$.

A means to overcome this problem was presented in [91] (see also [2]). It can be cast into our general framework using $C = n = \dim(\mathbb{F})$ sampling operators. These works consider the unaugmented learning problem. However, it is readily extended to the gradient-augmented setting.

The basic idea is to define the $i$th sampling measure as

$$\mathrm{d}\mu_i(\theta) = \nu_i(\theta)\,\mathrm{d}\rho(\theta), \quad \nu_i(\theta) = \sum_{k=0}^{d} |\partial_k \Phi_{\nu^i}(\theta)|^2, \qquad i = 1, \ldots, n. \tag{B.14}$$

Notice that this is a probability measure, since the $\Phi_{\nu^i}$ are orthonormal in $H_\rho^1(\mathbb{R}^d)$, Then, assuming that $m = kn$ for some $k \in \mathbb{N}$, we draw $m_i = k$ samples i.i.d. according to $\mu_i$ for each $i = 1, \ldots, n$, giving $m$ samples in total. We omit the full analysis of this case, but we remark in passing that this leads to near-optimal sample complexity. This is due, in essence, to the fact that

$$\sum_{i=1}^{n} \nu_i(\theta) = \widetilde{K}(\mathbb{F})(\theta),$$

where $\widetilde{K}(\mathbb{F})$ is as in (B.10). The key point is that (B.14) is well suited to hierarchical approximation, since it draws $k$ samples corresponding to each basis function $\Phi_{\nu^i}$. If the subspace $\mathbb{F} = \mathrm{span}\{\Phi_{\nu^i} : i = 1, \ldots, n\}$ is augmented to $\mathbb{F}' = \mathrm{span}\{\Phi_{\nu^i} : i = 1, \ldots, n'\}$, where $n' \geq n$, then we can recycle the existing samples, and draw additional samples for the new basis functions $\Phi_{\nu^i}, i = n + 1, \ldots, n'$. See [2, 91] for further details.

# C Fourier imaging with generative models

In this appendix, we describe the example considered in Section 3.2. We consider a discrete imaging scenario where the object to recover is a $d$-dimensional image of size $n \times \cdots \times n$. In this paper, we consider 3D imaging, i.e., $d = 3$.

Let $N = n^d$ and $f^* \in \mathbb{C}^N$ be the complex-valued vectorized image. Let $F \in \mathbb{C}^{N \times N}$ be the matrix of the $d$-dimensional Discrete Fourier Transform (DFT) of length $n$ with normalization $F^* F = I$. In (subsampled) Fourier imaging, one selects $m \leq N$ frequencies (typically, $m \ll N$), which correspond to $m$ rows of $F$. Let $\Omega \subseteq \{1, \ldots, N\}$, $|\Omega| = m$, be the set of frequencies sampled. Then the noisy measurements of the unknown image $f^*$ take the form

$$P_\Omega F f^* + e,$$

where $e \in \mathbb{C}^m$ is noise and $P_\Omega \in \mathbb{C}^{m \times N}$ is the *row selector* matrix, i.e., $P_\Omega F$ selects the rows of $F$ corresponding to the indices in $\Omega$.

## C.1 Formulation in terms of the general sampling framework

Depending on the imaging modality, it may be possible to select individual frequencies according to some sampling strategy. However, in applications such as MRI, this may not be possible. Typically, the scanner is only allowed to sample along piecewise smooth curves in frequency space [80, 89]. Fortunately, as we now describe, both models can be cast in our general sampling framework.

Let $\mathbb{X} = \mathbb{X}_0 = \mathbb{C}^N$ with the Euclidean inner product and norm and $C = 1$. Let $\Delta_1, \ldots, \Delta_M \subseteq \{1, \ldots, N\}$ be a partition of $\{1, \ldots, N\}$, i.e., $\cup_{i=1}^M \Delta_i = \{1, \ldots, N\}$. For convenience, we assume the $\Delta_i$ have the same cardinality, i.e., $|\Delta_i| = p$, $\forall i$. Then we let $D = \{1, \ldots, M\}$, $\rho$ be the uniform probability measure on $D$ and $\mathbb{Y} = \mathbb{C}^p$ with the Euclidean inner product. Next, we define the sampling operator

$$L : D \to \mathcal{B}(\mathbb{C}^N, \mathbb{C}^p),$$

by

$$L(i)(x) = \sqrt{M} \left( (Fx)_j \right)_{j \in \Delta_i} \in \mathbb{C}^p, \quad \forall x \in \mathbb{C}^N. \tag{C.1}$$

In other words, $L(i)$ is the linear operator that outputs the Fourier transform of an image $x$ at the frequencies in the $i$th partition element $\Delta_i$. Observe that Assumption 2.4 holds for this example with $\alpha = \beta = 1$. Indeed, for any $x \in \mathbb{C}^N$, we have

$$\int_D \|L(\theta)(x)\|^2 \, \mathrm{d}\rho(\theta) = \frac{1}{M} \sum_{i=1}^M \|L(i)(x)\|_\mathbb{Y}^2 = \sum_{i=1}^M \sum_{j \in \Delta_i} |(Fx)_j|^2 = \|Fx\|_2^2 = \|x\|_2^2. \tag{C.2}$$

With this in hand, our goal now is to design a suitable sampling measure $\mu$ on $D = \{1, \ldots, M\}$, so that each random draw from $\mu$ corresponds to a sample of frequencies from the corresponding partition element. Let us first describe two examples.

**Example C.1 (Sampling individual frequencies)** Suppose the modality allows for sampling one frequency at a time. Here, we let $M = N$ and define $\Delta_i = \{i\}$, $i = 1, \ldots, N$.

We consider Example C.1 later in Fig. 8. Fig. 2 in the main paper is, however, based on the following example.

**Example C.2 (Sampling horizontal lines in $k$-space)** *Phase encoding* is a common sampling strategy in MRI. Here one fully samples in the first $k$-space direction and subsamples in the others. Thus, the scanner traverses a piecewise smooth trajectory in $k$-space. Mathematically, this can be formulated as follows. Let $d = 3$ for concreteness (other values of $d$ can be easily handled), $N = n^3$ and $M = n^2$. 3D frequencies are naturally enumerated over the set of multi-indices $\{1, \ldots, n\}^3$. Now consider a partition $\Delta'_{i_1 i_2}$, $i_1, i_2 = 1, \ldots, n$, of this set defined by

$$\Delta'_{i_1 i_2} = \{(j, i_1, i_2) : j \in \{1, \ldots, n\}\}.$$

Thus, $\Delta'_{i_1 i_2}$ consists of $k$-space values along a horizontal line with $y$-$z$ coordinate $(i_1, i_2)$. We now let $\Delta_{i_1 i_2} \subset \{1, \ldots, N\}$ be the image of $\Delta'_{i_1 i_2}$ with respect to whatever ordering is used to convert triples $(i, j, k) \in \{1, \ldots, n\}^3$ to single indices in $\{1, \ldots, N\}$ (in our experiments, we use lexicographical ordering). Then the sets $\Delta_{i_1 i_2}$ define a partition of $\{1, \ldots, N\}$, as required.

Note that the above formulation is by no means limited to these two examples. Many other popular sampling strategies can equally well be considered. This includes radial line sampling, spiral sampling and various other strategies used in practice. We remark in passing that the setup described here is closely related to the *block sampling* setup in compressed sensing. See [18, 22].

**Remark C.3** It is worth noting two extensions of this setup. First, in practice one may wish to consider overlapping partitions $\Delta_1, \ldots, \Delta_M$. Here, one requires a small modification of the sampling operators to ensure Assumption 2.4 holds. For an index $j \in \{1, \ldots, N\}$ let $r_j \in \mathbb{N}$ be the number of partition elements in which $j$ appears, i.e., $r_j = |\{i : j \in \Delta_i\}|$. Then define

$$L(i)(x) = \sqrt{M}((Fx)_j/\sqrt{r_j})_{j \in \Delta_i}. \tag{C.3}$$

We now argue as in (C.2) to deduce that empirical nondegeneracy holds with $\alpha = \beta = 1$.

Second, it is sometime useful in practice to allow the partition elements to have different cardinalities, i.e., $|\Delta_i| = p_i$, $i = 1, \ldots, M$. This can be handled straightforwardly by zero padding. Let $\mathbb{Y} = \mathbb{C}^p$, where $p = \max_{i=1,\ldots,M}\{p_i\}$. Then we define $L(i) : \mathbb{C}^N \to \mathbb{C}^p$ by appending $p - p_i$ zeros to the vector defined in (C.1) (or (C.3) if the partition is overlapping).

## C.2   The approximation space: compressive imaging with generative models

The approach considered in this paper is based on the work [20] which introduced the generative compressed sensing framework. Their subsequent work on posterior sampling further delved into the problem [68]. In compressed sensing, the primary objective is to leverage the inherent structure in the solutions of underdetermined systems of noisy linear measurements. Most of the existing literature in this field focuses on imposing various forms of sparsity, such as standard sparsity, joint sparsity, hierarchical sparsity, or block sparsity [5, 48]. However, the generative compressed sensing framework takes a different approach. It assumes that the solutions to these problems lie near the range of a generative model $G : \mathbb{R}^p \to \mathbb{R}^N$. By employing a well-trained generative model $G$ that is relevant to the reconstruction task at hand, one can solve an optimization problem to reconstruct images or other signals. In this case, the approximation space $\mathbb{F} = \mathrm{range}(G) \subset \mathbb{R}^N$, i.e., the range of the generative model $G$. This space can be seen as a low-dimensional manifold embedded in a higher-dimensional space. Even for simple DNN models this space is typically nonlinear, and certainly for the complex models considered for the experiments related to this section.

Importantly, the generative compressed sensing framework is not limited to image reconstruction but has broader applicability. The recent work [15] extends the analysis from [20] to the problem of demixing with subgaussian matrices. Other notable contributions include variational networks [59], generative adversarial networks trained with least-squares and pixel-wise $\ell_1/\ell_2$ cost functions [87], generative adversarial network frameworks for structure preserving compressive sensing MRI [40], and model-based approaches combined with regularization and unrolled network architectures [7]. Furthermore, a recent study considers the semi-parametric single-index model with a generative prior under a Gaussian measurement assumption [81]. These studies, along with other diverse works, demonstrate the impressive versatility of the generative compressed sensing approach.

## C.3   The generalized Christoffel function

Let $\mathbb{F} \subseteq \mathbb{X}$, where, as stated above, $\mathbb{X} = \mathbb{C}^N$ with the Euclidean inner product. Denote the Euclidean norm by $\|\cdot\|$. Then the generalized Christoffel function is defined by

$$K(\mathbb{F})(i) = \sup\left\{\frac{\|L(i)(f)\|^2}{\|f\|^2} : f \in \mathbb{F}\right\} = M \sup\left\{\frac{\sum_{j \in \Delta_i} |(Ff)_j|^2}{\|f\|^2} : f \in \mathbb{F}\right\} \tag{C.4}$$

for $i \in \{1, \ldots, M\}$. Since $D = \{1, \ldots, M\}$ is a discrete set, we consider discrete probability measures. Denote such a measure by $(\mu_i)_{i=1}^M$, i.e., $I \sim \mu$ if $\mathbb{P}(I = i) = \mu_i$. Then sampling according to the Christoffel function gives the values

$$\mu_i = \frac{K(\mathbb{F})(i)}{M\kappa(\mathbb{F})}, \quad i \in \{1, \ldots, M\},$$

where

$$\kappa(\mathbb{F}) = \int_D K(\mathbb{F})(\theta)\,\mathrm{d}\rho(\theta) = \frac{1}{M}\sum_{i=1}^M K(\mathbb{F})(i).$$

---
**Algorithm 1** Computing $\widetilde{K}$

---
**input:** Generative model $G$, number of iterations $t$, input distribution $\rho$ on the domain of $G$, DFT matrix $F$.
**initialize:** $\widetilde{K} = 0$
**for** $l = 1$ to $t$ **do**
    Generate two i.i.d. realizations $f_1, f_2$ via the generative model, i.e., $f_i = G(z_i)$ for $z_i \sim_{\text{i.i.d.}} \rho$
    Compute $a_i = M \sum_{j \in \Delta_i} |(Fg)_j|^2 / \|v\|^2$ for $g = f_1 - f_2$ and $i = 1, \dots, M$
    **for** $i = 1$ to $M$ **do**
        Replace $\widetilde{K}(i) = \max\{\widetilde{K}(i), a_i\}$
    **end for**
**end for**
**output:** Vector $\widetilde{K}$ with updated entries.

---

Note that the corresponding weight function is

$$\nu(i) = M\mu_i = \frac{K(\mathbb{F})(i)}{\kappa(\mathbb{F})}, \quad i \in \{1, \dots, M\}.$$

In order to apply Theorem 4.2 to this example, we need to identify a (subspace) cover $\widetilde{\mathbb{F}} = \mathbb{F}_1 \cup \cdots \cup \mathbb{F}_d$ of the set $\mathbb{F} - \mathbb{F}$, where, as above $\mathbb{F}$, is the range of a generative model. It is therefore important to first ask when the set $\mathbb{F} - \mathbb{F}$ admits such a cover for a reasonable value of $d$ (since $d$ appears logarithmically in the bound). This question has been addressed in [95, Lem. 3.1] in the case of ReLU generative models (see also [16, Lem. S2.2]). Roughly speaking, if $G : \mathbb{R}^p \to \mathbb{R}^N$ has $L$ layers and the widths of its hidden layers are $\mathcal{O}(p)$, then $\mathbb{F} - \mathbb{F}$ can be covered exactly with $\log(d) \lesssim pL$ subspaces of dimension $2p$.

There are two factors that prohibit direct application of Theorem 4.2 to this example. First, the generative models we use are not simple ReLU DNNs. Second, even if they were, identifying the aforementioned union of subspaces that form the cover and computing the constant $K(\widetilde{\mathbb{F}})$ is likely computationally infeasible, since $d$ grows exponentially in $pL$.

As noted in Section 5, however, we believe the estimate involving $K(\widetilde{\mathbb{F}})$ in Theorem 4.2 to be non-sharp, and conjecture that it should suffice to consider $K(\mathbb{F} - \mathbb{F})$ instead. Therefore, in what follows, we compute with (an approximation to) the former. Thus, in this example, the theory does not completely apply. Nevertheless, as noted, we see significant benefits from CS for this example.

### C.4 Numerical approximation to the generalized Christoffel function

In view of (C.4), computing $K(\mathbb{F} - \mathbb{F})$ involves solving $M$ (nonlinear, nonconvex) maximization problems over the input space $\mathbb{R}^p$ of the generative model $G$. This is extremely computationally expensive, since both $p$ and $M$ are large in practice. In our examples, $M = n^3$ (Theorem C.1) or $M = n^2$ (Theorem C.2), where $n = 128$, and $p = 1024$ (see next).

Therefore, in this work we consider a different approach, which is based on an iterative, random sampling procedure for computing an approximation $\widetilde{K}$ to $K$. This procedure is presented in Algorithm 1 and involves repeated random samples from the input distribution of the generative model to construct candidate elements $f_1 - f_2 \in \mathbb{F} - \mathbb{F}$. It has the advantages of being simple and not computationally intensive. Moreover, as our examples show, sampling from the resulting $\widetilde{K}$ achieves the objective of significantly outperforming MCS.

The convergence of Algorithm 1 in computing $\widetilde{K}$ can be seen in Fig. 7 where the relative $\ell_2$ error of the iterates are computed with the final $\widetilde{K}$ after the algorithm returns.

### C.5 Additional experimental details

The experiments in Figures 2 & 8 were conducted using a single generative neural network model trained on brain MRI data. The `braingen-0.1.0` generative model was obtained from `https://github.com/neuronets/trained-models/releases` and was trained using the `nobrainer`

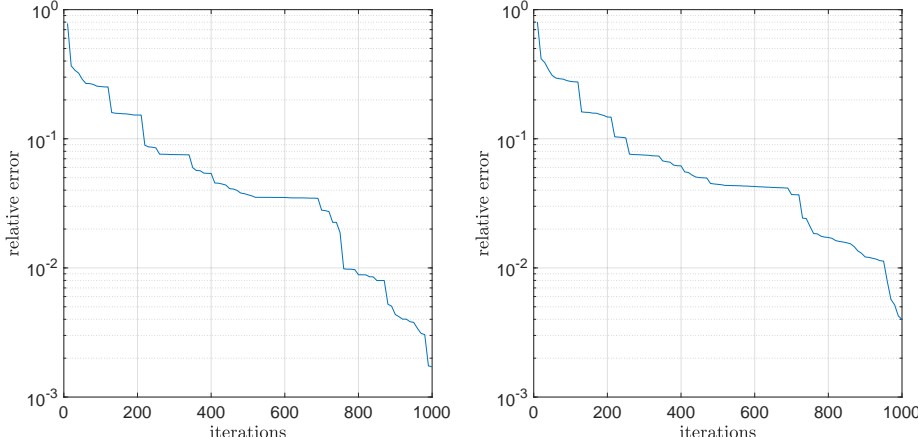

Figure 7: Convergence of the relative $\ell_2$ error of iterations of Algorithm 1 in computing $\widetilde{K}$ for (left) Example C.2 and (right) Example C.1.

framework [70] available under Apache License, Version 2.0. This model was trained using a progressively growing strategy [71] for generation of T1-weighted brain MR scans. The model takes as input a latent vector $v \in \mathbb{R}^{1024}$, here generated by sampling from the standard normal distribution, i.e. $v_i \sim N(0, 1)$, $i = 1, \ldots, 1024$. The output dimension of the selected `braingen-0.1.0` model is $N = n^3$ where $n = 128$.

First we generate $\widetilde{K}$ using 1000 iterations of Algorithm 1. To generate the results, we chose an image from the range of the generator using random seed '12345'. We then ran 20 random trials reconstructing this image with the same computed $\widetilde{K}$, solving problem (2.2) using 5000 iterations of the Adam optimizer [72], implemented in TensorFlow. A constant learning rate multiplier of 100 was used to improve the convergence in the gradient update step. The experiments were conducted on a single computer with an AMD Ryzen 9 5950x, 64 GB of DDR4-3600 RAM, and a NVIDIA GeForce RTX 3090 Ti. The total runtime for these experiments was approximately 2 weeks. In Fig. 2 we consider the setup of Example C.2. In Fig. 8 we consider the setup of Example C.1. As in Section B.5, our error plots display the mean PSNR value and also a shaded region corresponding to one standard deviation. The PSNR is defined as

$$\text{PSNR} = 20 \cdot \log_{10}\left(\frac{\text{MAX}_{f^*}}{\sqrt{\text{MSE}}}\right),$$

where MSE is the *Mean Squared Error*

$$\text{MSE} = \frac{1}{N}\sum_{i=1}^{N}|f_i^* - \hat{f}_i|^2.$$

Here $f^*$ is the ground truth image, $\hat{f}$ is its reconstruction and $\text{MAX}_{f^*}$ is the maximum possible pixel value of the image, here equal to 255.

## C.6 Further experiments and discussion

In Fig. 2 consider the strategy of sampling horizontal lines in $k$-space, see Example C.2. In Fig. 8 we perform the same experiment with the strategy of sampling individual frequencies in $k$-space, see Example C.1. While CS with line sampling performs similarly to CS with individual frequency sampling, MCS line sampling in $k$-space can be seen to have a 15-20% drop in image reconstruction quality over MCS individual frequency sampling in $k$-space.

Although, as noted, the theory does not directly apply to this example, it is worth examining the relevant constants to gain some further insight into the performance of CS over MCS. For MCS, Theorem 4.2 suggests that the sample complexity depends on the term

$$C_{\text{MCS}} = \operatorname*{ess\,sup}_{i \sim \rho} \widetilde{K}(\mathbb{F} - \mathbb{F})(i) = \max_{i=1,\ldots,M} \widetilde{K}(\mathbb{F} - \mathbb{F})(i),$$

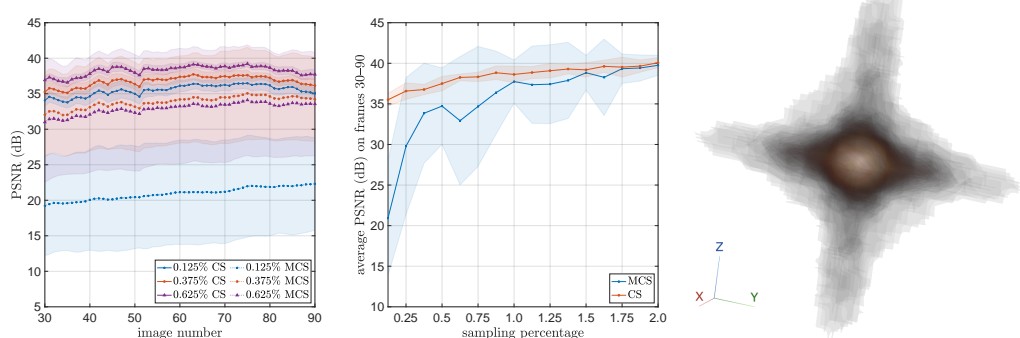

Figure 8: **CS for MRI reconstruction using generative models.** Plots of (left) the average PSNR vs. image number of the 3-dimensional brain MR image for both CS and MCS methods at 0.125%, 0.375%, and 0.625% sampling percentages, (middle) the average PSNR computed over the frames 30 to 90 of the image vs. sampling percentage for both CS and MCS methods, and (right) the empirical function $\widetilde{K}$ used for the CS procedure. In this figure, we consider Example C.1 instead of Example C.2, which was used in Fig. 2.

Table 1: Values of $C_{\mathsf{MCS}}$ and $C_{\mathsf{CS}}$ for experiments with Examples C.1 & C.2.

|  | Example C.1 | Example C.2 |
|---|---|---|
| $C_{\mathsf{MCS}}$ | $3.2794 \times 10^5$ | 4616.4 |
| $C_{\mathsf{CS}}$ | 7.6029 | 3.9595 |

i.e., the maximum of the approximate Christoffel function $\widetilde{K}$. Conversely, for CS it suggests that the sample complexity depends on its average, i.e.,

$$C_{\mathsf{CS}} = \int_D \widetilde{K}(\mathbb{F} - \mathbb{F})(\theta)\,\mathrm{d}\rho(\theta) = \frac{1}{M}\sum_{i=1}^M \widetilde{K}(\mathbb{F} - \mathbb{F})(i).$$

In Table 1 we report these values for the two experiments. As is evident, CS leads to a greatly reduced constant in both cases over MCS, which is in agreement with the general improvement in PSNR for this method. We note, however, that these constants do not capture all the phenomena seen in the experiments. For example, MCS performs worse for Example C.2 than it does for Example C.1. Yet the size of the constants would suggest otherwise. This is an interesting gap between the theory and practice.

### C.7 Variations involving $C > 1$

The setup described so far (sampling the Fourier transform) is a discrete model for so-called *single-coil* MRI. However, modern MR scanners are typically employ parallel coils. Here, multiple receive coils acquire Fourier measurements in parallel. Let $C \geq 1$ be the number of coils. Then one can model the measurements in parallel MRI as

$$b_c = P_{\Omega_c} F S_c f^* + e_c, \quad c = 1, \ldots, C.$$

Here $b_c \in \mathbb{C}^{m_c}$ are the measurements in the $c$th coil, $\Omega_c \subseteq \{1, \ldots, N\}$ with $|\Omega_c| = m_c$ and $S_c$ is a diagonal matrix, known as the *sensitivity profile* of the coil. See, e.g., [76, 116].

This is a multimodal sampling model that can be cast into our framework. Indeed, we replace the single sampling operator (C.1) with $C$ sampling operators defined by

$$L_c(i)(x) = \sqrt{M/N}((FS_c x)_j)_{j \in \Delta_i} \in \mathbb{C}^p, \quad \forall x \in \mathbb{C}^N,\, c = 1, \ldots, C.$$

Note that empirical nondegeneracy (2.5) is easily shown to be equivalent to the condition

$$\alpha \leq \sum_{c=1}^C |(S_c)_{ii}|^2 \leq \beta, \quad \forall i = 1, \ldots, N.$$

Sensitivity profiles are often designed so that this property holds with moderate $\alpha$, $\beta$. Note that in practice the sensitivity profiles are not known, and have to be computed either via a calibration pre-scan or learned during the process of recovering the image. See, e.g., [5, Sec. 3.5.2-3.5.3] and references therein.

# D  Solving PDEs using PINNs

In this appendix, we provide further details for the example considered in Section 3.3 involving solving the nonlinear Burger's equation using PINNs. As mentioned, we implement CS in an adaptive manner for this example, using a strategy we term *Christoffel Adaptive Sampling (CAS)*. This strategy not only applies to the numerical solution of PDEs, but to any regression problem with DNNs and general, multimodal data. We then apply this to the numerical solution of PDEs through the framework of PINNs.

## D.1  PINNs for solving PDEs

Consider a spatial domain $D \subset \mathbb{R}^d$ and time interval $I = [0, T]$, where $d \geq 1$ and $T \geq 0$, respectively. Then we seek to learn an unknown function $u^* : D \times [0, T] :\to \mathbb{C}$ given as the solution of a PDE of the form

$$\begin{aligned}
\mathcal{L}u &= f \quad \text{in } D \times I \\
u &= u_{\mathsf{i}} \quad \text{in } D \times \{0\} \\
u &= u_{\mathsf{b}} \quad \text{on } \partial D \times (0, T].
\end{aligned}$$

Here $\mathcal{L}$ is a differential operator in the space and time variables, $u_{\mathsf{i}}$ is a known initial condition and $u_{\mathsf{b}}$ is a known boundary condition. Throughout this section, we use $f$ to denote the right-hand side of the PDE and $u^*$ to denote the target function (instead of $f^*$), as is customary in the numerical PDE literature.

The general idea behind PINNs [69, 105] is to consider training data of the form

$$\{((x_i^{\mathsf{d}}, t_i^{\mathsf{d}}), f(x_i^{\mathsf{d}}, t_i^{\mathsf{d}}))\}_{i=1}^{m_{\mathsf{d}}} \cup \{((x_i^{\mathsf{i}}, t_i^{\mathsf{i}}), u^*(x_i^{\mathsf{i}}, t_i^{\mathsf{i}}))\}_{i=1}^{m_{\mathsf{i}}} \cup \{((x_i^{\mathsf{b}}, t_i^{\mathsf{b}}), u^*(x_i^{\mathsf{b}}, t_i^{\mathsf{b}}))\}_{i=1}^{m_{\mathsf{b}}}$$

where $(x_i^{\mathsf{d}}, t_i^{\mathsf{d}}) \in D \times (0, T]$ are sample points in the interior of the domain, $(x_i^{\mathsf{i}}, t_i^{\mathsf{i}}) \in D \times \{0\}$ are sample points at the initial condition and $(x_i^{\mathsf{b}}, t_i^{\mathsf{b}}) \in \partial D \times (0, T]$ are sample points on the space-time boundary of the domain. Typically, these sample points are generated via MCS. One then seeks a minimizer of the problem

$$\min_{u \in \mathbb{U}} \frac{1}{m_{\mathsf{d}}} \sum_{i=1}^{m_{\mathsf{d}}} |\mathcal{L}u(x_i^{\mathsf{d}}, t_i^{\mathsf{d}}) - f(x_i^{\mathsf{d}}, t_i^{\mathsf{d}})|^2 + \frac{\lambda_{\mathsf{i}}}{m_{\mathsf{i}}} \sum_{i=1}^{m_{\mathsf{i}}} |u(x_i^{\mathsf{i}}, t_i^{\mathsf{i}}) - u_0(x_i^{\mathsf{i}}, t_i^{\mathsf{i}})|^2$$

$$+ \frac{\lambda_{\mathsf{b}}}{m_{\mathsf{b}}} \sum_{i=1}^{m_{\mathsf{b}}} |u(x_i^{\mathsf{b}}, t_i^{\mathsf{b}}) - u_{\mathsf{b}}(x_i^{\mathsf{b}}, t_i^{\mathsf{b}})|^2,$$

where $\mathbb{U}$ is a class of DNNs $u : \mathbb{R}^d \times \mathbb{R} \to \mathbb{R}$ and $\lambda_{\mathsf{i}}, \lambda_{\mathsf{b}} > 0$ are hyperparameters that control fitting the PDE versus fitting the boundary/initial conditions. Note that we write $\mathbb{U}$ and $u \in \mathbb{U}$ throughout this section instead of $\mathbb{F}$ and $f \in \mathbb{F}$, to avoid confusion with the right-hand side of the PDE.

## D.2  PINNs formulation in terms of the general sampling framework

We now show how to reformulate PINNs in terms of the general sampling framework. Since there are three types of data, we let $C = 3$. Then we define $D_1 = D \times (0, T)$, $D_2 = D \times \{0\}$ and $D_3 = \partial D \times (0, T]$, and let $\rho_1$, $\rho_2$ and $\rho_3$ be measures on these domains. We then consider the sampling operators

$$\begin{aligned}
L_1(\theta)(u) &= \mathcal{L}u(\theta), \quad y = (x, t) \in D_1 \\
L_2(\theta)(u) &= \sqrt{\lambda_2}u(\theta), \quad y = (x, t) \in D_2 \\
L_3(\theta)(u) &= \sqrt{\lambda_3}u(\theta), \quad y = (x, t) \in D_3.
\end{aligned} \tag{D.1}$$

Next, we let $\nu_c : D_c \to \mathbb{R}$ and $\mu_c$ be as in Assumption 2.2. Finally, for each $c = 1, 2, 3$, we draw $m_c$ samples $(\Theta_{ic})_{i=1}^{m_c}$, where $\Theta_{ic} = (X_{ic}, T_{ic})$, and consider the weighted regression problem

$$\min_{u \in \mathbb{U}} \sum_{c=1}^{3} \frac{1}{m_c} \sum_{c=1}^{m_c} \frac{1}{\nu_c(\Theta_{ic})} |L_c(\Theta_{ic})(u) - L_c(\Theta_{ic})(u^*)|^2. \tag{D.2}$$

There are several factors that prohibit direct application of CS to (D.2). The first is that the sampling operator $L_1$ is potentially nonlinear, since it involves the generally nonlinear PDE operator $\mathcal{L}$. This means that there is no theoretical basis for the use of CS. However, even if it were linear (e.g., for a linear PDE), it could be very difficult to compute the generalized Christoffel functions $K_c(\mathbb{U})$ in practice, since evaluating $K_c(\mathbb{U})$ at a single point $\theta$ involves solving a maximization problem over the space $\mathbb{U}$ (a class of DNNs). The second, and arguably more critical issue, is that PINNs are typically implemented in the highly overparametrized regime, where the number of DNN parameters greatly exceeds the amount of training data. CS is designed to provide sampling measures that ensure 'good' recovery over the whole of the space $\mathbb{U}$. But this is an extremely conservative approach for DNN-based learning problems, since only certain DNNs are likely to be encountered in training.

We tackle these problems in reverse order. For the second, we present an adaptive sampling approach for deep learning. This is described in the next subsection. Note that this does not assume the underlying problem is a PDE. Our presentation follows the lines of the general sampling framework. Having done this, we then describe its application to PINNs, including suitable modifications of the sampling measures and loss function that take into account the nonlinearity of the PDE operator.

### D.3 CAS for deep learning with general, multimodal samples

In this section, we consider DL based on general, multimodal samples as introduced in our general sampling framework. What follows is an extension of a method first introduced in [3], which in turn is based on ideas from [35]. Let $\mathbb{X}$ be a Hilbert space of functions $\mathbb{R}^{d_1} \to \mathbb{R}^{d_2}$ and $\mathbb{U} \subset \mathbb{X}$ be a class of DNNs of a fixed architecture. As in Section 2, we consider sampling operators

$$L_c : D_c \to \mathcal{B}(\mathbb{X}_0, \mathbb{Y}_c), \quad c = 1, \dots, C,$$

where the $\mathbb{Y}_c$ are Hilbert spaces. We also write $u^* \in \mathbb{X}$ for the unknown function we seek to learn.

We now describe the adaptive sampling method. This exploits the so-called *adaptive basis viewpoint* of DNNs, as developed in [35]. Let $n \in \mathbb{N}$ be the number of nodes in the penultimate layer of the network architecture used to define $\mathbb{U}$ and, for convenience, assume that the output is not subject to a bias term, just a weight. Therefore, we can write any $u \in \mathbb{U}$ as

$$u = C^\top H, \tag{D.3}$$

where $H : \mathbb{R}^{d_1} \to \mathbb{R}^n$ is a DNN of the same architecture as $u$, except for the final layer, and $C = (c_{ij})_{i,j=1}^{n,d_2}$ is the weight matrix on the output layer. Now let $u_i : \mathbb{R}^{d_1} \to \mathbb{R}$ be the $i$th component of $u$, $i = 1, \dots, d_2$, and $H_i : \mathbb{R}^{d_1} \to \mathbb{R}$ be the $i$th component of $H$, $i = 1, \dots, n$. Then we can express $h_j$ as

$$u_j = \sum_{i=1}^{n} c_{ij} H_i, \quad j = 1, \dots, d_2.$$

In particular, each $u_j$ is an element of the subspace $\mathrm{span}\{H_1, \dots, H_n\}$.

Now consider an initial DNN $\hat{u} = \hat{u}^{(0)} = (\hat{u}_j^{(0)})_{j=1}^{d_2}$ obtained via a standard (random) initialization strategy. Write $\hat{u}$ as in (D.3) as

$$\hat{u}^{(0)} = (\widehat{C}^{(0)})^\top \widehat{H}^{(0)},$$

let $\widehat{H}^{(0)} = (\widehat{H}_i^{(0)})_{i=1}^n$ and consider the subspace

$$\mathbb{H}^{(0)} = \mathrm{span}\{\widehat{H}_1^{(0)}, \dots, \widehat{H}_n^{(0)}\}$$

so that $\hat{u}_j^{(0)} \in \mathbb{H}^{(0)}$, $j = 1, \dots, d_2$. Let $p_0 = \dim(\mathbb{H}^{(0)}) \leq n$. We now perform CS using this subspace. In other words, we define sampling measures

$$\mathrm{d}\mu_c^{(1)}(\theta) = \nu_c^{(1)}(\theta)\,\mathrm{d}\rho_c(\theta), \quad \nu_c^{(1)}(\theta) = \frac{K_c(\mathbb{H}^{(0)})(\theta)}{p_0}, \qquad c = 1, \dots, C,$$

then, for each $c$, we generate $m_c^{(1)}$ sample points $\Theta_{ic}$ i.i.d. from $\mu_c^{(1)}$ and generate samples as

$$y_{ic} = L_c(\Theta_{ic})(u^*), \quad i = 1, \ldots, m_c^{(1)}, \ c = 1, \ldots, C.$$

Given these samples, we then train a new DNN by applying a standard optimizer to the minimization problem

$$\min_{u \in \mathbb{U}} \sum_{c=1}^{C} \frac{1}{m_c^{(1)}} \sum_{i=1}^{m_c^{(1)}} \frac{1}{\nu_c^{(1)}(\Theta_{ic})} \| y_{ic} - L_c(\Theta_{ic})(u) \|_{\mathbb{Y}_c}^2.$$

Let $\hat{u}^{(1)} \in \mathbb{U}$ be resulting DNN. The idea now is to repeat this process by drawing $m_c^{(2)} - m_c^{(1)}$ new samples using CS for the new subspace $\mathbb{H}^{(1)}$ and then using the full set of $m_c^{(2)}$ samples to compute a new DNN $\hat{u}^{(2)}$. Continuing in this way, we generate a sequence of DNNs $\hat{u}^{(1)}, \hat{u}^{(2)}, \ldots \in \mathbb{U}$ approximating the unknown function $u^*$. To be precise, consider step $l \geq 1$. Let $\hat{u}^{(l-1)}$ be given, write

$$\hat{u}^{(l-1)} = (\widehat{C}^{(l-1)})^\top \widehat{H}^{(l-1)}, \qquad \widehat{H}^{(l-1)} = (\widehat{H}_i^{(l-1)})_{i=1}^n$$

and define

$$\mathbb{H}^{(l-1)} = \mathrm{span}\{\widehat{H}_1^{(l-1)}, \ldots, \widehat{H}_n^{(l-1)}\}, \qquad p_{l-1} = \dim(\mathbb{H}^{(l-1)}).$$

We use this to construct sampling measures

$$d\mu_c^{(l)}(\theta) = \nu_c^{(l)}(\theta)\, d\rho_c(\theta), \quad \nu_c^{(l-1)}(\theta) = \frac{K_c(\mathbb{H}^{(l-1)})(\theta)}{p_{l-1}}, \qquad c = 1, \ldots, C,$$

and then, for each $c$, we generate $m_c^{(l)} - m_c^{(l-1)}$ new sample points $\Theta_{ic}, i = m_c^{(l-1)} + 1, \ldots, m_c^{(l)}$, i.i.d. from $\mu_c^{(l-1)}$ and corresponding samples

$$y_{ic} = L_c(\Theta_{ic})(u^*), \quad i = m_c^{(l-1)} + 1, \ldots, m_c^{(l)}, \ c = 1, \ldots, C.$$

We then combine these with the existing $m_c^{(l-1)}$ samples, giving a total of $m_c^{(l)}$ samples for each $c$, and then use this to train a new DNN via the minimization problem

$$\min_{u \in \mathbb{U}} \sum_{c=1}^{C} \frac{1}{m_c^{(l)}} \sum_{i=1}^{m_c^{(l)}} \frac{1}{\nu_c^{(l)}(\Theta_{ic})} \| y_{ic} - L_c(\Theta_{ic})(u) \|_{\mathbb{Y}_c}^2.$$

We write $\hat{u}^{(l)} \in \mathbb{U}$ for the resulting DNN.

### D.4   CAS for PINNs

Our aim now is to apply the CAS method in the previous subsection to PINNs. In this case, we set $d_1 = d + 1$ and $d_2 = 1$. As we now describe, we make one main modification to CAS in this case, which is to replace the true generalized Christoffel functions $K_c$ with surrogates $\widetilde{K}_c$ that are more amenable to computations.

Recall that $C = 3$ in this case and that the sampling operators are as in (D.1). To define $\widetilde{K}_1$ we simply forget the differential operator $\mathcal{L}$ and define

$$\widetilde{K}_1(\mathbb{H}^{(l-1)})(x,t) = \sup \left\{ \frac{|h(x,t)|^2}{\|h\|_{L^2_{\rho_1}(D_1)}^2} : h \in \mathbb{H}^{(l-1)} \backslash \{0\} \right\}, \quad (x,t) \in D_1 = D \times (0,T).$$

Next, for the initial ($c = 2$) and boundary ($c = 3$) data, we define

$$\widetilde{K}_2(\mathbb{H}^{(l-1)})(x,t) = \sup \left\{ \frac{|h(x,t)|^2}{\|h\|_{L^2_{\rho_2}(D_2)}^2} : h \in \mathbb{H}^{(l-1)} \backslash \{0\} \right\}, \quad (x,t) \in D_2 = D \times \{0\}$$

and

$$\widetilde{K}_2(\mathbb{H}^{(l-1)})(x,t) = \sup \left\{ \frac{|h(x,t)|^2}{\|h\|_{L^2_{\rho_3}(D_3)}^2} : h \in \mathbb{H}^{(l-1)} \backslash \{0\} \right\}, \quad (x,t) \in D_3 = \partial D \times (0,T).$$

Let $\tilde\nu_c^{(l)}$ and $\tilde\mu_c^{(l)}$ denote the corresponding weight function and sampling measures, i.e.,

$$\mathrm{d}\tilde\mu_c^{(l)}(x,t) = \tilde\nu_c^{(l)}(x,t)\,\mathrm{d}\rho_c(x,t), \quad \tilde\nu_c^{(l-1)}(x,t) = \frac{\widetilde{K}_c(\mathbb{H}^{(l-1)})(x,t)}{p_{l-1,c}}, \qquad c = 1,2,3. \quad \text{(D.4)}$$

Then at step $l$ of the procedure, we generate $m_c^{(l)} - m_c^{(l-1)}$ new sample points $(X_{ic}, T_{ic})$, $i = m_c^{(l-1)} + 1, \ldots, m_c^{(l)}$, i.i.d. from $\tilde\mu_c^{(l-1)}$ and compute the corresponding samples

$$y_{ic} = L_c(X_{ic}, T_{ic})(u^*), \quad i = m_c^{(l-1)} + 1, \ldots, m_c^{(l)}, \ c = 1,2,3,$$

where the $L_c$ are as in (D.1). We then combine these with the existing $m_c^{(l-1)}$ samples and use this to train a new DNN via the minimization problem

$$\min_{u \in \mathbb{U}} \sum_{c=1}^{3} \frac{1}{m_c^{(l)}} \sum_{i=1}^{m_c^{(l)}} \frac{1}{\tilde\nu_c^{(l)}(\Theta_{ic})} |y_{ic} - L_c(X_{ic}, T_{ic})(u)|^2. \quad \text{(D.5)}$$

We term this method *CAS for PINNs*. It is summarized in Algorithm 2.

Steps 2-3 of this algorithm require some further explanation. First, as in previous examples (see Appendix B), we introduce finite grids $Z_c = \{z_{ic}\}_{i=1}^{N_c}$, $c = 1,2,3$, to represent the domains $D_c$, $c = 1,2,3$. This is to enable efficient sampling from the measures $\tilde\mu_c$, which are now discrete measures defined over the corresponding grids. See the next section for the construction of these grids.

Second, recall from Lemma E.1 that in order to compute the functions $\widetilde{K}_c$, we need to construct orthonormal bases for space

$$\mathrm{span}\{\widehat{H}_1^{(l-1)}, \ldots, \widehat{H}_N^{(l-1)}\}$$

over the finite grids $Z_{ic}$. Previously, in Appendix B.4, we did this via QR factorization. However, the dictionary $\{\widehat{H}_i^{(l-1)}\}_{i=1}^{N}$ generated in Step 1 of the algorithm is often numerically redundant. Therefore, in this case we use a truncated SVD instead. Specifically, if

$$B_c = \left(\frac{1}{\sqrt{N_c}} \widehat{H}_j^{(l-1)}(z_{ic})\right)_{i,j=1}^{N_c,N},$$

then we compute its SVD $U_c \Sigma_c V_c^*$ and define the numerical dimension

$$p_{l-1,c} = \max\{i : \sigma_{ic}/\sigma_{1c} > \delta_{\mathrm{tol}}\} \in \{1, \ldots, N\}.$$

Using this, we now obtain an orthonormal basis over the grid $Z_c$ using the first $p_{l-1,c}$ left singular vectors, i.e., the first $p_{l-1,c}$ columns of $U_c$. Given this basis, we can then compute the function $\widetilde{K}_c(\mathbb{H}^{(l-1)})$ over the grid via Lemma E.1. To precise,

$$\widetilde{K}_c(\mathbb{H}^{(l-1)})(z_{ic}) = \sum_{j=1}^{p_{l-1,c}} |(U_c)_{ij}|^2, \quad i = 1, \ldots, N_c,$$

and the corresponding measures $\tilde\mu_c$ are given by $\theta \sim \tilde\mu_c$ if

$$\mathbb{P}(\theta = z_{ic}) = \frac{\widetilde{K}_c(\mathbb{H}^{(l-1)})(z_{ic})}{p_{l-1,c}}, \quad i = 1, \ldots, N_c.$$

## D.5 Additional experimental details

In this section, we describe the setup for our experiments. The problem we consider is the Burger's equation with $D = (-1,1)$, $T = 1$,

$$\mathcal{L}u = \partial_t u + u\partial_x u - (0.01/\pi)\partial_{xx} u,$$

and initial and boundary conditions

$$u(x,0) = -\sin(\pi x), \quad x \in D,$$

---

**Algorithm 2** CAS for PINNs

---

**input:** PDE operator $\mathcal{L}$, initial and boundary condition $u_{\mathsf{i}}$ and $u_{\mathsf{b}}$. Class of DNNs $\mathbb{U}$ with fixed architecture, no bias on the output layer and penultimate layer of with $N$. Finite grids $Z_c = \{z_{ic}\}_{i=1}^{N_c} \subset D_c$, where $N_c \geq N$, $c = 1, 2, 3$, numbers of samples $0 = m_c^{(0)} < m_c^{(1)} < m_c^{(2)} < \dots$, $c = 1, 2, 3$, and tolerance $\delta_{\mathrm{tol}} > 0$.

**initialize:** Initialize the DNN $\widehat{h}^{(0)}$.

**for** $l = 1$ to $t$ **do**

Construct the dictionary $\{\widehat{H}_1^{(l-1)}, \dots, \widehat{H}_N^{(l-1)}\}$ via the relation

$$\hat{u}^{(l-1)} = \sum_{j=1}^{N} (\hat{c}_j^{(l-1)}) \widehat{H}_j^{(l-1)}.$$

Construct the sampling measures $\tilde{\mu}_c^{(l)}$, $c = 1, 2, 3$, as in (D.4).

For each $c = 1, 2, 3$, draw $m_c^{(l)} - m_c^{(l-1)}$ sample points $(X_{i,c}, T_{i,c})$, $i = m_c^{(l-1)} + 1, \dots, m_c^{(l)}$.

Train the DNN $\hat{u}^{(l)} \in \mathbb{U}$ by applying a standard optimizer to (D.5) with initialization $\hat{u}^{(l-1)}$.

**end for**

**output:** A sequence of DNN approximations $\widehat{u}^{(1)}, \widehat{u}^{(2)}, \dots, \widehat{u}^{(t)}$ to $u$.

---

and

$$u(-1, t) = u(1, t) = 0, \quad t \in (0, 1],$$

respectively. As mentioned, we employ three finite grids

$$
\begin{aligned}
Z_1 &= \{z_{i1}\}_{i=1}^{N_1} = \{(x_{i1}, t_{i1})\}_{i=1}^{N_1} \in (-1, 1) \times (0, 1], & N_1 &= 16,000, \\
Z_2 &= \{z_{i2}\}_{i=1}^{N_2} = \{(x_{i2}, t_{i2})\}_{i=1}^{N_2} \in (-1, 1) \times \{0\}, & N_2 &= 2000, \\
Z_3 &= \{z_{i3}\}_{i=1}^{N_3} = \{(x_{i3}, t_{i3})\}_{i=1}^{N_3} \in \{-1, 1\} \times (0, 1], & N_3 &= 2000.
\end{aligned}
$$

Following Appendix B, these are generated once beforehand by MCS with respect to the uniform measure over each domain.

We now describe further aspects of our experiments.

*Methods considered.* As in previous appendices, we compare CS with MCS. Specifically, we consider DNNs trained using training data constructed via CAS (i.e., Algorithm 2), and DNNs trained from standard training data obtained via MCS from the uniform measure over each domain. As in previous example, we labels these as 'CS' and 'MCS', respectively. For Algorithm 2 we use the threshold parameter $\delta_{\mathrm{tol}} = 10^{-6}$, since our computations are performed in single precision.

*Choice of architectures and initialization.* We consider fully-connected DNNs with activation function $\rho$, $L \geq 1$ hidden layers and an equal number of $N \geq 1$ nodes per hidden layer. We term these $\rho$ $L \times N$ DNNs. Since we are solving a regression problem, we assume the output layer is not subject to $\rho$. We also set the bias to be zero in this layer. For the activation function $\rho$, we consider either the Rectified Linear Unit (ReLU), hyperbolic tangent (tanh), Exponential Linear Unit (ELU), or sigmoid. These are defined as follows:

$$
\begin{aligned}
&\text{ReLU:} & \rho(x) &:= \max\{x, 0\}, \\
&\text{tanh:} & \rho(x) &:= \frac{e^{2x} - 1}{e^{2x} + 1}, \\
&\text{ELU:} & \rho(x) &:= \{x : x \geq 0, (e^x - 1) : x < 0\}, \\
&\text{sigmoid:} & \rho(x) &:= \frac{1}{1 + e^x}.
\end{aligned}
$$

Note that we choose DNNs with a fixed number of nodes $N$ for each layer. Moreover, we also focus on DNNs with depth $L$ such that the ratio $L/N$ is small. This is motivated by results in [62]. To be precise, $L/N = 0.1$ in our experiments.

For the initialization strategy, we consider weights and biases with entries drawn from the *Normal initialization* with mean 0 and variance 0.1. Having in mind the size of architectures described in this

Table 2: Number of samples per iteration for each domain

| parameter | Number of samples $m_c^{(l)}$ | | | | | | | | |
|---|---|---|---|---|---|---|---|---|---|
| $l$ | 1 | 2 | 3 | 4 | 5 | 6 | 7 | 8 | 9 |
| $c = 1$ | 400 | 1350 | 2300 | 3250 | 4200 | 5150 | 6100 | 7050 | 8000 |
| $c = 2$ | 100 | 200 | 300 | 400 | 550 | 650 | 750 | 850 | 1000 |
| $c = 3$ | 100 | 200 | 300 | 400 | 550 | 650 | 750 | 850 | 1000 |
| Total | 600 | 1750 | 2900 | 4050 | 5300 | 6450 | 7600 | 8750 | 10000 |

work, this initialization leads to smaller variance than commonly used initializations used PINNs such as the well-known Xavier and He Normal initializations. Besides, it has been shown that is an effective choice for function approximation, see [63, 65]. In all our experiments, the DNNs were initialized with the same seed to compare the effect of varying the samples.

*Optimizers for training and parametrization.* We use *Adam* as the optimizer in combination with an exponentially decaying learning rate with respect to the number of epochs. For both CAS and the procedure based on MCS, we train the networks following a schedule number of epochs, described as follows:

$$n^{(0)} = 5000, \quad n^{(l)} = n^{(0)} \times l, \quad l \geq 1.$$

For the CAS strategy, we train the networks for $n^{(l)}$ epochs on each new set of samples, using the weights and biases from the previous set as initialization. We repeat this procedure 9 times, i.e. we train for a total of $225,000$ epochs. For MCS, we train the network for a total of $225,000$ epochs as well, stopping every $n^{(l)}$ epochs to add more data points drawn from the uniform distribution, and continuing to train from the previous point.

*Training data and design of experiments.* To measure the performance of these methods, we present the average testing error and run statistics over 20 trials for both sampling strategies. We train our DNNs over a set of training data consisting of the values

$$\bigcup_{c=1}^{3} \{((X_{ic}, T_{ic}), u(X_{ic}, T_{ic}))\}_{i=1}^{m_c^{(l)}}.$$

For each $c = 1, 2, 3$, we generate each data set of size $m_c^{(l)}$, with $0 < m_c^{(1)} < \ldots < m_c^{(l)} < \ldots < m_c^{(9)}$, we sample 20 i.i.d. sets of points $\{\Theta_{ic}\}_{i=1}^{m_c^{(9)}}$ from the sampling measures for each strategy. The precise values for the numbers of samples $m_c^{(l)}$ are given in Table 2.

*Testing data and error metric.* Let $u^*$ be the solution of the PDE and $\hat{u}$ be a DNN approximation to it. We compute the (discrete) relative error

$$E(u^*) = \sqrt{\frac{\frac{1}{M} \sum_{z \in Z^t} |u^*(z) - \hat{u}(z)|^2}{\frac{1}{M} \sum_{z \in Z^t} |u^*(z)|^2}} = \sqrt{\frac{\sum_{z \in Z^t} |u^*(z) - \hat{u}(z)|^2}{\sum_{z \in Z^t} |u^*(z)|^2}}.$$

where $Z^t = Z_1^t \cup Z_2^t \cup Z_3^t$ and $M = M_1 + M_2 + M_3$. Here, for each $c = 1, 2, 3$, $Z_c^t$ is a test grid of size $M_c$, with points drawn randomly from the uniform measure over the corresponding domain. We set $M_1 = 8000$, $M_2 = 1000$ and $M_3 = 1000$ in this work.

To generate the training and testing data, we have to evaluate the true PDE solution $u$ over the corresponding grids. We compute this solution to an accuracy of $10^{-7}$ using the numerical solver *BURGERS_SOLUTION*, which is implemented in [14]. This is available from `https://people.math.sc.edu/Burkardt/m_src/burgers_solution/burgers_solution.html` under the GNU LGPL license.

*Plots.* As in Appendix B.5, our error plots display the geometric mean error value and a shaded region corresponding to one (geometric) standard deviation. We also plot the points generated by the final iteration of CAS. We visualize the density of these points using the package *densityScatterChart* in MATLAB using 'ksdensity' as density method and $\alpha \in [0.2, 0.5]$. See `https://www.mathworks.com/matlabcentral/fileexchange/95828-densityscatterchart?s_tid=FX_rc3_behav` for more details.

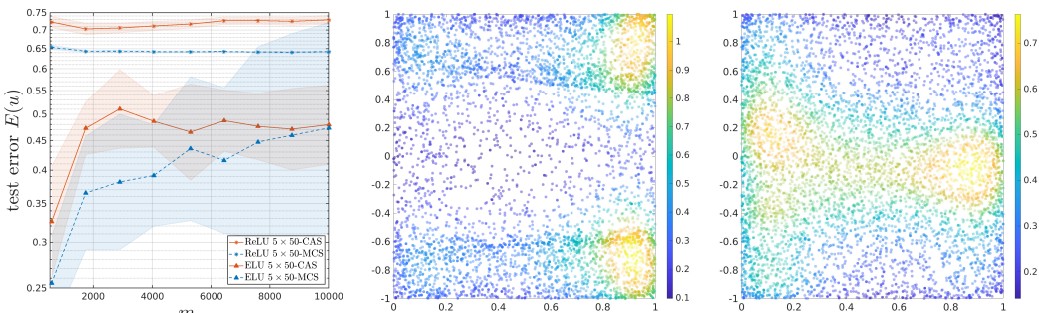

Figure 9: **CAS for solving PDEs with PINNs.** Plots of (left) the average test error $E(u^*)$ versus the number of samples used for training ReLU and ELU $5 \times 50$ DNNs using CAS (solid line) and MCS (dashed line), (middle) the samples generated by the CAS for the ReLU $5 \times 50$ DNN and (right) the samples generated by CAS for the ELU $5 \times 50$ DNN. The colour indicates the density of the points.

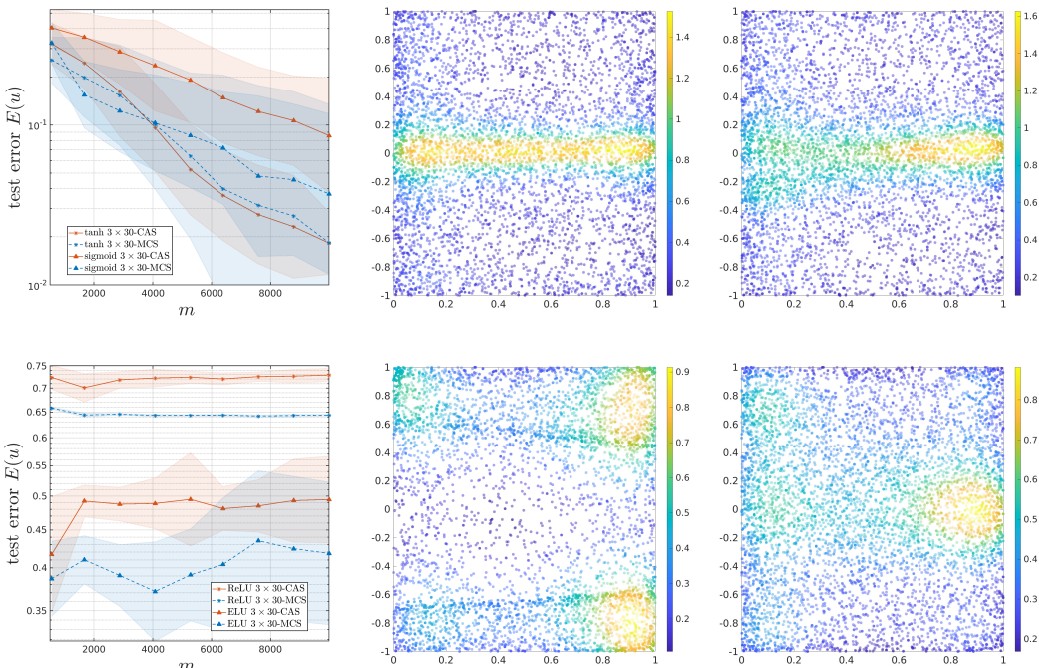

Figure 10: **CAS for solving PDEs with PINNs.** Plots of (left) the average test error $E(u^*)$ versus the number of samples used for training $3 \times 30$ DNNs using CAS (solid line) and MCS (dashed line), (middle) the samples generated by the CAS for the $\tanh$ (top) and ReLU (bottom) $3 \times 30$ DNNs and (right) the samples generated by CAS for the sigmoid (top) and ELU (bottom) $3 \times 30$ DNNs. The colour indicates the density of the points.

*Implementation, hardware and compute time.* Our experiments have been implemented in TensorFlow version 2.5. This implementation has been inspired by works [3, 19].

The computations were performed in single precision using NVDIA Tesla P100 GPUs on Compute Canada's Cedar compute cluster. To be precise, each node used in these experiments has 24 CPU cores and 128GB of memory and four Tesla P100 GPUs with 16GB memory. See `https://docs.alliancecan.ca/wiki/Cedar` for further information.

The compute time is roughly one day for the computations involving the $\rho$ $3 \times 30$ and $\rho$ $5 \times 50$ DNNs and 2 days for those involving the $\rho$ $10 \times 100$ DNNs. Therefore, the total compute time is approximately four days.

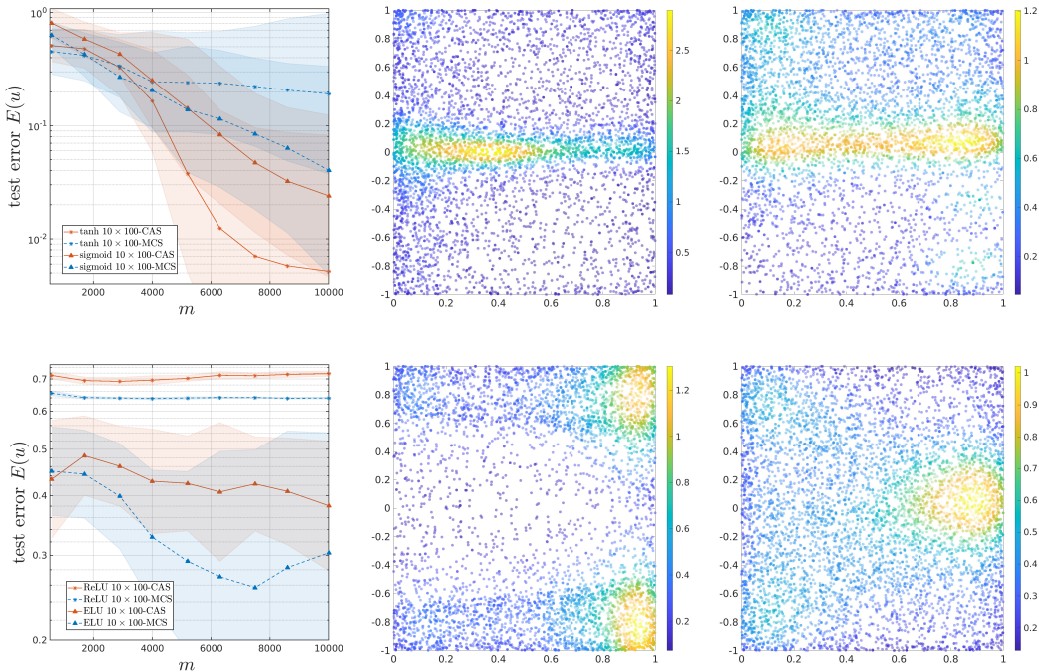

Figure 11: **CAS for solving PDEs with PINNs.** Plots of (left) the average test error $E(u^*)$ versus the number of samples used for training $10 \times 100$ DNNs using CAS (solid line) and MCS (dashed line), (middle) the samples generated by the CAS for the $\tanh$ (top) and ReLU (bottom) $10 \times 100$ DNNs and (right) the samples generated by CAS for the sigmoid (top) and ELU (bottom) $10 \times 100$ DNNs. The colour indicates the density of the points.

### D.6 Further experiments and discussion

In Fig. 9 we show the same results as in Fig. 3, except using the ReLU and ELU activation functions instead. Here, none of the methods lead to a decreasing test error as the number of samples increases, although the errors for MCS are somewhat larger than those of CAS. As is evident from the plots of the samples points, CAS fails to adapt the sampling measures to the PDE solution in this experiment, unlike in the case of Fig. 3.

This experiment indicates that the performance of CAS in relation to MCS is heavily influenced by the choice of architecture. To further explore this phenomenon, in Fig. 10 and Fig. 11 we consider $3 \times 30$ and $10 \times 100$ DNNs, respectively. In the former case, we once more see no decrease in the test error for the ReLU and ELU activation functions. The test error for the $\tanh$ and sigmoid activation functions does decrease with increasing $m$. Yet, CAS leads only to a marginal benefit over MCS for the $\tanh$ activation, and leads to worse performance for the sigmoid activation. As can be seen from the plot of the points, the CAS fails to adequately cluster the points in the case of the latter.

In Fig. 11, we consider $10 \times 100$ DNNs and see decreasing errors for the $\tanh$ and sigmoid activations, with CAS giving better performance than MCS. For the ELU activation function, while the error is (slowly) decreasing, CAS actually performs worse than MCS. Once more, for the ReLU activation function, we see no convergence.

The discrepancy in performance between the different activation functions is mirrored when we examine the training loss. In Fig. 12 we show the training loss versus number of epochs. For the best-performing $\tanh$ and sigmoid activations, we see significant decrease of the training loss, whereas for the other activations we see slower or sometimes no decrease. Note that all DNNs are trained using the same hyperparameters. This behaviour is in some senses unsurprising, since the worst-performing activation functions are only piecewise smooth and the training loss involves the PDE operator. Another notable feature is that MCS generally attains a smaller training error than CAS. Therefore, there is scope to further improve the performance of CAS by improving the training procedure.

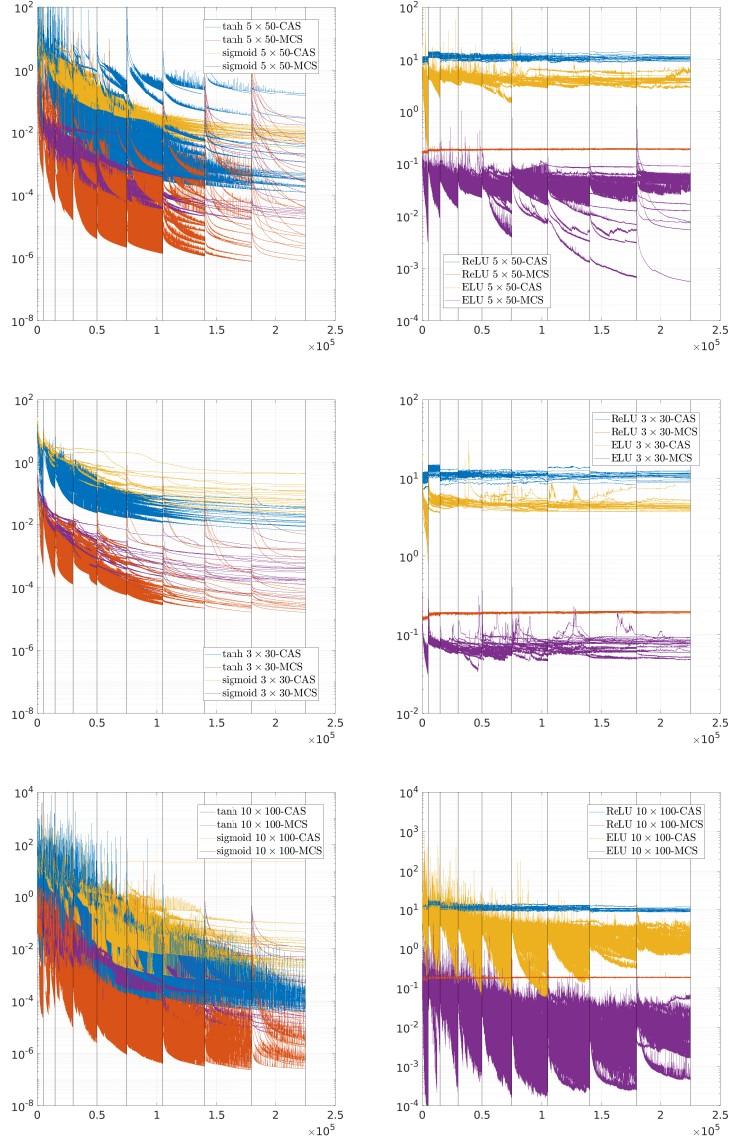

Figure 12: Plots of the training loss versus the number of epochs for the CAS and MCS methods with $5 \times 50$ DNNs using 20 trials. The vertical black lines indicate the number of epochs at which more samples are added in the training.

Finally, in Fig. 13 we plot the functions $\widetilde{K}_c$, $c = 1, 2, 3$, that are used to generate the sample points at the final iteration (see Appendix D.4). Here, once again we see that the $\tanh$ and sigmoid activation functions lead to functions $\widetilde{K}_c$ that are peaked according to the behaviour of the true PDE solution $u^*$. However, the ReLU and ELU activation functions fail to capture this behaviour.

# E   Proofs of the theoretical results

In this appendix, we give the proofs of the results from the main paper. We commence with the following lemma, which gives several properties of the generalized Christoffel function that will be needed later.

**Lemma E.1** (Properties of $K(\mathbb{F})(\theta)$). *The generalized Christoffel function $K$ (see Definition 2.6) has the following properties.*

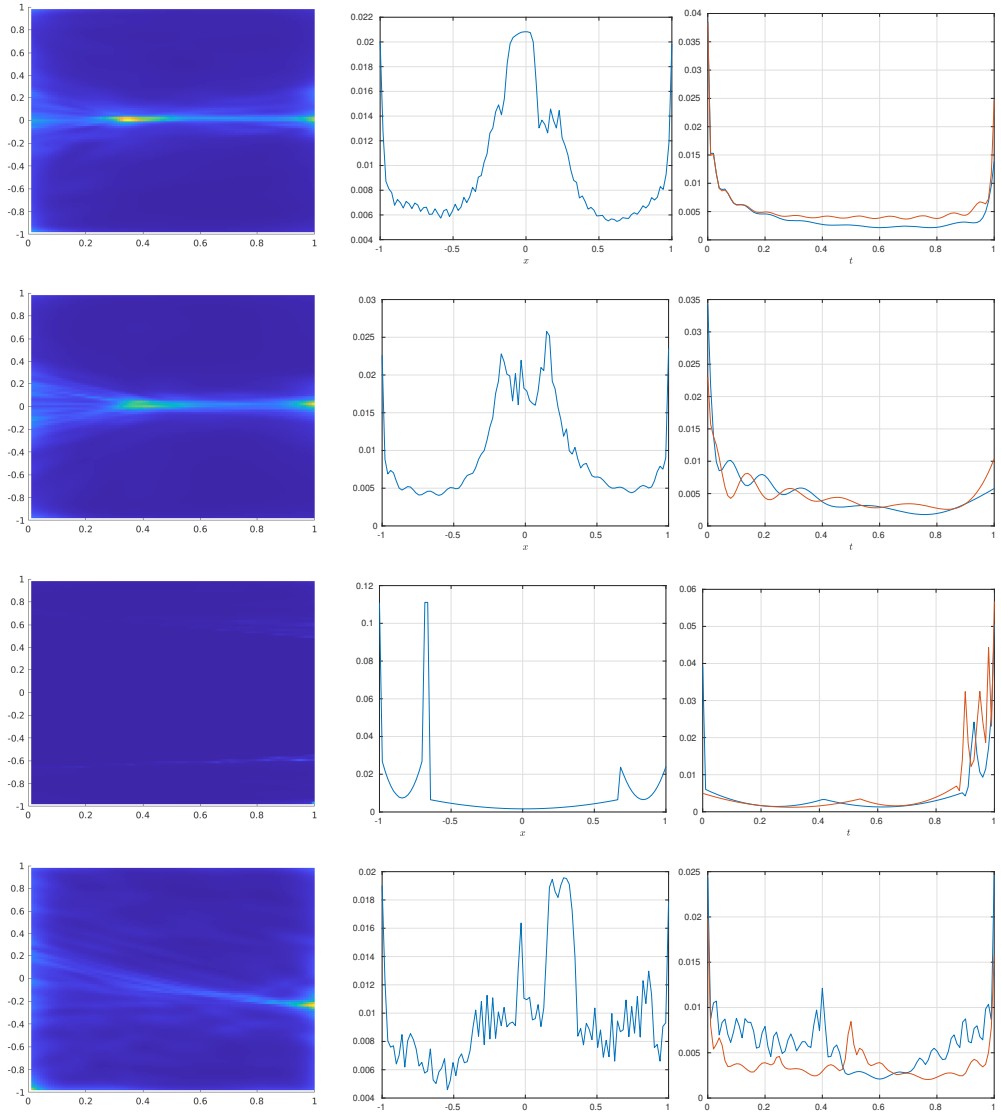

Figure 13: Plots of the functions $\widetilde{K}_c$, $c = 1, 2, 3$ (left to right), that are used the generate the sample points in the final iteration of CAS in the case of the $5 \times 50$ DNNs with tanh, sigmoid, ReLU and ELU activation functions (top to bottom). In the third column, the blue and red curves correspond to the $x = 1$ and $x = -1$ boundaries, respectively.

(i) Let $\mathbb{F}$ be a finite-dimensional subspace of $\mathbb{X}$. Then

$$K(\mathbb{F})(\theta) \leq \sum_{i=1}^{n} \|L(\theta)(f_i)\|_{\mathbb{Y}}^2 \tag{E.1}$$

for any orthonormal basis $\{f_i\}_{i=1}^n$ of $\mathbb{F}$. Furthermore, if $\mathbb{Y}$ has dimension $p$ then

$$K(\mathbb{F})(\theta) \geq \frac{1}{p} \sum_{i=1}^{n} \|L(\theta)(f_i)\|_{\mathbb{Y}}^2.$$

In particular, (E.1) holds with equality when $\dim(\mathbb{Y}) = 1$.

(ii) Let $\mathbb{F}_1, \mathbb{F}_2 \subseteq \mathbb{X}$. Then

$$K(\mathbb{F}_1 \cup \mathbb{F}_2)(\theta) = \max\left\{K(\mathbb{F}_1)(\theta), K(\mathbb{F}_2)(\theta)\right\}.$$

*(iii) Let $\mathbb{F}_1, \mathbb{F}_2$ be finite-dimensional subspaces of $\mathbb{X}$. Then*

$$K(\mathbb{F}_1 + \mathbb{F}_2)(\theta) \leq 2\left(K(\mathbb{F}_1)(\theta) + K(\mathbb{F}_2)(\theta)\right).$$

*(iv) Let $\mathbb{F} \subseteq \mathbb{X}$ and $\mathcal{B}(\mathbb{F}) = \{f/\|f\|_{\mathbb{X}} : f \in \mathbb{F}\backslash\{0\}\}$. Then*

$$K(\mathcal{B}(\mathbb{F}))(\theta) = K(\mathbb{F})(\theta).$$

*Proof.* We commence with the proof of (i). Write $f \in \mathbb{F}$, $f \neq 0$, as $f = \sum_{i=1}^{n} c_i f_i$ for $c_i \in \mathbb{C}$ with $\sum_{i=1}^{n} |c_i|^2 = \|f\|_{\mathbb{X}}^2$. Then

$$\|L(\theta)(f)\|_{\mathbb{Y}}^2 = \left\|\sum_{i=1}^{n} c_i L(\theta)(f_i)\right\|_{\mathbb{Y}}^2 \leq \sum_{i=1}^{n} |c_i|^2 \sum_{i=1}^{n} \|L(\theta)(f_i)\|_{\mathbb{Y}}^2 = \|f\|_{\mathbb{X}}^2 \sum_{i=1}^{n} \|L(\theta)(f_i)\|_{\mathbb{Y}}^2,$$

and therefore

$$K(\mathbb{F})(\theta) \leq \sum_{i=1}^{n} \|L(\theta)(f_i)\|_{\mathbb{Y}}^2.$$

For the lower bound, let $\{e_1, \ldots, e_p\}$ be an orthonormal basis of $\mathbb{Y}$. Fix $j \in \{1, \ldots, p\}$ and consider $f = \sum_{i=1}^{n} c_i f_i$, where $c_i = \langle e_j, L(\theta)(f_i)\rangle_{\mathbb{Y}}$. Then

$$\|L(\theta)(f)\|_{\mathbb{Y}}^2 = \left\|\sum_{i=1}^{n} \langle e_j, L(\theta)(f_i)\rangle_{\mathbb{Y}} L(\theta)(f_i)\right\|_{\mathbb{Y}}^2 \geq \left(\sum_{i=1}^{n} |\langle e_j, L(\theta)(f_i)\rangle_{\mathbb{Y}}|^2\right)^2.$$

We also have

$$\|f\|_{\mathbb{X}}^2 = \sum_{i=1}^{n} |\langle e_j, L(\theta)(f_i)\rangle_{\mathbb{Y}}|^2.$$

Therefore, we deduce that

$$K(\mathbb{F})(\theta) \geq \sum_{i=1}^{n} |\langle e_j, L(\theta)(f_i)\rangle_{\mathbb{Y}}|^2.$$

Since $j$ was arbitrary, we now sum over $j$ and use Parseval's identity to obtain

$$pK(\mathbb{F})(\theta) \geq \sum_{i=1}^{n} \sum_{j=1}^{p} |\langle e_j, L(\theta)(f_i)\rangle_{\mathbb{Y}}|^2 = \sum_{i=1}^{n} \|L(\theta)(f_i)\|_{\mathbb{Y}}^2,$$

as required.

Part (ii) follows immediately from the definition of $K$. Now consider (iii). Write $\mathbb{F} = \mathbb{F}_1 + \mathbb{F}_2 = \mathbb{G}_1 \oplus \mathbb{G}_2$, where $\mathbb{G}_1 = \mathbb{F}_1$ and $\mathbb{G}_2 = \mathbb{F}_2 \cap \mathbb{F}_1^{\perp}$ satisfies $\mathbb{G}_2 \perp \mathbb{G}_1$. Let $f \in \mathbb{F}$, $f \neq 0$, be arbitrary and write $f = g_1 + g_2$ for $g_i \in \mathbb{G}_i$, $i = 1, 2$. Then

$$\begin{aligned}
\frac{\|L(\theta)(f)\|_{\mathbb{Y}}^2}{\|f\|_{\mathbb{X}}^2} &\leq 2\frac{\|L(\theta)(g_1)\|_{\mathbb{Y}}^2 + \|L(\theta)(g_2)\|_{\mathbb{Y}}^2}{\|g_1\|_{\mathbb{X}}^2 + \|g_2\|_{\mathbb{X}}^2} \\
&\leq 2\left(\frac{\|L(\theta)(g_1)\|_{\mathbb{Y}}^2}{\|g_1\|_{\mathbb{X}}^2} + \frac{\|L(\theta)(g_2)\|_{\mathbb{Y}}^2}{\|g_2\|_{\mathbb{X}}^2}\right) \\
&\leq 2\left(K(\mathbb{G}_1)(\theta) + K(\mathbb{G}_2)(\theta)\right) \\
&\leq 2\left(K(\mathbb{F}_1)(\theta) + K(\mathbb{F}_2)(\theta)\right),
\end{aligned}$$

where in the last step we used the fact that $\mathbb{G}_2 \subseteq \mathbb{F}_2$. Since $f$ was arbitrary, the result now follows.

Finally, consider (iv). By the definition of $K$, $\mathcal{B}(\mathbb{F})$ and linearity,

$$\begin{aligned}
K(\mathcal{B}(\mathbb{F}))(\theta) &= \sup\left\{\|L(\theta)(f/\|f\|_{\mathbb{X}})\|_{\mathbb{Y}}^2 : f \in \mathbb{F}\backslash\{0\}\right\} \\
&= \sup\left\{\frac{\|L(\theta)(f)\|_{\mathbb{Y}}^2}{\|f\|_{\mathbb{X}}^2} : f \in \mathbb{F}\backslash\{0\}\right\} = K(\mathbb{F})(\theta),
\end{aligned}$$

as required. $\qquad\square$

For the remaining results it is useful to introduce some additional notation. Let

$$\overline{\mathbb{Y}} = \oplus_{c=1}^{C} \oplus_{i=1}^{m_c} \mathbb{Y}_c$$

be the *overall measurement space*. Note that this is a Hilbert space, when equipped with suitable norm. If $y = (y_{ic})$ is an arbitrary element of $\mathbb{Y}$, we take this norm to be

$$\|y\|_{\overline{\mathbb{Y}}} = \sqrt{\sum_{c=1}^{C} \sum_{i=1}^{m_c} \|y_{ic}\|_{\mathbb{Y}_c}^2}.$$

Now let $m = m_1 + \ldots + m_C$, $D = \otimes_{c=1}^{C} \otimes_{i=1}^{m_c} D_c$ be the tensor-product space and write $(\theta_{ic})$ for an arbitrary element of $D$, where $\theta_{ic} \in D_c$ for $i = 1, \ldots, m_c$ and $c = 1, \ldots, C$. Then we define the *overall sampling operator*

$$M : D \to \mathcal{B}(\mathbb{X}_0, \overline{\mathbb{Y}}), \ \theta := (\theta_{ic}) \mapsto \left( \frac{1}{\sqrt{\nu_c(\theta_{ic})m_c}} L_c(\theta_{ic}) \right). \tag{E.2}$$

Note that the operator on the right-hand side is given by

$$f \in \mathbb{X}_0 \mapsto \left( \frac{1}{\sqrt{\nu_c(\theta_{ic})m_c}} L_c(\theta_{ic})(f) \right) \in \overline{\mathbb{Y}}.$$

Further, we note that the nondegeneracy condition is equivalent to

$$\alpha\|f\|_{\mathbb{X}}^2 \le \mathbb{E}_{\overline{\Theta} \sim \tau} \|M(\overline{\Theta})(f)\|_{\mathbb{Y}}^2 \le \beta\|f\|_{\mathbb{X}}^2, \quad \forall f \in \mathbb{X}_0. \tag{E.3}$$

Here $\tau = \otimes_{c=1}^{C} \otimes_{i=1}^{m_c} \mu_c$ is the probability measure of the full sample $(\Theta_{ic}) := \overline{\Theta}$.

Next, given measurements as in (2.1) or (4.4), we define the scaled measurements $\bar{y} = (\bar{y}_{ic}) \in \mathbb{Y}$ by

$$\bar{y}_{ic} = \frac{1}{\sqrt{\nu_c(\theta_{ic})m_c}} y_{ic}, \quad i = 1, \ldots, m_c, \ c = 1, \ldots, C. \tag{E.4}$$

Then, with $M$ as defined in E.2, the regression problem (2.2) is equivalent to

$$\hat{f} \in \underset{f \in \mathbb{F}}{\operatorname{argmin}} \|\bar{y} - M(\Theta)(f)\|_{\overline{\mathbb{Y}}}^2. \tag{E.5}$$

With this in hand, we now prove Theorem 4.2. For this, it is first useful to prove a special case of Theorem 4.2 when $\mathbb{F}$ is itself a subspace.

**Lemma E.2** (Special case of Theorem 4.2 for subspaces)**.** *Consider the setup of Theorem 4.2, where $\mathbb{F}$ is a subspace of dimension at most $n$. Suppose that*

$$m_c \ge c_\gamma \cdot \alpha^{-1} \cdot \operatorname*{ess\,sup}_{\theta \sim \rho_c} \{K_c(\mathbb{F})(\theta)/\nu_c(\theta)\} \cdot \log(2n/\delta), \quad \forall c = 1, \ldots, C,$$

*for some $0 < \gamma < 1$, where*

$$c_\gamma = ((1+\gamma)\log(1+\gamma) - \gamma)^{-1}. \tag{E.6}$$

*Then* (2.5) *holds with probability at least* $1 - \delta$.

*Proof.* The proof of the result in this case is a straightforward application of the matrix Chernoff bound [115]. Let $\{f_i\}_{i=1}^{n}$ be an orthonormal basis for $\mathbb{F}$. Write an arbitrary $f \in \mathbb{F}$ as $f = \sum_{i=1}^{n} c_i f_i$, so that $\|f\|_{\mathbb{X}} = \|c\|_{\ell^2}$, where $c = (c_i)_{i=1}^{n} \in \mathbb{C}^n$. Then

$$\begin{aligned}
\|M(\Theta)(f)\|_{\overline{\mathbb{Y}}}^2 &= \sum_{c=1}^{C} \frac{1}{m_c} \sum_{i=1}^{m_c} \frac{1}{\nu_c(\Theta_{ic})} \|L_c(\Theta_{ic})(f)\|_{\mathbb{Y}_c}^2 \\
&= \sum_{c=1}^{C} \frac{1}{m_c} \sum_{i=1}^{m_c} \frac{1}{\nu_c(\Theta_{ic})} \sum_{j,k=1}^{n} \overline{c_j} c_k \langle L_c(\Theta_{ic})(f_j), L_c(\Theta_{ic})(f_k) \rangle_{\mathbb{Y}_c} \\
&= c^* \left( \sum_{c=1}^{C} \sum_{i=1}^{m_c} X_{ic} \right) c,
\end{aligned}$$

where $X_{ic}$ are the independent, self-adjoint, $n \times n$ random matrices with

$$X_{ic} = \left( \frac{1}{m_c \nu_c(\Theta_{ic})} \langle L_c(\Theta_{ic})(f_j), L_c(\Theta_{ic})(f_k) \rangle_{\mathbb{Y}_c} \right)_{j,k=1}^d .$$

Thus, the result is equivalent to the condition

$$(1 - \epsilon)\alpha \leq \lambda_{\min} \left( \sum_{c=1}^C \sum_{i=1}^{m_c} X_{ic} \right) \leq \lambda_{\max} \left( \sum_{c=1}^C \sum_{i=1}^{m_c} X_{ic} \right) \leq (1 + \epsilon)\beta.$$

Notice that $X_{ic} \succeq 0$ and, due to (E.3),

$$\alpha \leq \lambda_{\min} \left( \sum_{c=1}^C \sum_{i=1}^{m_c} \mathbb{E}X_{ic} \right) \leq \lambda_{\max} \left( \sum_{c=1}^C \sum_{i=1}^{m_c} \mathbb{E}X_{ic} \right) \leq \beta.$$

Now, with $f = \sum_{i=1}^n c_i f_i \in \mathbb{F}$, we have

$$c^* X_{ic} c = \frac{1}{m_c \nu_c(\Theta_{ic})} \| L_c(\Theta_{ic})(f) \|_{\mathbb{Y}_c}^2 \leq \frac{1}{m_c \nu_c(\Theta_{ic})} K_c(\mathbb{F})(\Theta_{ic}) \| f \|_{\mathbb{X}}^2,$$

Thus,

$$\lambda_{\max}(X_{ic}) \leq m_c^{-1} \operatorname*{ess\,sup}_{\theta \sim \rho_c} \{ K_c(\mathbb{F})(\theta) / \nu_c(\theta) \},$$

almost surely. The result now follows from the matrix Chernoff bound. $\qquad \square$

*Proof of Theorem 4.2.* Let $f \in \mathcal{B}(\mathbb{F} - \mathbb{F})$. Then there is $j \in \{1, \dots, d\}$ and $f_j \in \mathbb{F}_j \cap \mathcal{B}(\mathbb{F} - \mathbb{F})$ such that $\|\|f - f_j\|\| \leq t$, where $t = \sqrt{\alpha \epsilon / 2}$. Observe that

$$\| M(\overline{\Theta})(f_j) \|_{\overline{\mathbb{Y}}} + \| M(\overline{\Theta})(f - f_j) \|_{\overline{\mathbb{Y}}} \geq \| M(\overline{\Theta})(f) \|_{\overline{\mathbb{Y}}} \geq \| M(\overline{\Theta})(f_j) \|_{\overline{\mathbb{Y}}} - \| M(\overline{\Theta})(f - f_j) \|_{\overline{\mathbb{Y}}}.$$

Now, with probability one,

$$\| M(\overline{\Theta})(f - f_j) \|_{\overline{\mathbb{Y}}}^2 = \sum_{c=1}^C \frac{1}{m_c} \sum_{i=1}^{m_c} \frac{1}{\nu_c(\Theta_{ic})} \| L_c(\Theta_{ic})(f - f_j) \|_{\mathbb{Y}_c}^2 \leq \|\|f - f_j\|\|^2 \leq \alpha \epsilon / 2.$$

Using this and the fact that $\beta \geq \alpha$, we deduce that

$$\| M(\overline{\Theta})(f_j) \|_{\overline{\mathbb{Y}}} + \beta \epsilon / 2 \geq \| M(\overline{\Theta})(f) \|_{\overline{\mathbb{Y}}} \geq \| M(\overline{\Theta})(f_j) \|_{\overline{\mathbb{Y}}} - \alpha \epsilon / 2.$$

Now let $E$ be the event

$$(1 - \epsilon)\alpha \| f \|_{\mathbb{X}}^2 \leq \| M(\overline{\Theta})(f) \|_{\overline{\mathbb{Y}}}^2 \leq (1 + \epsilon)\beta \| f \|_{\mathbb{X}}^2, \quad \forall f \in \mathbb{F} - \mathbb{F},$$

and note that our goal is to show that $\mathbb{P}(E) \geq 1 - \delta$. By linearity, $E$ is equivalent to

$$(1 - \epsilon)\alpha \leq \| M(\overline{\Theta})(f) \|_{\overline{\mathbb{Y}}}^2 \leq (1 + \epsilon)\beta, \quad \forall f \in \mathcal{B}(\mathbb{F} - \mathbb{F}).$$

Hence, by the above argument, we have

$$\mathbb{P}(E) \geq \mathbb{P} \left( \bigcap_{j=1}^d E_j \right),$$

where $E_j$ is the event

$$(1 - \epsilon/2)\alpha \| f_j \|_{\mathbb{X}}^2 \leq \| M(\overline{\Theta})(f_j) \|_{\overline{\mathbb{Y}}}^2 \leq (1 + \epsilon/2)\beta \| f_j \|_{\mathbb{X}}^2, \quad \forall f_j \in \mathbb{F}_j.$$

By the union bound

$$\mathbb{P}(E^c) \leq \sum_{j=1}^d \mathbb{P}(E_j^c).$$

Suppose first that $n \geq 1$. Then $K(\mathbb{F}_j)(\theta) \leq K(\widetilde{\mathbb{F}})(\theta)$ and the condition (4.1) on the $m_c$ and Lemma E.2 give that $\mathbb{P}(E_j^c) \leq \delta/d$.

The gives the result for $n \geq 1$.

For $n = 0$, we have that $\mathbb{F}_j = \{f_i\}$ is a singleton. Notice that $\mathbb{F}_j \subseteq \operatorname{span}(\mathbb{F}_j)$ and $K_c(\mathbb{F}_j)(\theta) = K_c(\operatorname{span}(\mathbb{F}_j))(\theta)$. Hence we may apply Lemma E.2 with $n = 1$. We deduce once more that $\mathbb{P}(E_j^c) \leq \delta/d$. This gives the result for $n = 0$ as well. $\qquad \square$

**Remark E.3 (The norm in Theorem 4.2)** Write $\mathbb{F}' = \mathcal{B}(\mathbb{F} - \mathbb{F})$. Then it follows from the definition of the $K_c$ that

$$\||f_1 - f_2|\| \leq \sqrt{\sum_{c=1}^{C} \mathrm{esssup}_{\theta \sim \rho_c} K_c(\mathbb{F}' - \mathbb{F}')/\nu_c(\theta_c)} \|f_1 - f_2\|_{\mathbb{X}}, \quad \forall f_1, f_2 \in \mathbb{F}'.$$

Therefore, the cover in Theorem 4.2 could be constructed with respect to the $\mathbb{X}$-norm, up to a change in the constant $t$. Notice that the quantity that bounds $\||f_1 - f_2|\|$ in terms of $\|f_1 - f_2\|_{\mathbb{X}}$ involves the generalized Christoffel functions $K_c(\mathbb{F}' - \mathbb{F}')$ and weight functions $\nu_c$. Therefore, sampling according to the generalized Christoffel function (on a slightly modified set) also minimizes this constant, and therefore the size $d$ of the cover in Theorem 4.2.

*Proof of Corollary 4.3.* If $\mathbb{F} - \mathbb{F} = \cup_{i=1}^d \mathbb{F}_i$ is a union of $d$ subspaces of dimension at most $n$, then clearly

$$\mathcal{B}(\mathbb{F} - \mathbb{F}) = \cup_{i=1}^d \mathcal{B}(\mathbb{F}_i) \subset \cup_{i=1}^d \mathbb{F}_i.$$

It follows that $\{\mathbb{F}_i\}_{i=1}^d$ provides a $(\|\cdot\|, n, t)$-subspace covering of $\mathcal{B}(\mathbb{F} - \mathbb{F})$ for any $t \geq 0$. Hence we may apply Theorem 4.2 with this cover. Since $\widetilde{\mathbb{F}} = \cup_{i=1}^d \mathbb{F}_i = \mathbb{F} - \mathbb{F}$ in this case, the result now follows immediately. $\square$

*Proof of Corollary 4.4.* When $n = 0$, the sets $\mathbb{F}_j = \{f_j\}$ are singletons and we must have $\mathbb{F}_j \subseteq \mathcal{B}(\mathbb{F} - \mathbb{F})$ by definition of the covering. Hence $\widetilde{\mathbb{F}} \subseteq \mathcal{B}(\mathbb{F} - \mathbb{F})$, which gives that $K_c(\widetilde{\mathbb{F}})(\theta) \leq K_c(\mathcal{B}(\mathbb{F} - \mathbb{F}))(\theta)$. Using Lemma E.1, we deduce that $K_c(\widetilde{\mathbb{F}})(\theta) \leq K_c(\mathbb{F} - \mathbb{F})(\theta)$. This implies the result. $\square$

*Proof of Lemma 4.6.* Since $K(\mathbb{F})$ is integrable, the quantity $\kappa(\mathbb{F})$ is well defined. Also, $K(\mathbb{F})(\theta) > 0$ almost everywhere, due to the assumption on $L$ and $\mathbb{F}$. In particular, $\kappa(\mathbb{F}) > 0$. It follows that $\nu^\star$ is measurable, positive almost everywhere and integrable with $\int_D \nu^\star(\theta) \, \mathrm{d}\rho(\theta) = 1$. Clearly,

$$\mathrm{ess} \sup_{\theta \sim \rho} \{K(\theta)/\nu^\star(\theta)\} = \kappa(\mathbb{F}).$$

We now show that this is the optimal value. Let $\nu$ be any other such function. Then

$$\mathrm{ess} \sup_{\theta \sim \rho} \{K(\mathbb{F})(\theta)/\nu(\theta)\} \geq \int_D \frac{K(\mathbb{F})(\theta)}{\nu(\theta)} \nu(\theta) \, \mathrm{d}\rho(\theta) = \int_D K(\mathbb{F})(\theta) \, \mathrm{d}\rho(\theta) = \kappa(\mathbb{F}).$$

This gives the result. $\square$

*Proof of Corollary 4.7.* We first prove the result when $d = 1$. Let $\{f_i\}_{i=1}^n$ be an orthonormal basis for $\mathbb{F}_1 = \mathbb{F} - \mathbb{F}$. Then Lemma E.1 gives that

$$K_c(\mathbb{F} - \mathbb{F})(\theta) \leq \sum_{i=1}^n \|L_c(\theta)(f_i)\|_{\mathbb{Y}_c}^2.$$

Therefore,

$$\sum_{c=1}^C \kappa_c(\mathbb{F} - \mathbb{F}) = \sum_{c=1}^C \int_{D_c} K_c(\mathbb{F} - \mathbb{F})(\theta) \, \mathrm{d}\rho_c(\theta) \leq \sum_{i=1}^n \sum_{c=1}^C \int_{D_c} \|L_c(\theta)(f_i)\|_{\mathbb{Y}_c}^2 \, \mathrm{d}\rho_c(\theta).$$

Assumption 2.4 and orthonormality now imply that

$$\sum_{c=1}^C \kappa_c(\mathbb{F} - \mathbb{F}) \leq \beta \sum_{i=1}^n \|f_i\|_{\mathbb{X}}^2 \leq \beta n.$$

Now suppose that $d > 1$. Lemma E.1 gives that

$$K_c(\mathbb{F} - \mathbb{F})(\theta) = \max_{j=1,\ldots,d} K_c(\mathbb{F}_j)(\theta)$$

and therefore

$$K_c(\mathbb{F} - \mathbb{F})(\theta) \leq \sum_{j=1}^{d} K_c(\mathbb{F}_j)(\theta).$$

Using the result for $d = 1$, we get

$$\sum_{c=1}^{C} \kappa_c(\mathbb{F} - \mathbb{F}) \leq \beta n d,$$

as required. $\qquad\qquad\square$

**Remark E.4 (Lower bound on $\sum_{c=1}^{C} \kappa_c(\mathbb{F} - \mathbb{F})$)** Suppose that $\dim(\mathbb{Y}_c) \leq p \in \mathbb{N}$, $\forall c$. Then Lemma E.1 gives, for a subspace $\mathbb{G}$ with an orthonormal basis $\{g_i\}_{i=1}^{n}$,

$$K_c(\mathbb{G})(\theta) \geq \frac{1}{p} \sum_{i=1}^{n} \|L_c(\theta)(g_i)\|_{\mathbb{Y}_c}^2.$$

Using this and following the steps of the above proof, we deduce the lower bound

$$\sum_{c=1}^{C} \kappa_c(\mathbb{F} - \mathbb{F}) \geq \alpha n / p.$$

Finally, we prove Theorem 4.8. For this, we first require the following lemma.

**Lemma E.5.** *Suppose that a realization $\theta = (\theta_{ic}) \in D$ of the $\Theta_{ic}$ gives that*

$$\alpha' \|f\|_{\mathbb{X}}^2 \leq \|M(\theta)(f)\|_{\overline{\mathbb{Y}}}^2, \quad \forall f \in \mathbb{F} - \mathbb{F}, \tag{E.7}$$

*for some $\alpha' > 0$. Then any $\gamma$-minimizer $\hat{f}$ of (2.2) satisfies*

$$\|\hat{f} - f^*\|_{\mathbb{X}} \leq \inf_{f \in \mathbb{F}} \left\{ \|f^* - f\|_{\mathbb{X}} + \frac{2}{\sqrt{\alpha'}} \|M(\theta)(f^* - f)\|_{\overline{\mathbb{Y}}} \right\} + \frac{2}{\sqrt{\alpha'}} \|\bar{e}\|_{\overline{\mathbb{Y}}} + \frac{\gamma}{\sqrt{\alpha'}},$$

*where $\bar{e} = \left( \frac{1}{\sqrt{\nu_c(\theta_{ic}) m_c}} e_{ic} \right)$.*

*Proof.* Let $f \in \mathbb{F}$. We write

$$
\begin{aligned}
\|\hat{f} - f^*\|_{\mathbb{X}} &\leq \|\hat{f} - f\|_{\mathbb{X}} + \|f^* - f\|_{\mathbb{X}} \\
&\leq 1/\sqrt{\alpha'} \|M(\theta)(\hat{f} - f)\|_{\overline{\mathbb{Y}}} + \|f^* - f\|_{\mathbb{X}} \\
&\leq 1/\sqrt{\alpha'} \|M(\theta)(\hat{f}) - \bar{y}\|_{\overline{\mathbb{Y}}} + 1/\sqrt{\alpha'} \|\bar{y} - M(\theta)(f)\|_{\overline{\mathbb{Y}}} + \|f^* - f\|_{\mathbb{X}}.
\end{aligned}
$$

We now use the fact that $\hat{f}$ is a $\gamma$-minimizer and the fact that $\bar{y} = M(\theta)(f^*) + \bar{e}$ to get

$$
\begin{aligned}
\|\hat{f} - f^*\|_{\mathbb{X}} &\leq 2/\sqrt{\alpha'} \|M(\theta)(f) - \bar{y}\|_{\overline{\mathbb{Y}}} + \|f^* - f\|_{\mathbb{X}} + \gamma/\sqrt{\alpha'} \\
&\leq 2/\sqrt{\alpha'} \|M(\theta)(f^* - f)\|_{\overline{\mathbb{Y}}} + 2/\sqrt{\alpha'} \|\bar{e}\|_{\overline{\mathbb{Y}}} + \|f^* - f\|_{\mathbb{X}} + \gamma/\sqrt{\alpha'},
\end{aligned}
$$

as required. $\qquad\qquad\square$

*Proof of Theorem 4.8.* Let $E$ be the event

$$(1 - \epsilon)\alpha \|f\|_{\mathbb{X}}^2 \leq \|M(\overline{\Theta})(f)\|_{\overline{\mathbb{Y}}}^2 \leq (1 + \epsilon)\beta \|f\|_{\mathbb{X}}^2, \quad \forall f \in \mathbb{F} - \mathbb{F}.$$

Due Theorem 4.2 and the given assumptions, we have that $\mathbb{P}(E^c) \leq \delta$. Now recall that $\tau$ is the probability distribution of the full draw $\overline{\Theta}$. Then

$$\mathbb{E}\|f^* - \check{f}\|_{\mathbb{X}}^2 = \int_E \|f^* - \check{f}\|_{\mathbb{X}}^2 \, d\tau + \int_{E^c} \|f^* - \check{f}\|_{\mathbb{X}}^2 \, d\tau.$$

In the first integral, we use the fact that $\mathcal{T}_\sigma$ is a contraction to get

$$\|f^* - \check{f}\|_{\mathbb{X}} = \|\mathcal{T}_\sigma(f) - \mathcal{T}_\sigma(\hat{f})\|_{\mathbb{X}} \leq \|f^* - \hat{f}\|_{\mathbb{X}}.$$

In the second integral, we use the fact that $\|\mathcal{T}_\sigma(g)\|_{\mathbb{X}} \leq \sigma$ to get

$$\|f^* - \check{f}\|_{\mathbb{X}} \leq 2\sigma.$$

Therefore, we obtain

$$\mathbb{E}\|f^* - \check{f}\|_{\mathbb{X}}^2 = \int_E \|f^* - \hat{f}\|_{\mathbb{X}}^2 \, d\tau + 4\sigma^2\delta. \tag{E.8}$$

Now, for the first term, we fix $f \in \mathbb{F}$ and apply Lemma E.5. Since $\alpha' \geq (1-\epsilon)\alpha$ whenever $E$ holds, we get

$$\int_E \|f^* - \hat{f}\|_{\mathbb{X}}^2 \, d\tau$$

$$\leq 4 \int_E \left( \|f^* - f\|_{\mathbb{X}}^2 + \frac{4}{(1-\epsilon)\alpha} \|M(\overline{\Theta})(f^* - f)\|_{\overline{\mathbb{Y}}}^2 + \frac{4}{(1-\epsilon)\alpha} \|\bar{e}\|_{\overline{\mathbb{Y}}}^2 + \frac{\gamma^2}{(1-\epsilon)\alpha} \right) \, d\tau.$$

For the second term, we use Assumption 2.4 and (E.3) to get

$$\int_E \|f^* - \hat{f}\|_{\mathbb{X}}^2 \, d\tau \leq 4 \left( 1 + \frac{4\beta}{(1-\epsilon)\alpha} \right) \|f^* - f\|_{\mathbb{X}}^2$$

$$+ \frac{16}{(1-\epsilon)\alpha} \sum_{c=1}^C \frac{1}{m_c} \sum_{i=1}^{m_c} \|e_{ic}\|_{\mathbb{Y}_c}^2 \int_{D_c} \frac{1}{\nu_c(\theta_c)} \, d\mu_c(\theta_c) + \frac{4\gamma^2}{(1-\epsilon)\alpha}$$

and therefore

$$\int_E \|f^* - \hat{f}\|_{\mathbb{X}}^2 \, d\tau \leq 4 \left( 1 + \frac{4\beta}{(1-\epsilon)\alpha} \right) \|f^* - f\|_{\mathbb{X}}^2$$

$$+ \frac{16}{(1-\epsilon)\alpha} \sum_{c=1}^C \frac{1}{m_c} \sum_{i=1}^{m_c} \|e_{ic}\|_{\mathbb{Y}_c}^2 \rho_c(D_c) + \frac{4\gamma^2}{(1-\epsilon)\alpha}.$$

Combining this with (E.8) and recalling that $f$ was arbitrary now gives the result. $\qquad\square$

**Remark E.6** As noted, the noise bound in Theorem 4.8 assumes $\rho_c(D_c) < \infty$. If this fails, one may also bound the noise term by assuming the noise takes the form

$$e_{ic} = L_c(\Theta_{ic})(g), \quad i = 1, \ldots, m_c, \ c = 1, \ldots, C,$$

for some $g \in \mathbb{X}$ with $\|g\|_{\mathbb{X}} \ll 1$. In this case, one has

$$\mathbb{E}\|\bar{e}\|_{\overline{\mathbb{Y}}}^2 = \sum_{c=1}^C \frac{1}{m_c} \sum_{i=1}^{m_c} \mathbb{E}\left( \frac{1}{\nu_c(\Theta_{ic})} \|L_c(\Theta_{ic})(g)\|_{\mathbb{Y}_c}^2 \right) = \sum_{c=1}^C \int_{D_c} \|L(\theta)(g)\|_{\mathbb{Y}_c}^2 \, d\rho_c(\theta) \leq \beta \|g\|_{\mathbb{X}}^2.$$

Hence Theorem 4.8 holds with $N = \beta\|g\|_{\mathbb{X}}^2$. For other ways to estimate the noise term under different noise models, see, e.g., [93].

*Proof of Theorem 2.7.* Part (i) follows from Theorem 4.2, Corollary 4.3 and Lemma 4.6 with the (arbitrary) value $\epsilon = 1/2$. Part (ii) follows from part (i) and Theorem 4.8. Finally, part (iii) follows from Corollary 4.7. $\qquad\square$

