# OpenReview forum: "CS4ML: A general framework for active learning with arbitrary data based on Christoffel functions"
_NeurIPS.cc/2023/Conference — NeurIPS 2023 spotlight_

### Official Review · Reviewer_n7M7 · 2023-07-07

**Soundness:** 3 good
**Presentation:** 3 good
**Contribution:** 4 excellent
**Rating:** 8
**Confidence:** 3

**Summary:**

This paper studies an extremely general notion of active learning in the $\ell_2$ norm. Suppose we are able to actively observe data in many different modalities. "Active Learning" means we can choose where we observe data in a domain, and "different modalities" means there are fundamentally different ways to observe the data. For example, an MRI machine can have many different coils in it, and for each coil we can measure data at a user-specified frequency. The fact that there are different coils with different physical properties corresponds to having "different modalities". The fact that we can choose which frequencies to measure data at corresponds to "active learning".

The paper studies an extremely general formalization of active learning in the $\ell_2$ norm with different modalities in active learning. Specifically, it suggests computing a "Generalized Christoffel Function" (i.e. leverage score function) for each modality, and to sample from each modality at random places, chosen with probability proportional to the generalized Christoffel function.

The paper allows for an exceptionally general notion of "sampling", where a sample could entail something classical in numerical linear algebra, like observing a row from a matrix. It can also mean something broader, like observing the gradient of a function. Also, while the error in each modality is measured with respect to the $\ell_2$ norm, it is the weighted $\ell_2$ norm with respect to a given measure $\rho$ for each modality. So, the user can choose to have (e.g.) a uniform error metric on a small interval, or to instead have a Gaussian-weighted error measure on the real line.

Lastly, the framework also has enough flexibility to give guarantees in some nonlinear spaces, with the main example being the union-of-subspaces.

Theoretical guarantees on sample complexity are given, roughly matching the rates that would appear in the special case of leverage score sampling for matrices. Further, many experiments and concrete instantiations of the general framework are presented. Applications vary from approximating function with multivariate polynomials, to optimizing MRI reconstructions, to solving PDEs with physically informed neural nets.

**Strengths:**

The paper is also 40 pages of really intense serious research. The appendix is chalk full of interesting special case studies, demonstrating the flexibility of their theory and the practical algorithms it suggests. The paper is so big and full of ideas that the NeurIPS peer review system, where I don't have enough time to read it cover-to-cover, really does this paper a disservice. I'd love to have 2 months to focus on this paper and understand the ins-and-outs of every case study and theorem. However, I don't have that time, so I'll presume the correctness of everything in the appendix and strongly recommend publishing this paper.

The paper is exceptionally general, doing a great job of generalizing much of the recent research on leverage score sampling beyond matrices. Lots of existing work has slowly generalized one or two ideas from leverage score sampling to Hilbert spaces or non-linearity or whatever else fits that work's setting. This paper really goes the extra mile and writes out a ridiculously general setting that covers:
- Fitting a polynomial to a function, where we can observe both the function and it's gradient
- Fitting a polynomial to a function, where we can choose if we want to observe the function or it's gradient at each query (in case one is more expensive than the other, we can consider them separate modalities with separate sample complexities).
- Iteratively refined approximations to solving a PDE
- Reconstructing MRI data from observations at carefully chosen frequencies

The general framework is phrased in terms of normed spaces, Hilbert spaces, measurable functions, and other abstract math that really gives the user freedom to do what the want. The theorems also apply to this extremely general framework, so the user immediately has Christoffel sampling guarantees with a likely near-optimal sample complexity out-of-the-box.

I think this paper is original in it's powerful results in extreme generality, and is general in it's applications. The paper's results are high quality sample complexity guarantees. The paper is broadly well written (but this is a bit of a weaker point). The paper should be significant in its expressive power owing to its applicability to many domains.

I strongly recommend publishing it.

**Weaknesses:**

The paper does suffer a bit on the clarity front. I'm familiar with leverage score sampling (and even some infinite-dimensional generalizations thereof), so I'd think that a generalized Christoffel function framework would be easy enough to understand. This was not the case though, and the extremely general framework was daunting an unapproachable to read about.

There's even an appendix subsection devoted to relating the extremely general framework back to the leverage score form I'm comfortable with. However, this description isn't super clear and even has (as far as I can tell) several minor errors which really harm legibility.

The general framework (section 2.1) is just a list of math definitions stated without any running example of what each mathematical object is. It's very unintuitive what the difference between a "measurement domain" and a "measurement space" are [Lines 85-87]. I'd strongly recommend the authors decide on a running example to present alongside all of the math definitions, which would make such definitions more intuitively distinct and clear.

The experiment and case study examples (section 3) are also pretty hard to approach. Section 3 consistently summarizes ideas at an extremely high level, and pushes the rigorous instantiations of the general framework to the appendix. While this makes sense from a page-limit perspective, it does harm the clarity and reading experience. Unfortunately, under the constraints of the NeurIPS format, I'm not sure a better option is available. Perhaps the authors can avoid this extreme appendicizing in the arXiv version of their paper?

Even when reading the appendices in good detail, there's bits and pieces that I don't follow (especially in the MRI example). Everything I understood was clearly correct, but sometimes the technical domain-specific language gets a bit overwhelming making the examples hard to follow.

This clarity point is not a game-breaking issue, but another pass at making the technical material more approachable would benefit the paper, especially when it's targeting such a broad audience of both theoretical and empirical data scientists.


_As a side note, it's curious to note one particular limitation of the framework -- it cannot explicitly handle ridge regression and ridge leverage scores, which is essential for some prior work on infinite dimensional leverage score sampling, like paper [12] in the references. I don't see any reason why CS4ML couldn't handle ridge regression / ridge leverage scores though._

**Questions:**

First, I have some moderately technical qualms with the paper:
1. In Figure 1, if I understand Appendix B.6 correctly, is it correct that MCS (monte carlo sampling / non-christoffel sampling) is performing worse with more samples because of numerical instability / poor conditioning? Or, like [Line 194] says, is it because the sample complexity of MCS sampling is so bad that the error.... _gets worse_ as the sample complexity grows?
1. The Appendix doesn't seem to have a fully accurate and consistent instantiation of leverage scores into the CS4ML framework. Specifically, $\mathbb{U} = \mathbb{C}$ on [Line 672] seems at odds with $\mathbb{U} \subseteq \mathbb{X}$ when $\mathbb{X} = L\_\rho^2(D)$. I'm pretty sure that $\mathbb{U} = \text{span}(A)$ in leverage score sampling? Also, it feels pretty inconsistent to describe $\mathbb{V}$ as a subset of $\mathbb{R}^N$ but to describe $\mathbb{X}$ as a weighted $L_2$ function space on $[n]$. Understanding that much is already non-obvious from the notation, but would be much clearer if you just wrote $\mathbb{X}=(\mathbb{R}^N, \|\|\cdot\|\|\_{\mathbb{X}})$ where $\|\|\vec v\|\|\_\mathbb{X}^2 = \frac1N \|\|\vec v\|\|_2^2$. Also, there's no reason (afaik) for $\mathbb{Y}$ to be complex instead of real. Further, the sampling function should more explicitly be about looking up an item from a vector, something like $L(y)(\vec u) = \sqrt{p(y)}u(y) = \frac{1}{\sqrt N} [u]\_{y}$. Lastly, from my algebra, I found that $\alpha=\beta=\frac1N$ in equation (2.4) for the leverage score setting. Did I make an error, or is there an adjustment to the instantiation that can instead make $\alpha=\beta=1$?

---

After that, I've got smaller qualms / typos / recommended edits:
1. [Line 92] The choice to make $(D_c,\mathcal{D}_c)$ makes for two symbols which look identical at a first (and second) glance. I'd change at least one of these symbols so it's clear that the first and second items in the pair are different.
1. [Line 116] Add parenthesis around 2.1
1. [Line 142] Remove $K=$
1. [Line 156] Why isn't there rescaling by $\sqrt{p(y)}$ here, like there was in the leverage score appendix's definition of $L$?
1. [Figure 1] Explain the confidence intervals in the caption
1. [Line 168] The placement of "See Section 4.3" is not great, since Section 4.3 is about translating an "in-probability bound" into an "expectation bound". I'd move "See Section 4.3" one sentence earlier.
1. [Line 226] Redefine PINN here. I know it's defined in the abstract, but I'd do it again here.
1. [Figure 3] Add titles to the plots
1. [Lines 230-239] I completely failed to understand what this paragraph is trying to say.... I just don't understand the problem or how it's being solved.
1. [Section 4] Swap $\varepsilon$ and $\delta$. I'd **very heavily** recommend this. The choice of $\delta$ to mean failure probability and $\varepsilon$ to mean accuracy is extremely standard. I would not mess with this; it's very confusing to read.
1. [Line 269] Interpret this big messy norm $\|\|\|\cdot\|\|\|$ somewhere. It looks scary, but I don't think it really should be that scary?
1. [Line 269] Missing $c$ subscript on $\nu$.
1. [Equation 4.1] Explore this sampling complexity. What is it really saying, comparing our sampling distribution to the Christoffel function? Can we use it to get a sorta "coherence assumption" for uniform sampling / MCS? This should be somewhat intuitive, but it really stands to be formally explored a bit more in the writing.
1. [Lines 271-272] Why is it "particularly relevant"? Why's it hard to understand the Christoffel function of the cover but easy to understand the Christoffel function of $\mathbb{U}$. I sorta intuitively believe it's true, but you should justify it.
1. [Line 311] Consider calling $\mathcal{T}_\theta$ a "shrinking operator" maybe? or a "regularization operator" or something? It's not very truncation-like imo, but it's really a small preference. This is a very mild/soft recommendation.
1. [Lines 316-318] Please formalize this added-penalty approach. It's the generalization of ridge regression, which feels very worth exploring.

---

> ### Author Rebuttal · Authors · 2023-08-09
>
> We thank the referee for their excellent review.
> ## Weaknesses
> ### 'The paper...'
> We agree. **Please see our global rebuttal for discussion and changes we will make.**
> ### 'There's even an....'
> **We will edit and move parts of this to the main paper.** See below for details.
> ### 'The general framework....'
> A running example is an excellent idea. We think the best example is the classic regression problem. **Please see our global rebuttal.** Regarding the terminology: the 'measurement domain' is how we enumerate the possible measurements. In classic regression, it is the domain of the function. In Fourier sampling (Example 2) it is the set of possible frequencies. The 'measurement space' is the space where the measurements lie. E.g. $\mathbb{R}$ for classic regression. **We will clarify this.**
> ### 'The experiment....'
> We agree that this can pose challenges. Our goal in this work is to introduce a general framework, then present three diverse examples to show its broad applicability. Unfortunately, it is hard to describe each example in the page limit, hence we pushed the details to the appendices. **The idea to reorganize the arXiv preprint is an excellent one. We will do this.**
> ### 'This clarity....'
> **We will edit the appendices for readability.**
> ### 'As a side...'
> Thanks for raising this interesting issue. We are inclined to agree, but would need to investigate further as the proofs in [12] are quite different to ours. This is an interesting aim for future work.
> ## Questions
> ### 1.
> Exactly. The worse performance arises from instability due to poor conditioning. As shown in Fig. 4, cond$(A)$ blows up with $m\approx n\log(n)/(d+1)$. In Fig. 5 we compute how large $m$ should be, given $n$, to ensure a cond$(A)\le tol$. As shown, $m$ needs to grow much more rapidly than $n\log(n)/ (d+1)$ for this to occur.
> ### 2.
> Thanks. We think there are two issues here. First, relating (A.1) to Definition 2.4. This is described in lines 672-674, but there were several errors: i) $\mathbb{U}=\mathbb{C}$ in line 672 should have been deleted, ii) it was not stated that $D$ should be equipped with the reference measure $\rho$ (e.g. the Lebesgue measure) that defines the density $p$ via the Radon-Nikodym derivative. There was an unfortunate notation clash, since we also used $\rho$ for the probability measure with density $p$. We will change this probability measure to $\pi$, so that $p=d\pi/d\rho$. iii), in l.674, it should be $\tau(\mathbb{V},p)(y)=K(\mathbb{V})(y)$. **We will fix this discussion.**
>
> Second, relating (A.2) to (A.1). We did this in lines 669-671. However, we agree it is easier to directly relate it to Definition 2.4 (a benefit of the general definition). This can be done in several ways, but perhaps the easiest is to set $\mathbb{X}=(\mathbb{R}^N,\|\|\cdot\|\|_{2})$, $\mathbb{Y}=(\mathbb{R},|\cdot|)$, $D=\{1,\ldots,N\}$ with the uniform measure $\rho$ and $L(y)(u)=[u]_y$. Then, with $\mathbb{V}$ as in line 669,
>
> $$K(\mathbb{V})(y)=\max_{v\in\mathbb{V},v\neq0}\frac{\|\|L(y)(v)\|\|^2_{\mathbb{Y}}}{\|\|v\|\|^2_{\mathbb{X}}}=\max_{x\neq0}\frac{[Ax]^2_y}{\|\|Ax\|\|^2_2},$$
> which is precisely (A.2). Note that in this case, nondegeneracy holds with $\alpha=\beta=1$, since $$\int_D\|\|L(y)(x)\|\|^2_{\mathbb{Y}}d \rho(y)=\sum^{N}_{i=1} (x_i)^2.$$
> We get $\alpha=\beta=1$ since we do not impose that $\rho$ is the uniform probability measure. **We will add this derivation.**
> ## 'After that....'
> ### 1-3.
> **We will fix these**.
> ### 4.
> Good question. It will hopefully be clear in the revision when we add the classic regression problem. Suppose that $\mathbb{X}=L^2_{\pi}(D)$ for some probability measure $\pi$ with density $p$. Then there are two ways to setup the problem:
>
> a) Consider $D$ with the measure $\rho$ that defines $p$ as $p=d\pi/d\rho$. Set $L(y)(u)=\sqrt{p(y)}u(y)$. Then
> $$
> K_a(y):=K(\mathbb{V})(y)=\sup\\{p(y)|v(y)|^2/\|\|v\|\|^2_{\mathbb{X}}:v\in\mathbb{V},v\ne0\\} =\tau(\mathbb{V},p)(y).
> $$
>
> b) Consider $D$ with $\rho=\pi$ and set $L(y)(u)=u(y)$. Then
> $$
> K_b(y):=K(\mathbb{V})(y)=\sup\\{|v(y)|^2/\|\|v\|\|^2_{\mathbb{X}}:v\in\mathbb{V},v\ne0\\}.
> $$
> Observe that $K_a(y)=K_b(y)p(y)$, so the two terms differ by $p(y)$. However, they lead to exactly the same Christoffel sampling measure $\mu^{\star}$ in (2.6). So the difference boils down to convention, i.e. whether to include $p$ or not. **We will explain this in the revision.**
> ### 5-8.
> **We will fix these**.
> ### 9.
> **We will rewrite this for clarity.**
> ### 10.
> **We will change this.**
> ### 11.
> **We will add this (likely in the appendices)**
> ### 12.
> **We will fix this.**
> ### 13.
> Good point. We think the best way to do this is to compare with MCS. In MCS, (4.1) involves $\mathrm{esssup}_{y\sim\rho_c}K_c(\mathbb{V})(y)$, whereas with CS it involves  $\kappa_c(\mathbb{V})=\int K_c(\mathbb{V})(y)d\rho_c(y)$. Thus, the difference relates to the maximal behaviour of $K_c(\mathbb{V})$ versus its integral (i.e. mean). If it is very flat, these bounds are similar. If it has spikes/peaks, they may differ substantially. An instance of this can already been seen in Fig 2, where $K$ is sharply peaked near the zero frequency (in the middle) and decays as frequency increases. **We will add this intuition.**
> ### 14.
> Good question. Our thought is that it may be possible to show that there is a cover, but hard to explicitly formulate it, and therefore difficult to sample from (4.2). Generative models are an example. The range of a ReLU generative model is contained in a union of subspaces (lines 1009-1015). But these subspaces are likely not easily quantified. And even if they were, since there are a lot of them, it would be computationally intractable to evaluate the Christoffel function over their union. However, this point was poorly explained. **We will rephrase this sentence.**
> ### 15.
> **We will change this.**
> ### 16.
> Apologies. We now don't think it is possible to get the same error bound using a regularized estimator. **We will delete the sentence.**

---

> > ### Comment · Reviewer_n7M7 · 2023-08-21
> > **Thanks for the response!**
> >
> > Apologies for the delay in my response.
> >
> > This was a delightful response from the authors, and I remain very very inclined to accept this paper. I look forward to reading this again, with updated notations and intuitions.
> >
> >
> >
> > As an additional side note, it may be interesting to see if the infinite-dimensional work of [[Shustin Avron]](https://arxiv.org/pdf/2104.05687.pdf) on leverage scores for general quasimatrices can be formalized as a special case of your results (when not regularizing so that $\lambda = 0$). No pressure to actually include it, but might be cool to look at, and also might provide proof strategies for ridge regression that work even if [12] doesn't provided anything immediately useful.

---

> > > ### Author Response · Authors · 2023-08-21
> > > **Thanks!**
> > >
> > > Thank you for your kind words and for the link to the interesting paper. We will certainly look into this.

---

### Official Review · Reviewer_7Xvg · 2023-07-12

**Soundness:** 1 poor
**Presentation:** 1 poor
**Contribution:** 1 poor
**Rating:** 3
**Confidence:** 2

**Summary:**

This paper proposes a general framework for active learning. It claims that the proposed method can help achieve near-optimal sample complexity. In additional to the theoretical results, this paper also present 3 use cases, showing favorable results for the proposed method.

**Strengths:**

Important topic with claimed good results.

**Weaknesses:**

1. Poor presentation: Very heavy theoretical setup buried the connection to solving any real active learning problem. No high-level description of the algorithm and its advantages over the existing active learning algorithms. No insights are provided based on the theoretical derivation.
2. Poor experiment comparison: The proposed method was only compared with random sampling. There's no comparison with any existing active learning method.

**Questions:**

1. line 19, p. 1: In ML literature, features are represented by $x_i$ instead of $y_i$ while the latter denotes the label, which is $f(y_i)$ in your case.
2. line 38, p. 2: In your example of computational imaging, is one image an example or one pixel an example? Both cases can occur in real applications.
3. line 60, p. 2: Do you mean the label function $f$ is in a Hilbert space? For any function with a set of real-valued parameters (e.g., a neural network), if it's represented by a vector of its real-valued parameters, is such a function in a Hilbert space?
4. line 61, p. 2: What does "data" in "data arises" mean? Examples of features and label pair?
5. line 179, p. 5: You claim "there are many settings where one can acquire both function values and values of its gradient." Can you give some real examples of such settings?

---

> ### Author Rebuttal · Authors · 2023-08-09
>
> We thank the referee for their informative feedback.
>
> ## Weaknesses
> ### 1. 'Very heavy theoretical...'
> The heavy setup was also noted as a weakness by other reviewers. **As discussed in our global rebuttal, we will edit Section 2 to improve this presentation. In particular, we will add the running example of the standard regression problem.** This should greatly improve the presentation and help connect it to what the reader is familiar with.
>
> ### 'No high-level...'
> We agree that the paper lacks a high-level description of the algorithm. However, what we propose is more of a 'framework' than an 'algorithm'. We do think Theorem 2.5 neatly summarizes the main facets of the approach. **If the referee thinks this can be made clearer, we are happy to hear suggestions.** The reason we do not consider this an 'algorithm' is because there is no general prescription for how to sample from the optimal measure (2.6). This is the main computational hurdle to tackle in practice. As our examples show, how to do this (or, indeed, whether it is possible) is very problem dependent. Nonetheless, this is a important point. **We will add a sentence to the conclusion about this.**
> ### 'and its advantages...'
> In terms of the advantage over other active learning strategies, we are currently unaware of any work that addresses the broad class of problems that our framework does. **Please see our global rebuttal for more discussion on this.**  Our work essentially shows that one can extend well-known leverage score sampling to a very broad class of problems, with theoretical guarantees. **We will add a sentence to this effect in Section 1.2.**
> ### 'No insights are...'
> We already give several insights in the paper. Directly below Theorem 2.5 we explain how one obtains near-optimal log-linear sample complexity whenever $\mathbb{U}$ is a subspace, or a union of a few (i.e., $d \ll \infty$) subspaces. This insight explains how our work extends classical leverage score sampling to a much more general setting. Further insights are also given in Remark 4.5. Finally, in Section 5 we return to the union-of-subspaces case with further discussion. Section A.4 (mentioned therein) also gives additional insight.
>
> We agree, however, that these theoretical contributions are hard to parse and spread over too many sections. **Referees tt34 and n7M7 also had some suggestions to improve this presentation, which we will implement. We will (i) move the discussion on sample complexity in Section 5 (lines 331-335) and consolidate it into an expanded discussion directly below Theorem 2.5, and (ii) add a brief comparison of the sample complexity bound for CS versus MCS in Section 4.2, demonstrating how this corresponds to the difference between the mean value of the Christoffel function and its maximum.**
> ### 2.
> This is a good point. **Please see our global rebuttal for discussion of this comment and the actions we intend to take.**
> ## Questions
> ### 1.
> We agree. This was also noted by reviewer tt34, along with other notational comments. We will change this to $y_{ic} = L(\theta_{ic})(f^*) + e_{ic}$. We prefer to use $\theta$'s to represent the measurement location: $x$'s often denote spatial locations, whereas in our work they could be, e.g., frequencies.
> ### 2.
> Good point. The short answer is neither. In Fourier imaging, an 'example' is a frequency. Adopting the notational changes proposed in our rebuttal to reviewer tt34, in this problem we wish to reconstruct a (vectorized) target image $f^* \in \mathbb{C}^N$ from measurements $y = A f^* + e$. Here $A = P_{\Omega} F$ is a subsampled Fourier matrix, where $\Omega \subseteq \{1,\ldots,N\}$ is a set of $m$ frequencies. In the active learning problem, we wish to select $\Omega$ (i.e., select frequencies) to recover $f^*$ as well as possible from $y$. This is detailed in Appendix C, but we agree it could be clearer both there and in Section 3.2 of the main paper. **We will add a few sentences for clarity, including the above explanation of what an example is in this context.**
> ### 3.
> Yes, the target object $f^*$ is an element of the (abstract) Hilbert space $\mathbb{X}$. In classical regression (see lines 19-22), $\mathbb{X} = L^2_{\rho}(D)$ for some measure $\rho$, i.e., the space of square-integrable functions over $D$ wrt $\rho$. The reason for considering abstract Hilbert spaces is motivated by applications where the target object may not be a scalar-valued function. For example, in parametric PDE learning problems in UQ (lines 47-48), the target object is a function that takes values in a Hilbert space. In this case, $\mathbb{X}$ is a Bochner space.  Similarly in operator learning, the target object is an operator, not a function. **As we mentioned above, we will add some sentences in Section 1.2 to better motivate this generalization. We will also add the classic regression problem as an example in Section 2.**
>
> We are not sure what the referee means by their question. If the function $f_{\theta} : D \rightarrow \mathbb{R}$ was square-integrable over its domain then it would belong to the Hilbert space $L^2(D)$. This doesn't seem restrictive, e.g., it holds for neural networks with (piecewise) continuous activation functions over compact domains. **We would welcome a clarification of this question.**
> ### 4.
> Good point. As stated in line 19 (modulo the new notation), in the classical setting by 'data' we mean pairs $(\theta_i , f^*(\theta_i))$. In our general framework, the data is pairs $(\theta_{ic},L_c(\theta_{ic})(f^*))$. The main point is that rather than function samples, i.e., $f^*(\theta_i)$, of the target object, we consider linear operators, i.e., $L_c(\theta_{ic})(f^*)$. We agree this was not sufficiently clear. **We will edit lines 96-100 to clarify exactly what the training data is.**
> ### 5.
> These were mentioned in lines 46-50. However, we agree the reader may well have forgotten this by line 179. **We will add a reference back to Section 1.1 in line 179.**

---

### Official Review · Reviewer_tt34 · 2023-07-13

**Soundness:** 3 good
**Presentation:** 2 fair
**Contribution:** 3 good
**Rating:** 7
**Confidence:** 3

**Summary:**

The paper proposes a framework for active learning (here meaning that the user controls the sampling strategy according to which the locations of the measurements are made) designed to handle various cases of regression (vector-valued, multimodal, and more).

To do so, it introduces the concept of generalized Christoffel functions, that will drive the choice of sampling strategies.
A statistical analysis of the risk associated to the empirical risk minimizer associated to the square loss is conducted, showing that a log-linear number of observations (with respect to quantities related to the geometry of the hypothesis space) allows for efficient training using the Christoffel sampling strategy.

Numerical experiments on three different problems (all cast inside the cs4ml framework) complement the approach, showing improvements over the naive Monte-Carlo sampling.

**Strengths:**

- The proposed approach is novel and encompasses many valuable problems to the machine learning community.
- Mathematical details are sound.
- Experiments show clear improvements.
- The appendix contains a lot of remarks, additional experiments, that further develop the framework.
- I share the author's enthusiasm regarding the potential of the framework, which is far from explored.

**Weaknesses:**

- It sometimes feels that the paper is overly complicated for the sake of generality, without benefits (see questions).

- While fairly general, the framework assumes that the user has access to the data generation strategy (not the targets, but the inputs) which is often not the case in practice.

- Overall, the paper is very dense and notation heavy. It is impossible to read it without a constant back and forth to the (31 pages long) appendix, which makes the reviewer wonder why the 9 pages limit exists for this conference. In my opinion, it would be beneficial to have a longer (i.e. journal) version of the paper, as the supplementary contains many interesting remarks that help a lot to the understanding of the ideas.

**Questions:**

I found the notation to be quite heavy and not intuitive at all. I suggest to change the name convention adopted here - while all notation choices are arbitrary, they need to be driven by simplicity and logic. In particular:

- I suggest adopting $x_{ic}$ for the measurement locations, to be conform to the classical $y = f(x) + \epsilon$ regression statistical model. Anything evoking something we usually put inside a function call or put measures on is ok ($\theta_{ic}$ ?).
- It feels weird not to have the random variable $Y_c$ living in the space $\mathbb{Y}$. Measurements could be $y_{ic}$ instead of $b_{ic}$.
- To lower the number of different letters, the true function associated to the generating process could be $f^\star$, and the candidates could be $f$ (instead of $u$), keeping the estimate $\hat{f}$.
- In the regression literature, hypothesis spaces are usually referred to as $H$ or $F$. If the measurements locations are $x$, the name of the object space has to change as well.
- In definition 2.4, you introduce yet another variable name $\mathbb{V}$ that could just be the variable name of the hypothesis space, to improve clarity.

Other questions/comments:

- Where do you use non-boundedness of the measurement operators ? Could you not use the standard notation that elements of $\mathcal{L}(\mathbb{X}, \mathbb{Y}_c)$ are bounded linear operators ? Usually unbounded operators are not defined on the whole space $\mathbb{X}$ but only on their domain.

- Could you comment on the need to parameterize the measures using two other measures ? $\mathrm{d}\mu_c(y) = \nu_c(y) \mathrm{d}\rho_cc(y)$ ? In the end, we first choose $\rho$ then take $\nu$ according to the CS strategy, so what is the influence different choices of $(\rho, \nu)$ yielding the same $\mu$ ?

- Empirical nondegeneracy as presented in Eq. 2.5 is a void concept, unless $\alpha' \approx \alpha$ and $\beta' \approx \beta$ are made precise.

- In Theorem 2.5, the symbol you use for the estimator is not defined until way too long after.

- The notion of optimal sample complexity is not well explained enough and I struggle to see exactly with respect to what it is optimal. How can it be optimal given the crude estimate for the sum of Christoffel functions ?

- Section 3.1: Have you studied the impact of the choice of $\rho$ on numerical performances ? What if $\rho$ were a laplace measure ? How do you choose $f$ in the experiments ? Also, why not write $(y_i, f(y_i), \nabla f (y_i))$ ? This avoids the supercharge of the subscript notation in line 752.

- Proposition 4.3, 4.4 could be renamed corollaries or examples.


Typos:

- Line 61: vector-valued instead of vector space valued
- In assumption 2.2, the mappings $L_c$ are from $D_c$ to $\mathbb{Y}_c$. (also in definition 2.4, check the whole text)
- In Eq. (2.2) and (2.3) it should be $\forall u \in \mathbb{U}$.
- "Hence, the sampling operators and measures preserve the X-norm in expectation" is very vague.
- Appendix B1: line 759: no need for the complex conjugate. Why do you need to work in C after having defined your function as R-valued ? Line 763 rho is the same measure for all indexes k.

**Limitations:**

Most limitations are discussed in the paper. I would also like to mention that the cs4ml framework is not suited to all regression problems, but only to those where one controls the sampling strategy. Alas, in practice you seldom know $\rho$, because the data distribution is often unknown. (Correct me if I'm wrong)

---

> ### Author Rebuttal · Authors · 2023-08-09
>
> We thank the referee for their excellent feedback.
> ## Weaknesses
> ### Bullet 1
> We agree that the framework is complicated. This was raised by other referees and we have discussed in our global rebuttal. In brief, we believe it is justified given the range of applications we address. However, we also agree this justification needs to be more clearly made. **Please see our global rebuttal for the changes we will make in this regard.**
> ### Bullet 2
> We are not sure what the referee means here. Our main assumption is that, for each $c$, we can query $L_c(y)(f)$ for any $y \in D_c$. This is essentially the same assumption as in standard active learning, where $L_c(y)(f) = f(y)$. Note this assumption holds for Examples 1-3.
>
> We agree that this is worth clarifying. **We will add this as an Assumption in Section 2 and remark that it holds for our examples.** The addition of the classic regression problem to Section 2 will also help clarify this point.
> ### Bullet 3
> This issue was also raised by other referees. We agree, and we will address this. **Please see our global rebuttal.** Regarding a journal version, referee n7M7 suggested that we edit the arXiv preprint so that it was not so appendix heavy. **We will do this.**
> ## Questions
> **We agree, and we will change this as follows.** We will rename the measurement locations as $\theta_{ic}$. We prefer not to use $x_{ic}$, since $x$ is often used to denote a spatial locations, whereas our samples could be, e.g., frequencies. $\Theta_{ic}$ will the corresponding random variable, instead of $Y_{ic}$. The measurements will be $y_{ic} = L(\Theta_{ic})(f^*) + e_{ic}$. We will write $\mathbb{F}$ for the hypothesis set, with elements $f \in \mathbb{F}$, object $f^*$ and estimator $\hat{f}$. We will also change Definition 2.4 to use $\mathbb{F}$.
> ## Other questions/comments
> ### Bullet 1
> This was done to allow for the pointwise sampling operator $f \mapsto f(y)$, which is unbounded over $L^2_{\rho}(D)$. However, we agree it is problematic. After considering it, we have concluded it is best to assume that $f^*$ and $\mathbb{F}$ lie in some normed vector subspace $\mathbb{X}_0$ of $\mathbb{X}$ and define $L_c : D_c \rightarrow \mathcal{B}(\mathbb{X}_0,\mathbb{Y}_c)$, so that $L_c(\theta_c)$ is a bounded linear operator. In the case of pointwise sampling, one can, e.g., let $\mathbb{X}_0$ be the space of continuous functions on $\bar{D}$. **We will change this.** It does not alter the main results/proofs significantly.
> ### Bullet 2
> Good question. For each $c$ the set $D_c$ generates the samples as $L_c(\theta_c)(f^*)$ for $\theta_c \in D_c$. The approach pursued in this paper is to find an ‘optimal' probability measure $\mu_c$ on $D_c$ (in the sense of sample complexity). To do this, we first assume there is a measure space $(D_c,\mathcal{D_c},\rho_c)$. Then in Assumption 2.1 we assume that the candidate $\mu_c$'s are absolutely continuous wrt $\rho_c$ with Radon-Nikodym derivatives $\nu_c$ that are positive a.e. This is done so that the search for the optimal $\mu_c$ is reduced to finding an optimal a.e. positive function $\nu_c$ that satisfies $\int_{D_c} \nu_c(y) d \rho_c(y) = 1$. We do this by showing (e.g. Theorem 4.2) how $\nu_c$ affects the sample complexity, and then optimize this bound (Section 4.2).
>
> Note that $\rho_c$ is often not something we can choose. For example, in standard regression, $\mathbb{X} = L^2_{\rho}(D)$ and $L(y)(f) = f(y)$. One can think of $\mathbb{X}$ as ‘the space in which we measure the error’, in which case $\rho$ is usually dictated to us. In particular, even though we draw samples according to $\mu$, the error is still measured wrt $\rho$. Typical examples here are the uniform measure on bounded domains or the Gaussian measure on $\mathbb{R}^d$ (as we use in Example 1).
>
> It is worth explaining this. As noted, **we will add the classic regression problem as an example in Section 2. We will also comment on $\rho$ and $\mu$ therein.**
> ### Bullet 3
> **We will change this to say 'empirical nondegeneracy holds with constant $0 < \delta < 1$ if (2.5) holds with $\alpha' = (1-\delta) \alpha$ and $\beta' = (1+\delta) \beta$'.** This is what we prove in Theorem 4.2.
> ### Bullet 4
> **We will move the definition of $\check{f}$ in Theorem 4.8 earlier in the paper.**
> ### Bullet 5
> For a single ($d = 1$) or for a small number ($d \approx 1$) of subspaces the bound (2.8) is near-optimal, i.e. log-linear in $n$. However, it is highly suboptimal for $d \gg 1$. We noted this in lines 331-335. However, we agree the reader may well miss this. **We will move this discussion to directly after Theorem 2.5 to clarify.**
> ### Bullet 6
> Good question. We have not done this. For the applications that motivated this example, namely regression problems in UQ, it is typical to use the Gaussian measure. We have not seen the Laplace measure used in such applications before. An advantage of the Gaussian measure is that the orthogonal polynomials are tensor-products of the 1D Hermite polynomials, and we also have explicit expressions for their gradients. This allows us to solve the training problem numerically (see Sections B.2-B.4). Note that $\rho$ also dictates the norm in which the error is measured, i.e., the $L^2_{\rho}$-norm. Therefore, it is unclear how best to compare different choices of it.
>
> The test function in Figure 1 was chosen as a simple example to demonstrate the main effects. We could show other test functions, with similar outcomes. **If the referee thinks this would be beneficial, we can add these to the Appendices.**
>
> **We agree about the notation and will change it.**
> ### Bullet 7
> **We will change them to corollaries.**
> ## Typos
> ### Bullets 1-3
> **We will fix these.**
> ### Bullet 4
> **We agree and will delete this.**
> ### Bullet 5
> **We will correct these typos.**
> ## Limitations
> We agree, but unless we have misunderstood, we believe that this is true of active learning in general. **As noted above we will add this as an assumption in Section 2.**

---

> > ### Comment · Reviewer_tt34 · 2023-08-19
> > **Acknowledging the rebuttal**
> >
> > I thank the authors for their detailed answer. I have no doubt that the paper's quality will be greatly improved once the proposed changes will be implemented, and I have raised my score accordingly.

---

> > > ### Author Response · Authors · 2023-08-21
> > >
> > > Thank you for your positive response!

---

### Official Review · Reviewer_9pYz · 2023-07-15

**Soundness:** 4 excellent
**Presentation:** 4 excellent
**Contribution:** 4 excellent
**Rating:** 7
**Confidence:** 1

**Summary:**

The authors introduce an active learning framework for regression problems based on the concept of generalized Christoffel functions. The proposed approach is applicable to a broad range scenarios and it is evaluated on several scientific computing tasks.

**Strengths:**

The manuscript present a comprehensive explanation over the main ideas behind the proposed CS4ML framework. The main motivation is relevant (active learning for diverse data scenarios) and the authors point promising directions for further investigations.

The work is well organized, which is especially important for theory focused contributions. I also praise the detailed provided supplementary material.

**Weaknesses:**

The experimental section, although tackling diverse tasks, unfortunately compares the proposal only to an inactive learning strategy (Monte Carlo sampling), which diminishes the relative impact of the introduced method. I appreciate this point being highlighted as a limitation by the authors, but it does leave the reader wanting a bit more.

**Questions:**

Although the manuscript lists 118 references, I think it lacks a more thorough in-text discussion with other available active learning strategies. Which literature gaps are addressed? Which scenarios are not entirely covered by standard methods?

As a final minor observation, the left plot in Fig. 2 present some difficult to distinct line patterns which could be improved.

**Limitations:**

The manuscript states sufficiently its limitations.

---

> ### Author Rebuttal · Authors · 2023-08-09
>
> We thank the referee for these excellent comments. We will discuss them in reverse order:
>
> ### ``Although the manuscript....''
>
> In terms of active learning, the main contribution of this article is to extend certain active learning techniques (namely, leverage score sampling) to much broader types of measurements.
> The majority of past work on active learning considers pointwise samples of a target function. In fact, we are unaware of any works that systematically tackle more general types of measurements like we do in this paper. This is the main gap in the literature we aim to address.
>
> We fully agree, though, there should have been a more thorough in-text discussion of the active learning literature in the paper. Some of this discussion can be found in Appendix A. **As also discussed in our global rebuttal, we will move parts of this to Sections 1 and 2 of the final paper. We will also add more discussion as relevant.**
>
> ### ``The experimental section,...''
>
> For the reasons described above, we are not aware of other active learning techniques for which we could make a 'global' comparison across all three examples. As we commented briefly in the conclusion section (Section 5), there has been much previous research on problem-specific active learning strategies that outperform Monte Carlo sampling. However, the main aim of our article is not necessarily to achieve state-of-the-art performance in each example. Rather, it shows how a single active learning technique (Christoffel sampling), which also comes with theoretical guarantees, can improve against inactive learning across a broad spectrum of problems.
>
> As we mentioned in our global rebuttal, it is important to compare different strategies. But we think such a comparison is well beyond the scope of the current article. It is likely that comprehensive comparisons in each of our three applications could each make for paper on their own. For example, there have been a slew of recent papers on sampling strategies for PINNs (Example 3 in our paper). We cite these in Section 5. To perform a robust comparison, we would need to consider each method in comparison to CS across a range of different PDEs, rather than just the Burger's equation problem we consider in this article to show proof-of-concept. This is certainly worth doing, due to high level of current interest in PINNs for PDEs. Similarly in MRI, there has been a large amount of past research on sampling strategy design (we also cite this in Section 5). A thorough comparison would need to consider different datasets -- i.e., not just brain images, but other types of common MR image datasets such as knee or abdomen images -- and different MRI modalities (i.e., single versus parallel MRI), as well as different generative neural network architectures.
>
> As also mentioned in our global rebuttal, another interesting question for future work is whether one could use CS as a starting point for devising even more advanced active learning methods. In general, this could involve using either the linear-sample sparsification techniques of [32], as briefly mentioned in Section 5, or by using 'boosting' techniques (Haberstich, Nouy \& Perrin). Or, there could be domain-specific tools one could employ to improve the performance of CS in a particular application (e.g., PINNs or computational imaging).
>
> In summary, we think that these are very interesting questions for future work, but that they are well beyond what we can feasibly achieve in this article. Nonetheless, it is worth elaborating on this matter in the revision. **To address this issue, we will expand the discussion in Section 5 on other active learning strategies for the main examples and re-emphasize the main contributions of CS4ML to this problem.**
>
> ### ``As a final...''
>
> Good point. We will also improve the left plot of Figure 2.

---

> > ### Comment · Reviewer_9pYz · 2023-08-11
> >
> > I thank the authors for the answers. I believe the addressed issues and modifications proposed across all the reviews will greatly improve the understanding and the contribution of the work. Thus, I maintain my acceptance rating.

---

> > > ### Author Response · Authors · 2023-08-21
> > >
> > > Thank you for your kind words.

---

### Author Rebuttal · Authors · 2023-08-09

We thank the referees heartily for their insightful comments and the time and effort they put into carefully reviewing our manuscript. Each referee has made insightful comments that will undoubtedly improve the final version of the paper. We have provided detailed responses to each review separately below. However, in this global rebuttal we want to also discuss some of the main comments raised by the referees.

Note that in this and other rebuttals, **all specific changes we intend to make in the final article are in bold. We do not anticipate that these changes will exceed the one additional page allowed in the camera-ready version of the paper.**

### Experimental comparison/clarification of contribution (referees 9pYz, 7Xvg):

Several referees commented that we only compared the proposed method against inactive learning (Monte Carlo sampling) and not other active learning techniques. It is certainly the case that for each of our three main examples, there has been much previous research on problem-specific active learning strategies. We commented on these briefly in the conclusion (Section 5). However, the main aim of our article was not necessarily to achieve the absolute state-of-the-art performance in each example. Rather, it shows how a single active learning technique (Christoffel sampling), which also comes with theoretical guarantees, can improve against inactive learning across a broad spectrum of problems. In particular, we extend classical leverage score sampling to a much broader class of problems, while maintaining its theoretical guarantees. We are unaware of any other current active learning strategy that can simultaneously address such a broad class of problems. **We will add a sentence to this effect in Section 1.2 (Contributions).**

We very much agree, however, that it is important to compare different strategies. But we think such a comparison is well beyond the scope of the current article. It is likely that a comprehensive comparison in each separate application could make for a paper on its own. We are interested in doing this, especially to see how well CS performs against other techniques. A related, and interesting, question for future work is whether one could use CS as a starting point for even more advanced active learning methods, e.g., by using the linear-sample sparsification techniques of [32], as briefly mentioned in Section 5, or by using 'boosting' techniques (Haberstich, Nouy \& Perrin).

**To address this issue, we will expand the discussion in Section 5 (Conclusions, limitations and future work) on other active learning strategies for the main examples and re-emphasize the main contributions of CS4ML to this problem.**



### Dense presentation (referees tt34, 7Xvg, n7M7):

Several referees found the presentation of the framework in Section 2 dense and difficult to follow.

We agree and will make a series of changes to address this. First, **referees tt34 and n7M7 also proposed notational changes to improve readability, which we will implement. Second, as recommended by referee n7M7, we will add the classical regression problem as a running example in Section 2 (The CS4ML framework and main results) to clarify how this sits within the general framework.** This will make the generalizations we introduce easier to understand and motivate. **As also suggested, we will also move parts of the discussion on leverage score sampling from Section A.2  to Section 2.** Many readers are likely familiar with this leverage scores, therefore doing this will greatly aid readability. Overall, we believe these changes will make Section 2 much clearer for the reader.


### Complicated framework (referees tt34,n7M7):

Several referees commented that the framework we propose is complicated/daunting due to its generality without proper justification.

We agree that the framework we propose is very general and the notation can feel quite heavy. As noted above, **we will implement a number of notational changes for clarity.** In terms of the formulation of the framework, however, we think its generality is justified by the broad range of applications it can address.

The main ways our framework generalizes standard active learning is that i) we allow the target object to an element of an arbitrary Hilbert space $\mathbb{X}$, ii) we allow arbitrary linear sampling operators, and measurements which are scalar or vector-valued, and iii) we allow for potentially multimodal data. i) is a useful extension, and not too difficult to achieve. Both ii) and iii) are motivated by the examples. ii) allows one to consider sampling with, e.g., the Fourier or Radon transform, both of which are very common in computational imaging (e.g., MRI reconstruction in Example 2 of the paper). It also allows vector-valued measurements, which arise in both Examples 1 and 2 of the paper. Finally, multimodal data is found in many real-world applications, and is encountered in Example 3, as well as practically-important extensions of Examples 1 and 2 (see Sections B.7 and C.7, respectively).

We discussed these motivations in Section 1.1 (Motivations) and later in Section 3 (Examples and numerical experiments) when we describe the three examples. However, they are worth elaborating for clarity. **We will add a short discussion in Section 1.2 (Contributions) to explain lines 60-63 and thereby justify the generalizations. Further, as discussed above, we will also add the classic regression problem as an example in Section 2.** This will allow us to better justify the generalizations there as well.

---

### Decision · Program_Chairs · 2023-09-21

**Decision:**

Accept (spotlight)

**Comment:**

Almost all of the reviewers agreed that this paper was an interesting read, and studies a very general form of the active learning problem with a number of compelling applications. In particular, it considered many variants of data acquisition beyond simple point-wise function sampling. Its results generalize tools like leverage score sampling that have recently applied for the more standard notion of active learning, so should be very interesting to many in the NeurIPS community.